# Reduced-complexity model for impact of anthropogenic CO₂ emissions on future glacial cycles

Stefanie Talento[1], Andrey Ganopolski[1]

[1]Potsdam Institute for Climate Impact Research, Potsdam, 14473, Germany

*Correspondence to*: Stefanie Talento (talento@pik-potsdam.de)

**Abstract.** We propose a reduced-complexity process-based model for the long-term evolution of the global ice volume, atmospheric CO₂ concentration and global mean temperature. The model only external forcings are the orbital forcing and anthropogenic CO₂ cumulative emissions. The model consists of a system of three coupled non-linear differential equations, representing physical mechanisms relevant for the evolution of the climate – ice sheets – carbon cycle system in timescales longer than thousands of years. Model parameters are calibrated using paleoclimate reconstructions and the results of two Earth system models of intermediate complexity. For a range of parameters values, the model is successful in reproducing the glacial-interglacial cycles of the last 800 kyr, with the best correlation between modelled and paleo global ice volume of 0.86. Using different model realisations, we produce an assessment of possible trajectories for the next 1 million years under natural and several fossil-fuel CO₂ release scenarios. In the natural scenario, the model assigns high probability of occurrence of long interglacials in the periods between present and 120 kyr after present, and between 400 kyr and 500 kyr after present. The next glacial inception is most likely to occur ~ 50 kyr after present with full glacial conditions developing ~ 90 kyr after present. The model shows that even already achieved cumulative CO₂ anthropogenic emissions (500 PgC) are capable of affecting the climate evolution for up to half million years, indicating that the beginning of the next glaciation is highly unlikely in the next 120 kyr. High cumulative anthropogenic CO₂ emissions (3000 PgC or higher), which could potentially be achieved in the next two to three centuries if humanity does not curb the usage of fossil-fuels, will most likely provoke Northern Hemisphere landmass ice-free conditions throughout the next half million years, postponing the natural occurrence of the next glacial inception to 600 kyr after present or later.

## 1    Introduction

The long-term evolution of the Earth system is not only of theoretical interest but a matter of utter practical relevance at present, when its assessment is required in the decision-making process for guaranteeing the safe permanent storage of radioactive waste. The safe disposal of such materials constitutes one of the urgent environmental and societal challenges that mankind faces in the 21st century. Disposal in deep geologically-stable rock formations is the only currently viable option to ensure long-term safety (IAEA Report 2007, Kim et al., 2011). While low- and intermediate-level nuclear wastes require at least 100 kyr years of safe storage, for high-level nuclear waste and spent nuclear fuel safety for 1 Myr should be envisaged. The evaluation of geological disposal systems in response to future environmental changes, driven by a combination of natural and anthropogenic forcings is, therefore, mandatory.

Numerous paleoclimate records show that during the last 2.7 Myr the natural evolution of the Earth system has been characterized by persistent alternations of glacial and interglacial episodes. The last 800 kyr were dominated by strongly asymmetric glacial cycles with average periodicity of about 100 kyr, a rather slow glacial build-up and much more rapid deglaciations (Hays et al. 1976; Lisiecki and Raymo, 2005). Antarctic ice core records show that atmospheric carbon dioxide ($CO_2$) concentration fluctuated nearly synchronously with the global ice volume, and that $CO_2$ concentration during glacial times was up to 100 ppm lower than during preindustrial time (Petit et al., 1999, Lüthi et al., 2008). The astronomical theory of glacial cycles (Milankovitch, 1941) postulates that growth and shrinkage of ice sheets is primarily controlled by changes in Earth's orbital parameters (eccentricity, obliquity and precession) through their effect on the amount of solar radiation received at high latitudes of the Northern Hemisphere (NH) during boreal summer. Different aspects of Milankovitch theory have been tested and supported with climate and Earth system models of different degree of complexity (e.g. Pollard, 1983; Berger et al., 1999; Ganopolski and Calov, 2011; Abe-Ouchi et al., 2013; Tabor and Poulsen, 2016; Willeit et al., 2019).

For the future million years, however, not only natural processes are of relevance. It has been shown that anthropogenic $CO_2$ emissions can affect future climate variability at very long timescales (Archer et al., 1997; Lenton and Britton, 2006; Ridgwell and Hargreaves, 2007). Even already achieved cumulative anthropogenic emission of the order of 500 PgC, when assuming no negative carbon emission in the future, can postpone the onset of the next glaciation by at least 50 kyr while an emission of 5000 PgC could completely change climate variability over the next 0.5 Myr (Archer and Ganopolski, 2005; Ganopolski et al., 2016).

The modelling of the climate evolution at 1Myr timescales is not "accessible" for complex Earth system models and remains problematic even for most of the intermediate-complexity models due to prohibiting computational cost. Moreover, while Earth orbital parameters are accurately known for the next millions of years (Laskar et al., 2004), large uncertainties associated with the total amount of future fossil fuel combustion, requires a computationally efficient modelling approach which permits to perform numerous long-term simulations to asses impact of these uncertainties.

Most of simple (conceptual) models of glacial cycles (e.g. Imbrie and Imbrie 1980; Paillard 1998; Tziperman 2006; Crucifix 2012) simulate only the response of global ice volume to orbital forcing and thus are not suitable for an assessment of long-term consequences of anthropogenic $CO_2$ emission. Some simple models do describe also $CO_2$ concentration (for example: Saltzman, 1987; Saltzman and Verbitsky, 1993) but these models are designed only for description of the natural evolution of carbon cycle. Thus, to our knowledge, there is no computationally inexpensive modelling tool which can simulate the future evolution of the global ice volume and $CO_2$ concentration in response to anthropogenic perturbation.

To provide a set of possible scenarios of the future coupled climate – ice sheets – carbon cycle evolution we develop a reduced-complexity model based on paleoclimate data and results of physically-based Earth system models. This approach constitutes a significant further development of the method proposed in Archer and Ganopolski (2005) and Lord et al. (2019). In the former, Paillard's conceptual model of glacial cycles was combined with the results of future $CO_2$ simulations performed with a simple marine carbon cycle. In the latter, Palliard's model was combined with simulations of the future evolution of anthropogenic $CO_2$ anomaly modelled as in Lord et al. (2016) based on results of the Earth system model of intermediate complexity cGENIE (Lenton et al., 2006).

In this manuscript, a more advanced semi-empirical model is calibrated using paleoclimate records from the last 800 kyr and the results of the CLIMBER-2 EMIC (Petoukhov et al., 2000; Ganopolski et al., 2001), while the future evolution of anthropogenic $CO_2$ anomaly is modelled following Lord et al. (2019, 2016).

We organize the manuscript as follows. In Sec. 2 we introduce the model and datasets. In Sec. 3 we present the model calibration. In Sec. 4 we present the results of the simulations for the next 1 Myr, considering scenarios with and without anthropogenic influence. In Sec. 5 we analyse the sensitivity of the results to several aspects of the model design and parameter selection strategy. Finally, in Sec. 6 we summarize the results and present the conclusions.

## 2    Model and datasets

### 2.1    Modelling approach

We develop a process-based model for the coupled evolution of global ice volume (v), atmospheric $CO_2$ concentration ($CO_2$) and global mean surface air temperature (T). For the past, the model's only external forcing is orbital forcing. For the future, we account additionally for the impact of the cumulative anthropogenic $CO_2$ emissions on atmospheric $CO_2$ concentration. The model consists of a system of three coupled, non-linear differential equations, describing Earth system evolution at the timescales from thousands to millions of years. While the model is designed for simulations of future glacial cycles, global temperature is a useful diagnostic which can become necessary in other potential applications.

It has been shown (Crucifix, 2012) that very different conceptual models of glacial cycles, based on completely different assumptions, can simulate past glacial cycles with similar and sufficiently high skill. Thus, paleodata alone are not sufficient to select the "right" model. Our approach to the construction of such model is principally different from previous studies because the model is designed based on the results of physically based Earth system climate models. In particular, based on the CLIMBER-2 model simulations (1) we define orbital forcing as the maximum summer insolation at $65^{\circ}$N; (2) it is required that for any orbital forcing there is at least one stable equilibria and for a certain range of orbital forcings two equilibria (Calov

and Ganopolski, 2005); (3) the transition from interglacial to glacial state occurs when the insolation drops below a certain threshold value which linearly depends on the logarithm of $CO_2$ concentration (Ganopolski et al., 2016). We only selected those model realization which have this relationship consistent with the analysis presented in Ganopolski et al. (2016) (see discussion below).

For model calibration, we use a hybrid approach, based on a combination of paleodata and CLIMBER-2 model simulations. We use paleoclimate reconstructions of the late Quaternary (last 800 kyr; see below). This period was selected for two reasons: (1) so far this is the only period of time for which accurate reconstructions of atmospheric $CO_2$ concentration are available; (2) this period dominated by the long glacial cycles which are expected to continue in the future, at least, for some time in the absence of significant anthropogenic influence (see discussion below). However, during that period the $CO_2$ atmospheric concentration was most of time lower than preindustrial levels and, always below the present one. Therefore, those records alone are insufficient to derive a model suitable for a modelling future climate evolution under anthropogenic influence. Given the lack of accurate $CO_2$ reconstructions from periods with $CO_2$ atmospheric concentration higher than preindustrial level, we calibrate the model behaviour in such a greenhouse world using results of the physically-based CLIMBER-2 model. The selected model realizations are used to simulate the next million years under natural and several anthropogenic scenarios. Future $CO_2$ concentration scenarios were computed using results of the studies performed with the cGENIE EMIC. Since there are no available simulations of $CO_2$ concentration under the combined effect of anthropogenic $CO_2$ emissions and future ice sheet growth (growth that can only happen after the $CO_2$ anomalies decrease considerably) we assume that natural and anthropogenic $CO_2$ anomalies can be simply added up.

## 2.2 Equation for global ice volume

The first model equation describes the temporal evolution of global ice volume $v$ expressed in non-dimensional units (with a value of 1 corresponding to the reconstructed Last Glacial Maximum (LGM, 21 kyr before present) ice volume. Since it is set that at present $v=0$, this variable in fact represents the anomaly of global ice volume relative to the preindustrial state. As the principal natural forcing for global ice volume is summer NH insolation this equation is only valid for NH ice volume. To account for the contribution of the Southern Hemisphere (SH), we use the same approach as in Ganopolski and Calov (2011) and assume that the ice volume anomaly in the SH closely follows the NH anomaly and makes up a constant, relatively small faction of the global ice volume anomaly. This approach, obviously, is not applicable for a possible future Antarctic and Greenland ice sheets melting under high $CO_2$ concentrations. This is why we do not consider future sea level rise above the preindustrial level and it is required that $v \geq 0$ at any time (see below).

Although the constraint of $v \geq 0$ for the future is a strong one, we expect the approach not to be a problem even under the most pessimistic anthropogenic scenario we consider in this manuscript (cumulative emission of 3000 PgC). On one hand, a scenario

leading to the complete melting of the Greenland ice sheet would change the global ice volume by a small value compared to the glacial-interglacial variability and, thus, could be neglected. On the other hand, a complete Antarctic deglaciation would definitely affect the global climate carbon cycle and could not be ignored. However, according to Garbe et al. (2020), a

135 significant Antarctic mass loss occurs only if global temperature anomaly stays above 8°C for a very long time and, as a consequence, even under the 3000 PgC emission scenario the contribution of the Antarctic ice sheet to changes in global ice volume could be neglected, at least, to a first approximation (see below for the estimated temperature change under different anthropogenic scenarios).

The mass-conservation equation states that:

$$\frac{dv}{dt} = (Accumulation - Ablation) \tag{1}$$

"Accumulation" represents here the global ice accumulation (in mass per time units) and "Ablation" represents all mass losses

from ice sheets including surface and basal melt, and icebergs calving. We assume that the total accumulation is proportional to the total ice sheet area (Ganopolski et al., 2010). In turn, ice sheet area is closely related to the ice volume. It is often assumed that $A = v^\gamma$, where $\gamma$ is about 0.8. Here for simplicity we assume $\gamma=1$, and thus:

$$Accumulation = b\, v \tag{2}$$

where $b$ is the proportionality constant.

On the other hand, the ablation is controlled by several processes of which the energy balance at the surface of the ice sheets is the most important one. Total ablation depends on the size of the ice sheets (and in the NH, especially, on the position of

155 their southern margin), summer insolation (orbital forcing) and $CO_2$ atmospheric concentration ($CH_4$ and $N_2O$ also play a role but we assume their radiative forcing to be proportional to the radiative forcing of $CO_2$ as in Willeit et al., 2019). The effect of $CO_2$ on the mass balance of ice sheet is introduced via the logarithm of its concentration because radiative forcing of $CO_2$ is proportional to the logarithm of concentration. As the metric for orbital forcing we use the maximum summer insolation at 65ºN computed using Laskar et al. (2004). To reproduce a rapid deglaciation process we introduce an additional term

proportional to dv/dt. To ensure existence of at least one equilibrium solution (solution with dv/dt = 0) for any combination of orbital and $CO_2$ forcing (Calov and Ganopolski, 2005; Abe-Ouchi et al. 2013) a negative term proportional to the ice volume in power large than one is required. Here to this end we use the term $v^{3/2}$ since it gives a better model performance. Thus, the total ablation is represented as:

$$Ablation = b_{01}v + b_2v^{3/2} + b_3(f - \overline{f}) + b_4 log(CO_2) + b_5\frac{dv}{dt}M_v \qquad (3)$$

where $b_{01}, b_2, \ldots b_5$ are constants, $\overline{f}$ is the average orbital forcing, and $M_v$ is a memory term that reflects the history of the ice volume during the last $\tau$ yr, as long as dv/dt is negative. At any time $t$, $M_v(t)$ is defined as:

$$M_v(t) = \delta\frac{\int_{t-\tau}^{t}v(x)dx}{\tau} \qquad (4)$$

$$\delta = 1 \;\; if \;\; \frac{dv}{dt} < 0; \;\; and \; \delta = 0 \;\; otherwise \qquad (5)$$

Finally, the mass-balance Eq. (1) is re-written as:

$$\frac{dv}{dt} = \frac{b_1v - b_2v^{3/2} - b_3(f-\overline{f}) - b_4 log(CO_2)}{1-b_5M_v} + b_6 \qquad (6)$$

Where $b_{1=}b - b_{01}$ and $b_6$ is an additional model parameter.

## 2.3 Equation for atmospheric $CO_2$ concentration

It is generally recognized that $CO_2$ (together with other well-mixed greenhouse gases such as $CH_4$ and $N_2O$) represents an important amplifier of glacial cycles and that the anthropogenic $CO_2$ emission will affect the climate during long periods of time (Archer and Brovkin, 2008) and, in particular, the timing and magnitude of future glacial cycles (Archer and Ganopolski, 2005). Although the mechanisms of glacial-interglacial atmospheric $CO_2$ variability are still not fully understood, significant progress in modelling the global carbon cycle operation during glacial cycles has been achieved in recent times (Brovkin et al., 2012; Menviel et al., 2012; Willeit et al., 2019). Since $CO_2$ and ice volume are highly correlated (during the past 800 kyr the coefficient of correlation between atmospheric $CO_2$ concentration and ice volume is -0.71), it is not surprising that simple conceptual models of glacial cycles can describe the ice volume evolution without an explicit treatment of $CO_2$ (e.g. Paillard 1998; Crucifix 2013). However, this close relationship between $CO_2$ and global ice volume is not valid for the Anthropocene and, thus, the modelling of the future climate evolution requires an explicit treatment of atmospheric $CO_2$ concentration which is the second equation of our model. The response of the global carbon cycle to external perturbations involves a broad range of timescales: from very short (annual) to geological. For the purpose of this study we treat natural and anthropogenic anomalies of $CO_2$ separately but consider them additive. Namely, we assume that on the relevant timescales ($10^3$-$10^5$ yrs), the

natural component of $CO_2$ concentration is in equilibrium with external conditions and is expressed through a linear combination of global temperature and global ice volume.

The most direct effect of temperature on $CO_2$ is through changes in the $CO_2$ solubility in ocean water (Zeebe and Wolf-Gladrow, 2001). Temperature, directly and through sea ice, also affects ocean circulation (Watson et al., 2015), ventilation

rate of the deep water (Kobayashi et al., 2015), changes in relative volume of different water masses (Brovkin et al., 2007) and metabolic rates of living organisms (Eppley, 1972; Laws et al., 2000).

The direct effect of ice volume on $CO_2$ changes is less straightforward (note that the strong effect of ice sheets on atmospheric temperature is already accounted for in the temperature term). An increase in ice volume leads to a decrease of the ocean

volume and, as a consequence, increased ocean salinity. As $CO_2$ solubility in the ocean decreases with salinity this would provoke to an increase of atmospheric $CO_2$ concentration, counteracting the temperature effect (Zeebe and Wolf-Gladrow, 2001). Higher global ice volume would also mean a smaller area of the globe is covered by forests and, therefore, a diminished carbon storage in the terrestrial reservoir, driving an increase in atmospheric $CO_2$ (Prentice et al., 2011). Increased glacial supply of iron-rich dust may help suppress the iron limitation in some areas (in particular in the south Atlantic Ocean) leading

to increased biological productivity and thus lower atmospheric $CO_2$ levels (Martin, 1990; Watson et al., 2000).

In addition, based on results of Ganopolski and Brovkin (2017) we parameterized $CO_2$ "overshoots" during glacial terminations by an additional term proportional to $dv/dt$. Note that this additional term is only applied when $dv/dt<0$. The justification for such parameterization is the results seen in many models (Gottschalk et al., 2019) that a shutdown of the Atlantic Meridional

Overturning Circulation (AMOC) causes an atmospheric $CO_2$ rise of about 10-20 ppm. A shutdown of the AMOC is caused by a large meltwater flux which is, in turn, controlled by changes in ice volume (i.e. $dv/dt$).

To describe the anthropogenic component of $CO_2$, we assume that anthropogenic $CO_2$ emission is a relatively short (order of $10^2$ yrs) pulse followed by zero emissions. In this case, the temporal dynamics of the anthropogenic component of $CO_2$ anomaly depends only on the cumulative carbon emission. To describe the long anthropogenic tail of $CO_2$ ($Anth_{CO2}$ term in the Eq. (7)

below) we use the analytical parameterization defined in Lord et al. (2016) based on the results of experiments with the Earth system model of intermediate complexity cGENIE (Lenton et al., 2006). In addition, we assume that natural and anthropogenic $CO_2$ anomalies can be simply summed up and that at the preindustrial time the global carbon cycle was in equilibrium. This is, obviously, a very strong assumption since even a rather small imbalance in the global carbon cycle which is impossible to

detect at the millennial timescales can result in a very large "drift" of the Earth system from its preindustrial state at the million years timescale. However, due to the absence of any practical alternative, we proceeded with this assumption.

After these considerations, the equation that governs the $CO_2$ evolution has the following shape:

$$CO_2 = c_1T + c_2v + c_3 min\left(\frac{dv}{dt}, 0\right) + c_4 + Anth_{CO_2} \tag{7}$$

Where $c_1 \ldots c_4$ are adjustable model parameters. The first three terms on the right-hand-side of equation 7 represent the already discussed effects of global temperature (T), ice volume and AMOC weakening or shutdown. The fourth term ($c_4$) is simply a constant and the last term represents the effect of anthropogenic $CO_2$ emissions. Refer to Sec. 4.1 for a detailed explanation
on the assumptions and shape of the function $Anth_{CO2}$(t).

## 2.4     Equation for global mean surface temperature

The last equation in our model describes global mean surface air temperature anomaly (relative to preindustrial values) as a linear combination of two terms: first, the direct effect of global ice volume (the larger the ice volume, the higher the planetary albedo and the lower the global temperature) and, second, a term representing the radiative forcing of $CO_2$, which is
proportional to the logarithm of $CO_2$ concentration:

$$T = d_1v + d_2 log\left(\frac{CO_2}{278}\right) \tag{8}$$

where $d_1$ and $d_2$ are adjustable model parameters.

## 245   2.5     Model constraints

The model of past and future glacial cycles is represented by the system of three equations (6), (7), (8) which contains three variables ($v$, $CO_2$ and $T$), orbital forcing $f$(t), anthropogenic $CO_2$ anomaly $Anth_{CO2}$(t) and twelve parameters ($b_i$ i=1:6; $c_j$ j=1:4, $d_k$ k=1:2). The orbital forcing $f$(t) is determined by astronomical parameters (eccentricity, precession and obliquity) which are accurately known for the entire period of interest (Laskar et al., 2004). The function $Anth_{CO2}$(t) is assumed to be null until t =
0 kyr, afterwards the function depends only on time and cumulative $CO_2$ emission as detailed in Sec. 4.1.

Model parameters are calibrated to yield solutions that are in a good agreement both with the paleoclimatic information over the last 800 kyr (see Sec. 2.6 for details on the paleoclimate reconstructions used) and CLIMBER-2 results.

We impose on the model a series of constrains based on paleorecords from the last 800 kyr: 1. Reproduction of present-day interglacial state; 2. Reproduction of LGM conditions; 3. Mid-Brühnes transition (MBT); 4. Compliance with $CO_2$ minimum level.

The first constraint, based on the observational fact that the present state is an interglacial one, we require that:

if $v=0$ and T = 0, then $CO_2$ = 278 ppm.                                                                (9)

This condition fixes the value for $c_4$ at 278.

As the confidence on empirical data for glacial conditions is higher at LGM than at any other glacial episode, the accurate
reproduction of LGM conditions is important. At LGM, the empirical ice volume in non-dimensional units was 1, $CO_2$
concentration 194 ppm and global cooling around 5°C (Schneider et al., 2006; Annan et al., 2013; Tierney et al., 2020). We
then require from the model that:

if $v = 1\ \&\ CO_2 = 194\ ppm \rightarrow T = -5°C$                                                        (10)

It is known that glacial cycles prior to the mid-Brünhes Transition (MBT) about 400 kyr ago differ in some respects from those
after MBT. In particular, pre-MBT interglacials were characterised by lower $CO_2$ and higher benthic d¹⁸O values than the post-
MBT episodes. The later implies a large interglacial, probably due to remaining continental ice sheets in the NH. The
mechanisms accounting for these differences are still not understood (Tzedakis et al., 2009; Berger et al., 2016) but they are
unlikely related to differences in orbital forcing before and after MBT. This is why it is not expected that a simple model forced
by orbital variations alone can simulate such behaviour. Thus, we prescribe that before the MBT the minimum ice volume
must be 0.05 in normalized units:

$v(t) = max(v(t), 0)$  for all t                                                                           (11)
$v(t) = max(v(t), 0.05)$ if t < -400kyr                                                                   (12)

The last condition based on empirical data prevents $CO_2$ to drop to levels significantly lower than the minimum atmospheric
$CO_2$ concentration registered in the last 800 kyr by paleorecords (172 ppm) at any given moment in time:

$CO_2(t) = max(CO_2(t), 150)$                                                                              (13)

We also impose a condition related to the current estimates for the equilibrium climatic sensitivity (ECS), i.e. global
temperature response to a doubling in atmospheric $CO_2$ concentration from preindustrial levels. We select equilibrium climate
sensitivity (ECS) equal to 3.9°C, which coincides with the multi-model mean in the Coupled Model Intercomparison Project
6 (CMIP6; Zelinka et al., 2020):

if $CO_2 = 2x278ppm\ \&\ v = 0 \rightarrow T = 3.9°C.$                                                      (14)

Equations (10) and (14) determine the values for the coefficients $d_1$ and $d_2$ at -3 and 5.56, respectively.


Recent studies (e.g. Nijsee et al., 2020) suggest that some of CMIP6 models have unrealistically high ECS and the IPCC AR6 report proposed a best estimate for ECS of 3°C. However, on the long time scales considered in this paper, the classical ECS which accounts only for fast climate feedback, but not for the feedbacks related to vegetation, ice sheets and atmospheric composition (apart from $CO_2$) likely underestimates the Earth system response to radiative perturbation. Nonetheless, to assess

uncertainties in future climate simulations associated with uncertainties in ECS, we performed an additional set of simulation with ECS=3°C (see Sec. 5).

In addition, we account for several modelling studies (Berger and Loutre, 2002; Cochelin et al., 2006; Ganopolski et al., 2016) indicating that the conditions for the new glacial inception will not be met in the near future even in the absence of

anthropogenic influence on climate. To assure that our model satisfies this requirement, we require for all valid model versions the average global ice volume not to exceed 0.025 (in normalized units) in the time period between 0 and 20 kyr:

$$\frac{\sum_{t=0}^{20\,kyr} v(t)}{21} < 0.025 \qquad\qquad (15)$$

For the Base experimental setting (other settings will be discussed in Section 5), the model is defined by equations 6-8 together with the conditions expressed in equations 9-15. A table of acronyms (Table 1) is provided as a quick reference.

## 3    Model calibration

In this section we describe the choice of model parameters using paleoclimate data and CLIMBER-2 modelling results. First,

we select a range of model parameters which enable us to simulate the past evolution of global ice volume, $CO_2$ and global temperature with a reasonable accuracy (see Sec. 3.2). We call this set of model realisation *Paleovalid*. We then apply an additional constraint based on the critical insolation – $CO_2$ relationship for glacial inceptions, which is of fundamental importance for the prediction of future glacial cycles in the response to anthropogenic $CO_2$ emissions. Since we find that paleodata do not provide additional constraints on this relationship (see Sec. 3.3), we consider the estimation of the critical

insolation – $CO_2$ curve derived from CLIMBER-2 in Ganopolski et al. (2016) and take forward only those solutions in line with it. All model realisations which satisfy this criterion are named *Accepted* and we use them for future scenarios simulations.

**Table 1: Table of acronyms**

|  | Acronym | Units |
|---|---|---|
| Global ice volume | v | Non-dimensional (varies between 0 and 1) |
| Atmospheric $CO_2$ concentration | $CO_2$ | ppm |
| Global surface temperature anomaly (with respect to preindustrial levels) | T | °C |
| Orbital forcing: maximum summer insolation at 65°N | f | $Wm^{-2}$ |
| Time | t | kyr |
| Memory term (see Equation (4)) | $M_v$ | Non-dimensional |
| Anthropogenic $CO_2$ concentration | $Anth_{CO2}$ | ppm |
| Last Glacial Maximum | LGM | - |
| Mid-Brühnes transition | MBT | - |
| Cumulative anthropogenic $CO_2$ emissions | E | ppm |
| Equilibrium climate sensitivity (see Equation (14)) | ECS | °C |
| Critical insolation – $CO_2$ relationship parameters (see Equation (18)) | K, R | $W\ m^{-2}$ |
| Set of model parameters | P: $b_1$,…,$b_6$, $c_1$,…, $c_4$, $d_1$, $d_2$ | see Table 2 |
| Parameters for evolution of $Anth_{CO2}$, derived in Lord et al. (2016) | $\alpha_i, \beta_{ij}, \gamma_i, \delta_{ij}$ i=1,…,5; j=1,…,3 | see Lord et al. (2016) |
| Pearson correlation between modelled and paleo variable x (v, $CO_2$ or T) considering the time period $[t_i,t_f]$ | $C_x(P,t_i,t_f)$ | - |
| Root Mean Squared Error | RMSE | - |
| Set of solutions consistent with paleodata | *Paleovalid* | - |
| Subset of *Paleovalid* consistent with CLIMBER-2 results | *Accepted* | - |

### 3.1 Empirical datasets

Paleo reconstructions covering the period [-800 kyr, 0 kyr] are used as part of the learning and/or validation sets for the model.
Reconstructed global ice volume is derived from sea level stack (Spratt and Lisiecki, 2016). Past $CO_2$ atmospheric concentration levels are derived from ice cores records (Lüthi et al., 2008). For global mean surface temperature anomalies (with respect to preindustrial conditions) we use two reconstructions: 1. Friedrich et al. (2016), which covers the period [-784 kyr, -10 kyr] and 2. Snyder (2016), covering [-800 kyr, -1 kyr]. All datasets are transformed into time-series with a 1 kyr time-step.


Obviously, the paleoclimate reconstructions are not perfect, and have associated uncertainties. While information for global ice volume and $CO_2$ atmospheric concentration are based directly on paleoclimate records and are considered quite accurate, global temperature reconstructions are based on a limited number of local temperature records, mostly from ocean sites, and have large uncertainties (as easily noticeable by the two paleorecords in Fig 1c).

## 3.2      Calibration using paleoclimate reconstructions

For each set of parameters P, we calculate $v$, $CO_2$ and T using the system of equations 6-8 and denote them: $v_{model(P)}, CO_{2model(P)}, dT_{model(P)}$.

We approach the task of the selection of set of parameters P as a non-linear optimisation problem with constraints. We wish to find P to maximize the optimization target function $C_v$:

$$C_v(P, t_i, t_f) = corr(v_{model(P)}(t_i, t_f), v_{Paleo}(t_i, t_f)) \tag{16}$$

where: $v_{model(P)}(t_i, t_f)$ denotes the modelled ice volume time-series in the period $[t_i, t_f]$; $v_{Paleo}(t_i, t_f)$ denotes the paleo ice volume time-series in the period $[t_i, t_f]$; $corr(x,y)$ denotes the linear Pearson correlation between x and y. The time interval for the optimisation is set to $[t_i, t_f] = [-800\ kyr, 0\ kyr]$. See Appendix A for a discussion of the dependence of model performance on the choice of this time interval. To select parameters that will optimise correlation at the same time as providing magnitudes in accordance to empirical estimations, an equality constraint is enforced: the maximum ice volume must be equal

to 1 within a tolerance of 0.15 (in non-dimensional units). Eq. (14) is enforced as inequality constraint.

The time-step for the calculation of the time-series is 1 kyr. We use the interior-point optimisation algorithm under the Matlab environment. Approximately 1000 optimisation routines were performed, each starting from a different combination of the set of nine adjustable parameters (optimisation seeds).


Finally, the ensemble of valid solutions, *Paleovalid*, is defined as:

$$Paleovalid = \{P/C_v(P, -800\ \text{kyr}, 0\ \text{kyr}) \geq 0.7\} \tag{17}$$

The correlation level of 0.7, although arbitrary, guarantees a good fit to the paleo climatic ice volume record. For the selection of solutions, no conditions are imposed on the goodness of fit between modelled and paleo $CO_2$ or temperature.

Naturally, there are many possible choices for model calibration procedures. In particular, it is possible to select a different optimisation target function. We selected to optimise the correlation between modelled and paleo ice volume because our main

objective is to simulate future glacials and, thus, the reproduction of these cycles is extremely relevant. Livadiotis and McComas (2013) present the maximization of the correlation fitting method. Those authors show that the method is mathematically well defined under certain conditions and that it should be preferred over the classical least squares fitting in situations in which the data sets exhibit variations that need to be described, such as is the case for glacial cycles. The sensitivity of our results to a different optimisation target is evaluated in Sec. 5. While we opt for a frequentist perspective (estimates of

unknown parameters are obtained without assigning them probabilities) parameter estimation can also be approached through a Bayesian point of view (e.g. Crucifix and Rougier, 2009).

Figure 1 displays the ensemble probabilistic distribution of solutions in *Paleovalid* for *v*, $CO_2$ and T as well as the corresponding paleoclimatic data for the period [-800 kyr, 0 kyr]. The ranges of the different model parameters across the

*Paleovalid* set are displayed in Table 2.

By definition, all the solutions derived from the *Paleovalid* ensemble closely follow the paleo ice volume curve (Fig. 1a), with the ensemble-mean correlation between modelled and paleo ice volume in [-800 kyr, 0 kyr] equal to 0.76. While most of the solutions derived from *Paleovalid* succeed in capturing the timing and magnitude of the major glaciations, there is a tendency

for them to overestimate the ice volume in MIS 18 and MIS 14 (-560 kyr and -520 kyr, respectively). As model parameters were chosen to maximise ice volume correlation only, it is expected that the performance in terms of $CO_2$ and global temperature is inferior than for global ice volume. The ensemble-mean correlation between modelled and paleo $CO_2$ is 0.5 and approximately 50% of the solutions display an amplitude range significantly larger than the observed one, reaching the imposed lower limit of 150 ppm (Fig. 1b). Note however the better magnitude agreement when comparing simulated $CO_2$ with

equivalent $CO_2$ concentration (as computed in Ganopolski et al., 2010) and which accounts for the additional radiative effect of $N_2O$ and $CH_4$. The imperfect $CO_2$ simulations represents however no major pitfall for a skill full ice volume reproduction, as shown by the 0.76 ensemble-mean correlation with paleodata. Interestingly, when using accurately known $CO_2$ time-series from ice cores, this ensemble-mean correlation for global ice volume only is slightly increased to 0.85. For global mean surface temperature (Fig. 1c), the ensemble-mean correlation between modelled and paleo record is 0.56 and the solutions amplitude

range within the paleo estimation limits.

**Table 2: Parameter ranges across solutions in the *Paleovalid* and *Accepted* sets as well as for the Best Solution.**

| Model Parameter | Units | *Paleovalid* set | *Accepted* set | Best Solution |
|---|---|---|---|---|
| $b_1$ | $s^{-1}$ | 0.075 – 0.27 | 0.12 – 0.26 | 0.22 |
| $b_2$ | $m^{-1/2}\ s^{-1}$ | 0.49 – 0.15 | 0.33 – 0.15 | 0.29 |
| $b_3$ | $m\ s^{-1}\ W^{-1}$ | $9 \times 10^{-4} - 3 \times 10^{-4}$ | $9 \times 10^{-4} - 5 \times 10^{-4}$ | $8 \times 10^{-4}$ |
| $b_4$ | $m^3\ s^{-1}$ | 0.62 – 0.02 | 0.12 – 0.02 | 0.095 |
| $b_5$ | Non-dimensional | 1 – 0.04 | 0.76 – 0.13 | 0.18 |
| $b_6$ | $m^3\ s^{-1}$ | 0.1 – 3.49 | 0.1 – 0.66 | 0.53 |
| $c_1$ | $ppm\ C^{-1}$ | 10.6 – 18.84 | 12.37 – 18.84 | 17.28 |
| $c_2$ | $ppm\ m^{-3}$ | -35.1 – -20.0 | -35.1 – -22.87 | -31.95 |
| $c_3$ | $ppm\ m^{-3}\ s$ | -120.1 – -119.9 | -120.03 – -119.9 | -120.0 |
| $c_4$ (derived from constraints) | ppm | 278 | 278 | 278 |
| $d_1$ (derived from constraints) | $C\ m^{-3}$ | -3 | -3 | -3 |
| $d_2$ (derived from constraints) | C | 5.56 | 5.56 | 5.56 |

**Table 3: Summary of *Paleovalid* and *Accepted* ensembles characteristics.**

| | *Paleovalid* | *Accepted* |
|---|---|---|
| **Correlation between modelled and paleo ice volume** | $\geq 0.7$ | $\geq 0.7$ |
| **K (W m$^{-2}$)** | Unrestricted | $\geq -150$ |
| **Number of elements** | 353 | 29 |

While the model demonstrates a good performance when tested against the same data that was used for its training, its ability to produce predictions (situation in which previously unseen data is used as model input) can only be evaluated by the use of independent training and validation samples. A robust estimation of the model's predictive skill, using disjoint training and validation sets, is found in Appendix A. In general, we conclude that the model has a satisfactory ability also when used in predictive mode and, thus, we confidently venture to utilize it as a tool for the assessment of possible future scenarios.


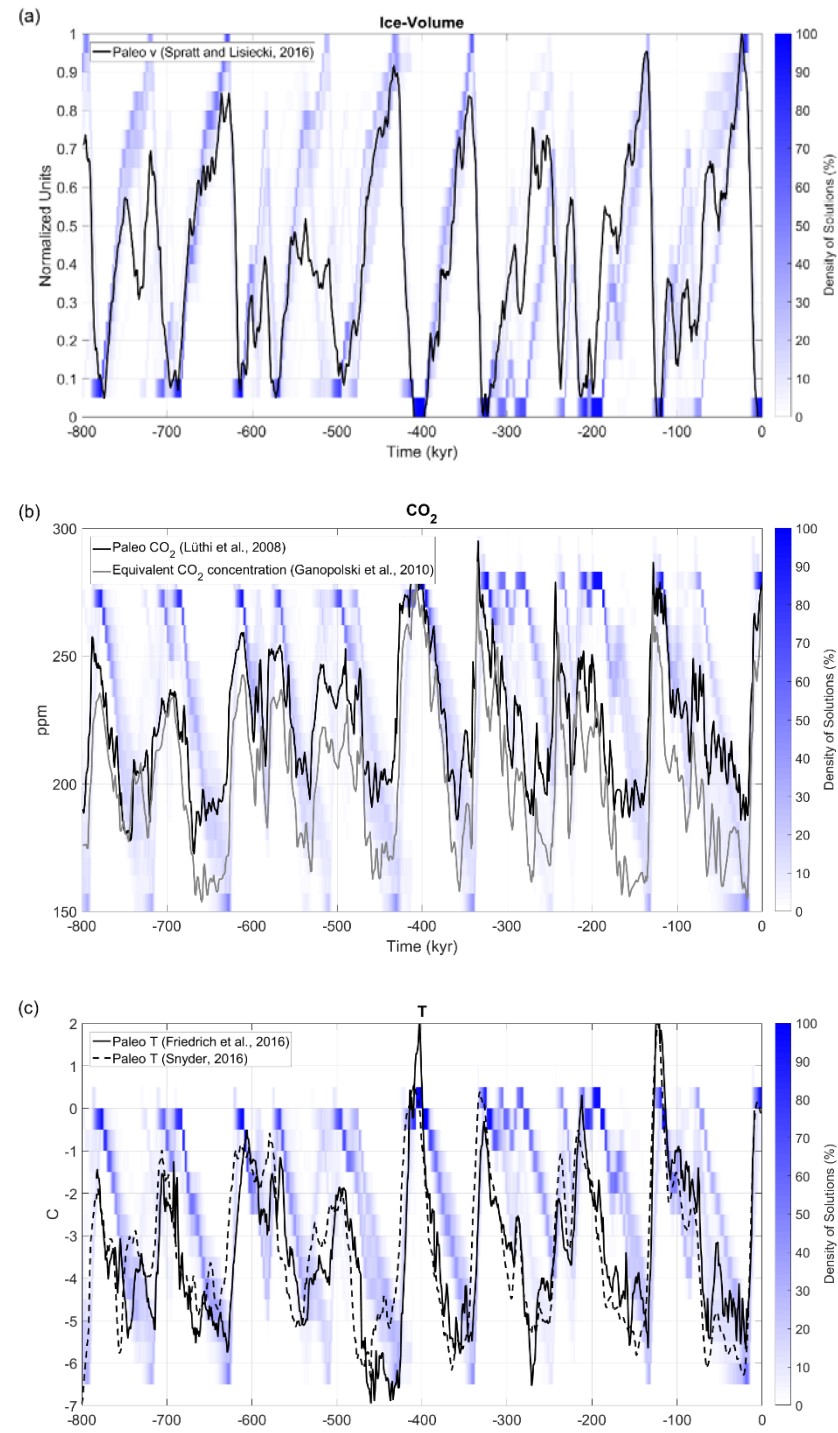

**Figure 1:** *Paleovalid* solutions and paleoclimatic records in the period [-800 kyr, 0 kyr]: a. Global landmass ice volume (normalized to unity); reconstruction (black line) from Spratt and Lisiecki (2016). b. Atmospheric CO₂ concentration (ppm); paleorecord from


Lüthi et al. (2016) and equivalent CO₂ concentration from Ganopolski et al. (2010) (grey line). c. Global mean surface temperature anomaly (C); paleorecords from Friedrich et al. (2016) (solid black) and Snyder (2016) (dashed). Blue shaded areas represent the density of *Paleovalid* solutions per unit of phase space (Units are: 1kyr for time, 0.05 for ice, 7ppm for CO₂ and 0.5°C for temperature).

### 3.3    Glacial inception: Critical insolation – CO₂ relationship

Using an ensemble of CLIMBER-2 model realizations consistent with paleoclimatic constraints, Ganopolski et al. (2016) found that the critical threshold for summer insolation at 65° N to trigger glacial inception is described by the following curve:


$$f_{critical} = K log(CO_2/280) + R \tag{18}$$

where $K = -77$ W m$^{-2}$ and $R = 466$ W m$^{-2}$.

The parameter K is a measure of the sensitivity of the critical orbital forcing to atmospheric $CO_2$ concentration. The higher K is, the more important $CO_2$ is, i.e. for the same $CO_2$ emission scenario, the effect of anthropogenic perturbation on glacial cycles will last longer.

Following Ganopolski et al. (2016) approach, we calculate the corresponding coefficient K for each of the ensemble members

in the *Paleovalid* set (Fig. 2). For the estimation of K we refer to Eq. (6), assuming equilibrium and interglacial conditions (i.e. $dv/dt=0$, $v=0$, $M_v=0$) it is derived that the critical insolation threshold:

$$f_{cr} = -\frac{b_4}{b_3} log(CO_2) - \frac{b_6}{b_3} + \bar{f} \tag{19}$$

It follows then that K is estimated as $-b_4/b_3$.

Our results indicate that with the model derived in this study the possible values of the coefficient K range between -1279 and -31 W m$^{-2}$, with a median of -393 W m$^{-2}$. This highlights that, even though all the solutions derived from the *Paleovalid* set have a high level of agreement with the paleoclimatic records along last 800 kyr, the relationship between $f_{critial}$ and $log(CO_2)$ is not constrained by paleoclimate data, i.e model realisations with completely different values of K can produce similarly good agreement with the paleoclimate reconstructions. However, for the future simulations different K values lead to very

different impact of anthropogenic $CO_2$ emissions on the Earth system evolution. This is why for performing future simulations we select only those *Paleovalid* solutions which are consistent with CLIMBER-2 simulations. However, since Ganopolski et al. (2016) presents only single values for K parameter and did not perform any uncertainty analysis of K, we applied a rather

"soft" constraint on its value, namely, it should satisfy the condition K ≥ -150 W m⁻². Even such "soft" constraints leads to a dramatic reduction of the remaining model realizations, which we call *Accepted*, and which consists only of 29 members. A summary of the characteristics of the *Paleovalid* and *Accepted* ensembles is found in Table 3. The probabilistic distribution of solutions for v, $CO_2$ and T derived from *Accepted* is displayed in Fig. 4 and the ranges of the different model parameters across *Accepted* is found in Table 2.

In the next section where we present results of future simulations we use only *Accepted* model versions. However, to demonstrate the importance of additional constraint on the value of K, we perform future simulations also with all *Paleovalid* versions. Fig. 3 illustrates these results through the timing of the first (after the present) full glacial conditions as a function of K. Not surprisingly, for the natural scenario, there is no clear dependence on K and the first full glacial conditions are predicted 100 or 200 kyr from present. The situation is very different for the scenarios with anthropogenic emissions. For example, for low (500 PgC) emissions, the model versions with K value satisfying the criteria for the *Accepted* model version, predict the next full glacial conditions around 200 kyr from present. However, most of the model versions with higher K predict the next full glacial conditions 700 kyr and even 900 kyr from present. The situation is similar for other anthropogenic scenarios. This analysis clearly shows the importance of applying a tight constraint on K value to produce realistic future scenarios.

In addition, we define the Best Solution $P_{Best}$ as:

$$P_{Best} = Argmax(C_v(P, -800\ kyr, 0 kyr), P \in Accepted) \tag{20}$$

$P_{Best}$ values are shown in Table 2. The time-series $v_{model(P_{Best})}, CO_{2model(P_{Best})}, T_{model(P_{Best})}$ are shown in Fig. 4. The model version named Best Solution is rather successful in reproducing the observed climatic variability in the three considered variables for the whole [-800 kyr, 0 kyr] period. The model skill in reproducing both the temporal variability and amplitudes of the time-series fluctuations is remarkable. In particular, the performance for ice volume is excellent with a correlation between model and paleodata of 0.86 (Fig. 4a). The model is able to correctly capture the timing of all the glaciations, being the LGM the highest volume event. The ice volume model and paleorecord discrepancy is largest for MIS 18 ([-750 kyr, -710 kyr]). The frequency spectra of $v_{model(P_{Best})}$ and the empirical estimation are also in good agreement (Fig. 5): there is a clear dominant peak at 100 kyr, with additional power at 41 and 23 kyr. For atmospheric $CO_2$ concentration, the correlation between modelled and recorded series is 0.62 (Fig. 4b). While the model succeeds in reproducing glacial-interglacial variability, it overestimates its magnitude: around 130 ppm instead of 80 ppm. It is, in particular, evident that during interglacials the $CO_2$ restores to its equilibrium value of 278 ppm, which according to the paleorecords is not accurate before the MBT. Finally, the correlation between modelled and paleo global mean surface temperature anomalies is 0.54 (Fig. 4c). The model slightly underestimates the variability of this variable, ranging between 0°C and -5°C, while the observational information suggests a

larger range between 2°C and -7°C. The high positive temperature anomalies during some previous interglacials, however, are questionable.

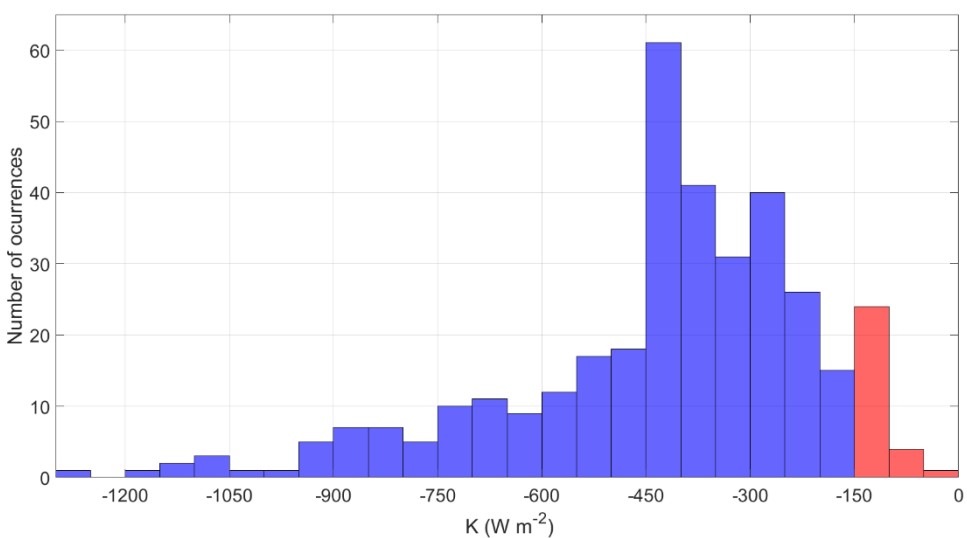

**Figure 2: Histogram for K (see Eq. (17)) across the different members of the *Paleovalid* ensemble of solutions. Red bars correspond to the *Accepted* solutions.**

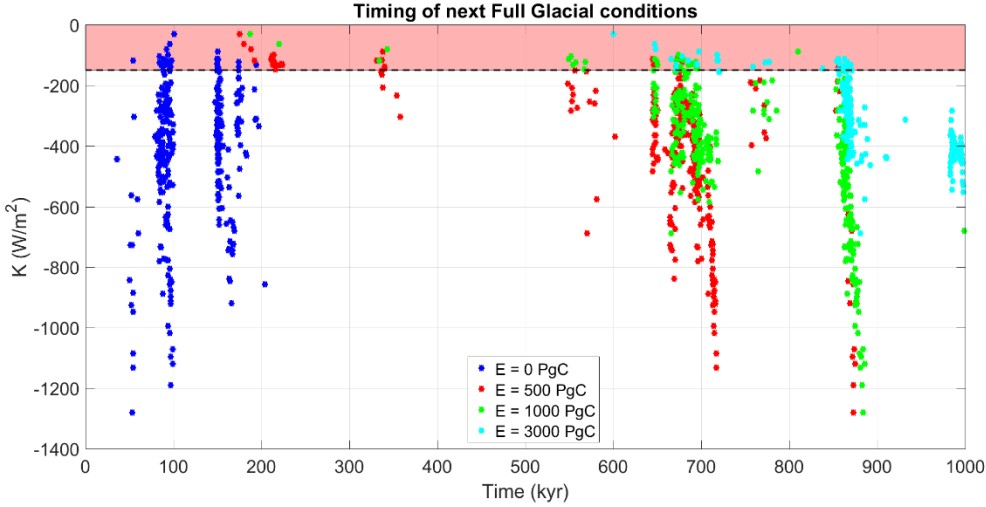

**Figure 3: Dependence of timing of the next full glacial conditions (v=0.5) on parameter K, for all solutions in *Paleovalid* and under**

**the different emission scenarios considered. The red shaded area indicates the solutions in the *Accepted* set.**

# 4 Simulations for the next 1 Myr

In this section we present results of simulations of the evolution of the Earth system in the next 1 Myr under natural and several anthropogenic emissions scenarios using the *Accepted* models ensemble.

## 4.1 CO₂ emission scenarios

Future scenarios are generated considering different temporal evolution paths for the excess atmospheric $CO_2$ concentration, which we assume depends only on the cumulative amount of fossil-fuel combustion, $Anth_{CO2}(t)$ (see Eq. (7)). For the natural scenario, $Anth_{CO2}(t)$ is set to zero at all times and, as a consequence, the climate system follows its natural evolution forced only by changes in orbital forcing.

For the fossil-fuel emission scenarios, we consider instantaneous releases of $CO_2$ at t = 0, of different magnitudes. If the duration of the emission pulse release is rather short (centuries) and is followed by zero $CO_2$ emission, the future long-term evolution of $CO_2$ depends primarily on the cumulative emission magnitude while the rate of release is only of secondary relevance (Eby et al., 2009). For the period 1750-2017, estimations place the fossil-fuel cumulative emissions in carbon equivalent at 660 +/- 95 PgC (Le Quéré et al., 2018). Projections for future cumulative emissions range between ~700 PgC and ~3000 PgC, being the former an estimate of fossil-fuel reserves exploitable with today's technical and economic constraints and the latter an estimation considering reserves that might become exploitable in the future (McGlade and Ekins, 2015). Taking into account the already achieved and future cumulative emissions estimations, we generate three different scenarios: low, intermediate and high emissions, corresponding to instantaneous pulse releases of magnitude 500 PgC, 1000 PgC or 3000 PgC, respectively. In each scenario, after the pulse release, $Anth_{CO2}(t)$ follows a multi-exponential decay function as proposed by Lord et al. (2016):

$$Anth_{CO_2}(t) = 0.469 * E \sum_{i=1}^{5} A_i e^{-t/\tau_i} \tag{21}$$

$$A_i(E) = \alpha_i + \beta_{i1}E + \beta_{i2}E^2 + \beta_{i3}E^3 \tag{22}$$

$$\tau_i(E) = \gamma_i + \delta_{i1}E + \delta_{i2}E^2 + 5\delta_{i3}E^3 \tag{23}$$

where E (PgC) represents the cumulative fossil-fuel emission magnitude, the constant 0.469 transforms PgC to ppm of $CO_2$ and $Anth_{CO2}(t)$ is expressed in units of ppm.

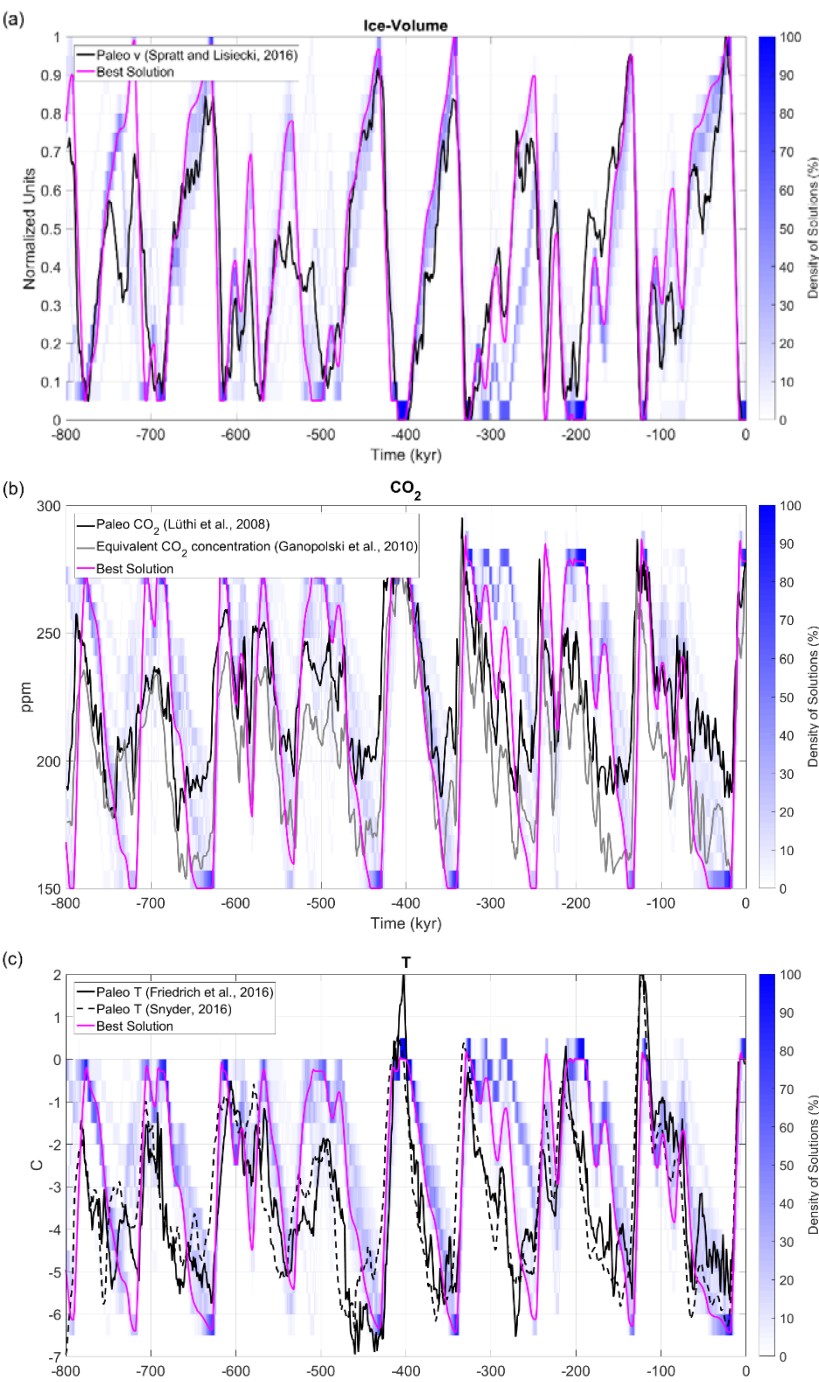

**Figure 4: Same as Fig.1 but for solutions in the *Accepted* ensemble. In addition, magenta lines show the evolution of the Best Solution.**

Parameter values ($\alpha_i, \beta_{ij}, \gamma_i, \delta_{ij}$ i=1,…,5; j=1,…,3) are derived in Lord et al. (2016) in order to produce a good fit to a series of 1 Myr pulse-type of experiments performed with the *c*GENIE EMIC (Lenton et al., 2006). Lord et al. (2016) consider total anthropogenic $CO_2$ emissions ranging between 0 PgC to 20.000 PgC, where the latter value is justified assuming future techno-economic advances could make additional non-conventional resources such as methane clathrates available for extraction. The use of results from long-term simulations with the cGENIE model represents a substantial improvement compared to Archer and Ganopolski (2005) who used $CO_2$ scenarios obtained with a simple marine carbon cycle model which did not explicitly account for a climate – weathering feedback.

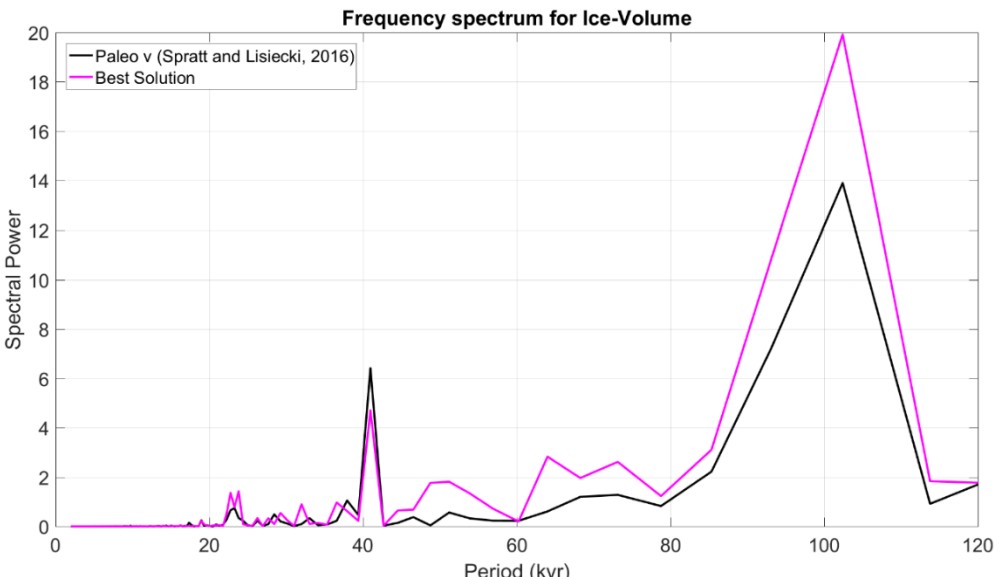

**Figure 5: Frequency spectrum for simulated (Best Solution, magenta) and reconstructed ice volume (Spratt and Lisiecki, 2016; black).**

The function $\text{Anth}_{CO2}(t)$ from present until 1 million years into the future, under the low, intermediate and high emissions scenarios is shown in Fig. 6. 100 kyr after the (low, intermediate or high) emissions pulse release ~ 5% of the initial $CO_2$ anomaly is still present in the atmosphere. For the low emissions scenario, only 500 kyr after the pulse emission do the remaining anthropogenic $CO_2$ concentration anomalies drop below 1% of the original perturbation, marking the return of the system to its natural state. For the medium and high emission scenarios, the return to natural conditions takes even longer.

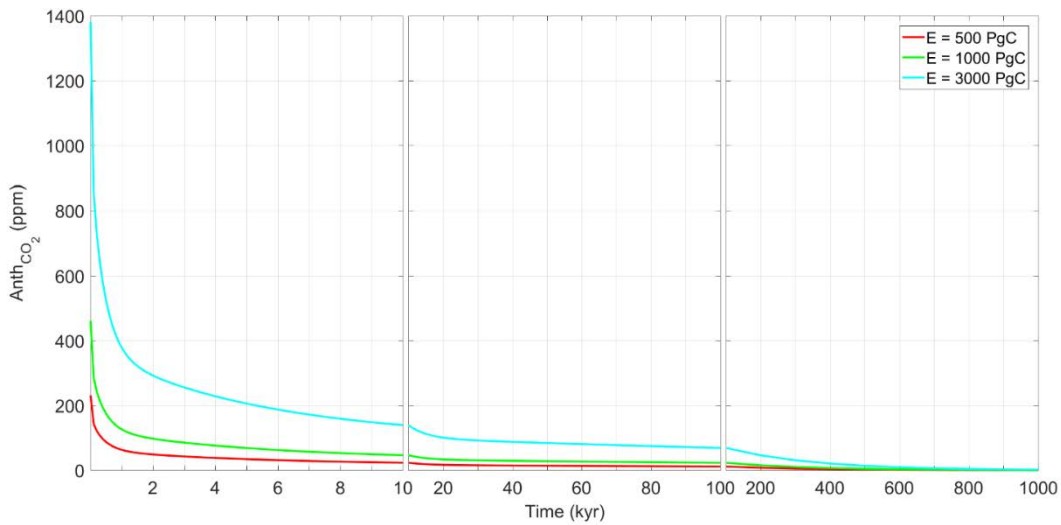

**Figure 6: Anth$_{CO2}$(t) expressed in ppm, following Lord et al. (2016) for the three different emission scenarios considered.**

 **4.2    Possible future natural Earth system trajectories**

Figure 7a displays future global ice volume evolution for the Best Solution and the probabilities of the trajectories of individual model realizations. It shows both periods of high agreement and periods of high discrepancy between solutions, highlighting the co-existence of more certain (deterministic) and less certain aspects of the Earth system response to orbital forcing. Most of the solutions agree that the planet will remain in a long interglacial state for the next 50 kyr, during which no large ice-

growth is expected (however, there is one solution in which glacial inception starts at 20 kyr and full glacial conditions are reached by 36 kyr). This behaviour is predefined by one of the model constrains, namely, that v should be equal zero at present and by a weak orbital forcing during the next 20 kyr. With high level of confidence, a second long interglacial is expected to occur between 450 and 500 kyr, which is another prolonged period of weak orbital forcing related to 400 kyr periodicity of eccentricity. Also with high certainty, short (~10 kyr) interglacials are expected to occur in 110, 230, 250, 270, 370, 620, 820

and 900 kyr from present.

In the conceptual model we define that full glacial conditions occur when the ice volume (v) reaches 0.5 in normalized units. If full glacial conditions are reached at a time T, then the corresponding glacial inception is defined as the first time before T in which v>0:

$$Glacial\ inception = \sup\{t\ /\ \ t < T\ \&\ v(t) > 0\,\} \tag{24}$$

Following those definitions, regarding the timing of the next full glacial conditions the solutions tend to cluster into two different possibilities: predicted to occur in ~90 kyr or in ~150 kyr (see also Fig. 3). For those cases, the glacial inception occurs at ~50 kyr or ~110 kyr, respectively. There is also some level of certainty in the timing of occurrence of major glaciations (defined here as periods with ice volume higher than 0.8 in normalized units): in ~100 kyr, ~210 kyr, ~410 kyr, ~450 kyr, ~680 kyr, ~790 kyr, ~870 kyr and 950 kyr from present. The possible future scenario indicates that only two of these major glaciations (the ones in ~210 kyr and ~870 kyr after present) have high probability of reaching LGM magnitudes (~1 in normalized units).

If we consider the possible future scenario produced by the trajectory of the Best Solution (Fig. 8) as the most likely path, under natural circumstances: 1. The next glacial inception is expected to occur ~ 50 kyr after present (with full glacial conditions reached ~90 kyr after present); 2. In the next 1 Myr two long interglacials should take place; 3. Nine major glaciations are expected, all of them with maximum magnitudes around 10-15% lower than LGM level. The natural glacial cycles of the future are strongly dominated by 100 kyr cyclicity as were the past ones.

## 4.3    Possible future trajectories under different anthropogenic $CO_2$ emissions scenarios

The evolution of the ice volume even in the low-emissions anthropogenic scenario (E = 500 PgC, which is lower than already emitted) is significantly different from the natural one in the next 200 kyr, indicating a long-lasting impact of fossil-fuel $CO_2$ releases (Fig. 7b). The uncertainty levels for the next 150 kyr and from 400 to 500 kyr are greatly reduced, with most of the solutions agreeing on ice-free NH conditions. Under this scenario, only after half a million years is the Earth system able to return to an evolution similar to the one expected under natural conditions. Under this setting, the next full glacial conditions are not expected to be able to occur before 180 kyr from present, implying a delay of at least 90 kyr from the natural expected evolution which is consistent with Archer and Ganopolski (2005) and Ganopolski et al. (2016). In particular, the ice volume evolution of the Best Solution under this scenario differs significantly from the natural one during two periods: [0 kyr, 200 kyr] and [400 kyr, 500 kyr]. In both time spans glaciations fail to be triggered for this scenario (Figs. 7b and 8a). In [200 kyr, 350 kyr] the low-emissions Best Solution trajectory is similar to the natural one, except for a lag of ~10 kyr. $CO_2$ and global mean surface temperature anomalies (Figs. 8b and 8c) pathways are also restored to natural conditions only after 500 kyr from present.

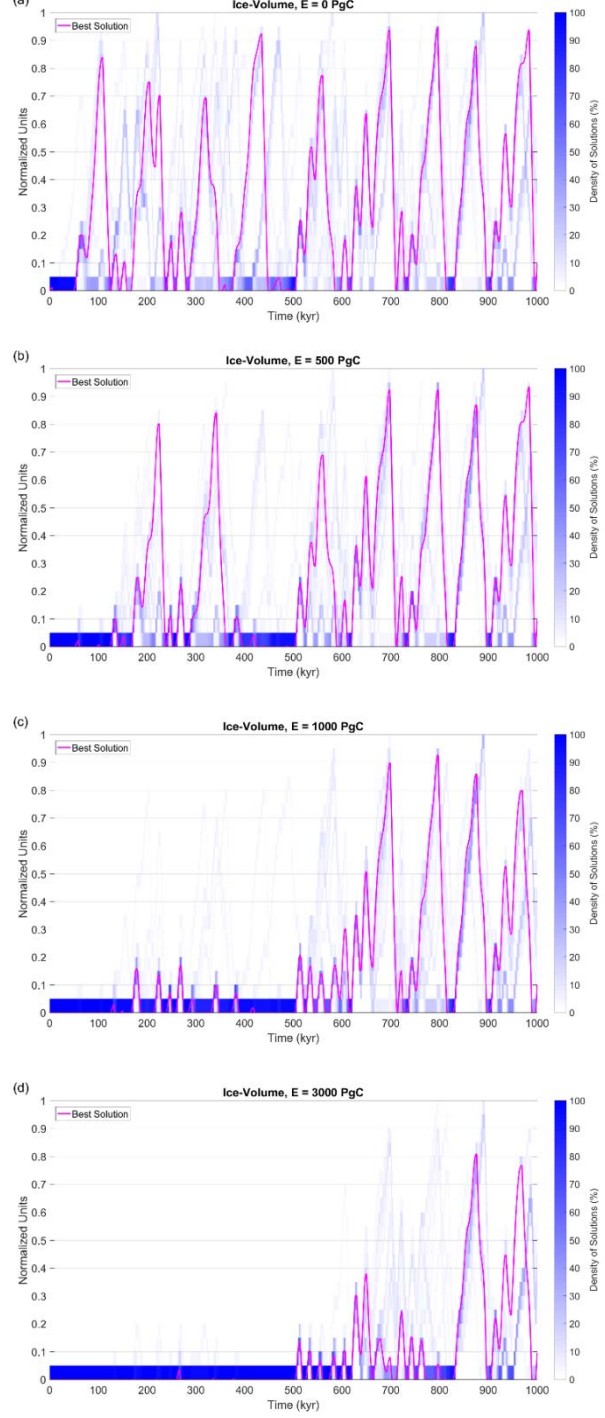

**Figure 7: Simulated ice volume for different future scenarios: a. Natural scenario. b. Low-emissions scenario (E = 500 PgC). c. Intermediate-emissions scenario (E = 1000 PgC) and d. High-emissions scenario (E = 3000 PgC). The magenta line depicts the**

behaviour of the Best Solution, the blue shades represent the density of *Accepted* solutions per unit of phase space (Units are: 1kyr
for time, 0.05 for ice, 7ppm for CO₂ and 0.5°C for temperature).

For the intermediate case (E = 1000 PgC) the effect of anthropogenic $CO_2$ emission on the Earth system is even more long-lasting. The ensemble of possible future scenario indicates significantly different conditions from the natural ones in the whole

next half million years, with very low probability of considerable ice growth during the totality of this time span (Fig. 7c). In most of the solutions the full glacial conditions occur for the first time either in ~ 550 kyr or between 600 and 700 kyr from present (Fig. 3). In particular, the Best Solution is part of the latter group, with the next full glacial condition taking place only at ~ 670 kyr after present. Additional insight into the relevance of the parameter K for the future behaviour under anthropogenic influence is presented in Figure 8: the larger the magnitude of K, the larger the delay in the next glacial inception.

Finally, with a cumulative emission of 3000 PgC, the Earth system is expected, with high confidence, to experience almost ice-free conditions for the next 600 kyr (Fig. 7d). Under this scenario, the next full glacial conditions are not expected to occur before 670 kyr from present, implying a lag of almost 600 kyr from the natural scenario. For example, in the Best Solution, only two major glaciations are predicted, peaking in 880 kyr and with a magnitude 20% lower than LGM (Figs. 7d and 9a). In

the 3000 PgC scenario, half a million years into the future, the $CO_2$ is expected to be 300 ppm (22 ppm higher than preindustrial levels) and the global mean surface temperature still 0.5°C higher than preindustrial level (Figs. 9b and 9c).

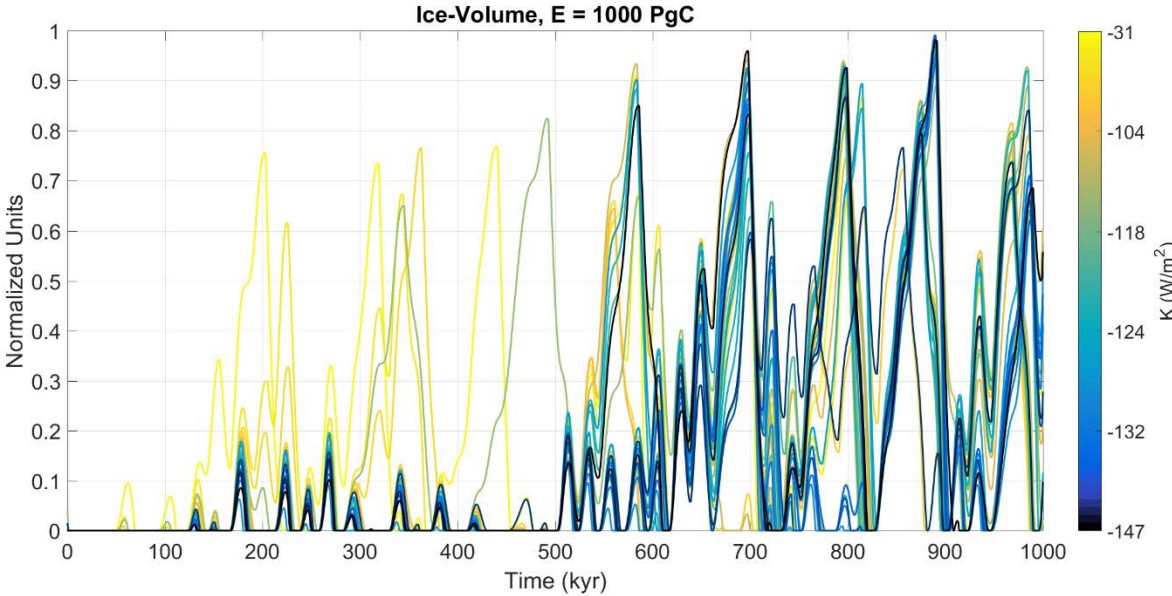

**Fig. 8: Ice volume time-series for each solution in *Accepted* for the intermediate-emissions scenario (E = 1000 PgC). Each solution is coloured according to its corresponding K value.**

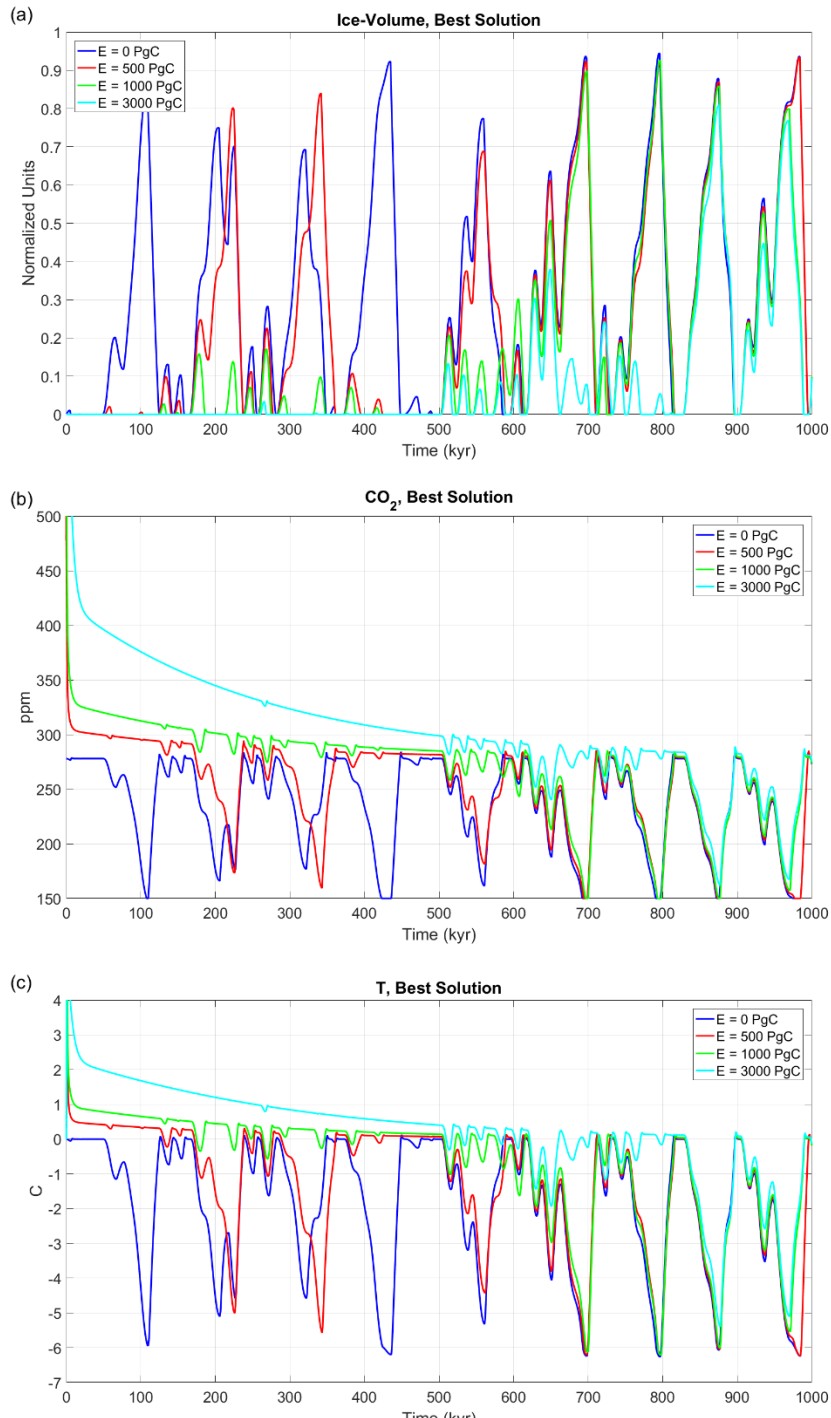

**Figure 9: Best Solution under different emission scenarios: a. Ice volume (normalized to unity), b. CO₂ (ppm), c. Global mean**
**surface temperature anomalies (C).**

## 5 Sensitivity of model results to the choice of constraints and parameters

In order to assess the robustness of the results to the model or parameter selection strategy, we performed a series of sensitivity experiments. Table 4 summarizes the settings for the Base experimental setup and five different sensitivity experiments. For each experiment, the optimisation process is carried out and a *Paleovalid* and *Accepted* sets of solutions are created (the definition of the *Paleovalid* might change from the standard one in equation 17 in some of the experiments, see Table 4). As a summary of results, Fig. 10 depicts the range of timings of the next full glacial conditions for each experiment and anthropogenic emissions scenario.

In the first sensitivity experiment (Sen_MBT) the ad-hoc imposition related to MBT (Eq. (12)) is lifted. Naturally, the correlation between modelled and paleo ice volume in Sen_MBT is slightly lower than in the base model formulation (the ensemble mean across *Accepted* drops from 0.79 to 0.72). The ensemble mean timing of next full glacial conditions under the natural, low and intermediate emissions scenarios for this setting does not substantially differ for the one in the Base experimental configuration. For the high emissions scenario, though, the timing for next full glacial conditions spans the full 200 kyr - 860 kyr period, with the ensemble mean being around 600 kyr in the future (~ 150 kyr earlier than in Base).

Second, we test the sensitivity of the results to the choice of ECS of the model. We report on the results ensuring an ECS of 3°C (instead of 3.9°C in Eq. (14)). The results of this experiment (Sen_ECS) indicate that the main conclusions are not overly sensitive to the change in the selected ECS for the natural, low and intermediate emissions scenarios. However, for the high emissions scenario the timing of next full glacial conditions is confined to 520 kyr – 620 kyr, somewhat earlier than expected using the Base configuration.

In the third experiment, Sen_future, we modify Eq. (15) so that it does not include any conditions on the future. Without this constraint, the new *Paleovalid* ensemble contains three solutions (out of 400 ensemble members) for which glacial inception occurs at some point between present and 20 kyr into the future. However, those solutions do not fulfil the requirements for being included in *Accepted*. The main results remain largely unchanged with respect to the Base experiment.

Fourth and fifth, we test the sensitivity of the results to the selected optimisation metric. In the fourth experiment (Sen_$C_v$_C $_{CO2}$) we select parameters in order to jointly maximise correlations between modelled and paleo ice volume and $CO_2$. For that purpose we maximise $C_v + C_{CO2}$.

We find that for the Sen_$C_v$_C $_{CO2}$ (Base) experiment the *Accepted* ensemble mean correlation between modelled and paleo ice volume is 0.71 (0.79) and between modelled and paleo $CO_2$ is 0.55 (0.5), therefore showing similar performance in terms

of correlations. The most notorious difference between this and the Base setting occurs in the natural scenario, for which the majority of the solutions agree next full glacial conditions will develop in 130 kyr from present.

In the fifth experiment, instead of optimising correlations, we select parameters to minimise the root mean squared error (RMSE) between modelled and paleo ice volume. Using a RSME instead of a correlation optimisation strategy leads to the strongest changes in the timing of the next full glacial conditions for the low and intermediate emissions scenarios: the ensemble mean occurs ~200 kyr and ~300 kyr earlier than in the Base configuration, respectively. While these differences in the ensemble mean behaviour are substantial, the spread of possible timings within the *Accepted* set is still comparable.

**Table 4: Summary of settings for sensitivity experiments.**

| | Experiment Name | | | | | |
|---|---|---|---|---|---|---|
| | **Base** | **Sen_ECS** | **Sen_MBT** | **Sen_future** | **Sen_Cv+CCO2** | **Sen_RMSEv** |
| **MBT condition (Equation 12)** | Yes | Yes | No | Yes | Yes | Yes |
| **ECS (in Equation 14)** | 3.9°C | 3°C | 3.9°C | 3.9°C | 3.9°C | 3.9°C |
| **Future condition (Equation 15)** | Yes | Yes | Yes | No (modified to: v(t=0)<0.025) | Yes | Yes |
| **Optimisation target** | Maximise Cv | Maximise Cv | Maximise Cv | Maximise Cv | Maximise Cv + CCO2 | Minimise RMSEv |
| **Condition on P to be in Valid** | Cv(P,-800kyr,0kyr)>=0.7 | Cv(P,-800kyr,0kyr)>=0.7 | Cv(P,-800kyr,0kyr)>=0.7 | Cv(P,-800kyr,0kyr)>=0.7 | Cv(P,-800kyr,0kyr)>=0.7 | RMSE(P)<0.25 * |

*0.25 corresponds to the RMSE of forecasting the average ice volume in the period [-800 kyr, 0kyr].

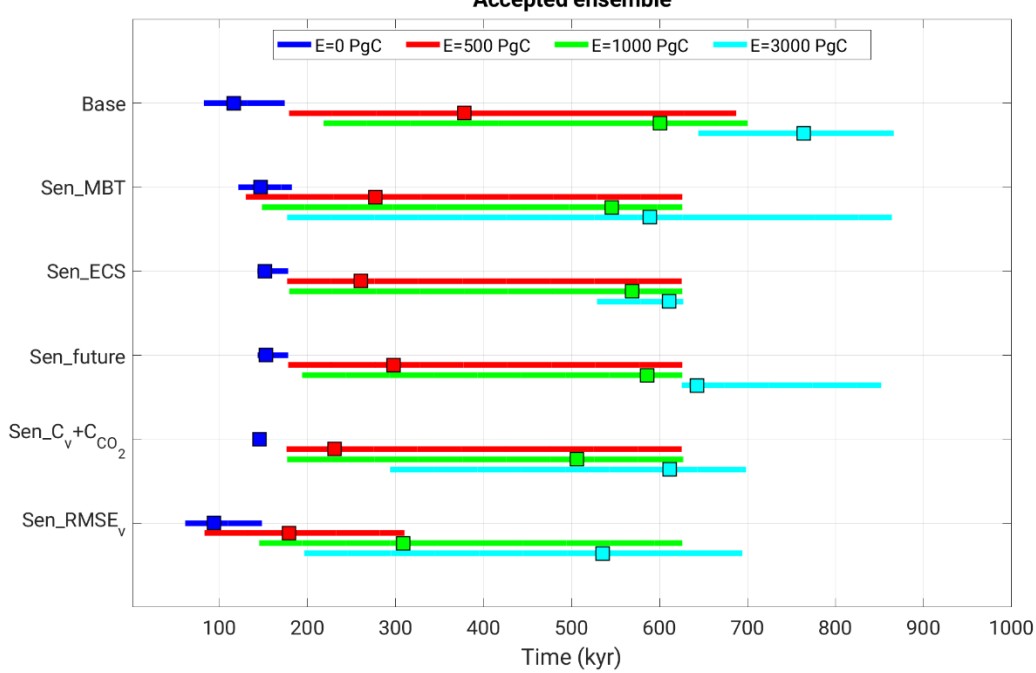

**Fig. 10: Timing of next full glacial conditions (first time after present when v=0.5) for different model configurations and emission scenarios, considering the _Accepted_ solutions.**

For a certain configuration and emission scenario, line segments indicate the 5-95% percentiles of the timings while squares indicate the ensemble mean.

## 6    Summary and conclusions

We propose a reduced-complexity process-based model of the evolution of global ice volume, atmospheric $CO_2$ concentration and global mean temperature at multi-millennial and orbital timescales. The model only external forcings are the orbital forcing and (for the future) anthropogenic $CO_2$ cumulative emissions. The model consists of a system of three coupled non-linear differential equations representing physical mechanisms relevant for the evolution of the climate – ice sheets – carbon cycle system. Model parameters are selected to achieve the best possible agreement with reconstructed ice volume variations during past 800 kyr and consistency with CLIMBER-2 model results. Global temperature anomaly at the Last Glacial Maximum and the current best estimate for the equilibrium climate sensitivity have been used as additional constraints on model parameters. Furthermore, the solutions are bounded to account for the difference between pre- and post- mid-Brühnes transition interglacial conditions and a minimum for $CO_2$ concentration is set to avoid unrealistic solutions.

The model is successful in reproducing the natural glacial-interglacial cycles of the last 800 kyr, in agreement with paleorecords, replicating both the timing and amplitude of the major fluctuations, with a correlation between modelled and reconstructed global ice volume of up to 0.86. The model also shows a promising performance when evaluated against data not used for training, indicating a certain skill in predictive mode.

Using different model realizations we generate an ensemble of possible future trajectories for the evolution of the Earth system over the next one million years under natural and three different anthropogenic $CO_2$ emission scenarios.

In the natural scenario, in which anthropogenic $CO_2$ emissions are assumed to be zero at all times, the model assigns high probability of occurrence of: 1. Long interglacials in two periods: from 0 till ca. 50 kyr after present and between 400 kyr and, 500 kyr after present and 2. The next glacial inception will most likely occur ~50 kyr after present with the development of full glacial conditions ~90 kyr after present. It is also clear, however, that even though there is a high level of agreement in the trajectories during the past 800 kyr, their paths tend to diverge for the future indicating that the past does not perfectly constraint the future evolution of the climate – ice sheets – carbon cycle system.

The selected model versions exhibit a large sensitivity to fossil-fuel $CO_2$ releases, with even already achieved emissions (500 PgC) causing a behaviour significantly different from the natural one over extremely long periods of time. The model predicts that a fossil-fuel $CO_2$ release of 500 PgC will make the beginning of the next Ice Age highly unlikely in the next 120 kyr and a delay of half a million years is a possibility that cannot be ruled out. Cumulative fossil-fuel $CO_2$ emissions of the order of 3000 PgC (which could be achieved in the next two to three centuries if humanity does not cease fossil-fuel usage; Meinshausen et al., 2011) will most likely lead to ice-free conditions over the NH continents throughout the next half million years. Under this scenario, the probability for the next glacial inception to occur before 600 kyr is extremely low. Thus our study demonstrates that, in spite of large uncertainties in model parameters, there is a qualitative difference in the long-term Earth system trajectories (Steffen et al., 2018) between cumulative emissions of 500 and 3000 PgC.

However, it is clear that large uncertainties pose serious challenges to our ability to model the deep future. First, the present study reveals that the results are very sensitive to the relationship between critical insolation threshold for glacial inception and $CO_2$ levels. In turn, this relationship is poorly constrained by the paleoclimatic data because during previous interglacials $CO_2$ was similar to or lower than the preindustrial $CO_2$ level. So far, this relationship has been analysed only with the CLIMBER-2 model (Calov and Ganopolski, 2005; Ganopolski et al., 2016). This is why in this work, we assumed the uncertainty in this relationship of 100%. Performing experiments similar to those described in Ganopolski et al. (2016) but with more advanced Earth system models can help to reduce uncertainties in future projections. Second, the knowledge on the long-term ($10^5$-$10^6$ yrs) carbon cycle dynamics remains poor, primarily because of the lack of accurate empirical data. In

particular, the results from Lord et al. (2016) used in this paper to simulate the decay of excess $CO_2$ in the atmosphere are largely theoretical and based on the assumption that the long-term carbon cycle is regulated solely through the silicate weathering omitting the role of other possible processes (i.e. organic carbon burial in marine sediments, kerogen weathering).

We consider our approach potentially useful in providing projections of the possible future Earth system evolution. However, it has a number of limitations. In particular, one key assumption of our procedure is that ice volume is at all times equal or higher than the preindustrial level. While a possible future melting of Greenland will not substantially affect this hypothesis, a possible future melting of Antarctica definitely will. Therefore, we must emphasize that the methodology presented here is not applicable for anthropogenic scenarios in which a substantial melting of the Antarctic ice sheet could be expected

(cumulative emissions higher than 3000 PgC).

**Appendix A: Estimation of predictive skill**

When a model is developed with a predictive goal (like in this study), it is important to have an assessment of its predictive skill. The best possible estimations of this skill are obtained by evaluating how well the model performs when presented with previously unseen data.

To follow the cross-validation assessment criterion (Stone, 1974), we divide the period [-800 kyr, 0 kyr] in halves: P1 = [-800

kyr, -400 kyr] and P2 = [-400 kyr, 0 kyr]. First, P1 is labelled training set and P2 validation set. Second, the model is trained using the information in P1: we run the Optimisation scheme to maximise the correlation between modelled and paleo ice volume during P1; we then accept all the models (parameter combinations) yielding correlations higher than 0.7. Third, all the accepted models are validated using the information in P2. We select three different performance metrics: the correlation between modelled and paleo ice volume, between modelled and paleo $CO_2$ or between modelled and global mean surface

temperature anomaly, calculated during P2. Finally, we repeat the previous three steps but labelling P1 as validation and P2 as training sets, respectively. For each metric, the average that the models achieve on the two possible validation sets considered is the estimate of the model's predictive skill.

In Table A1 we summarize the results of this cross-validation assessment, showing the results for the three performance

metrics. For reference, Table A1 also displays the values of the metrics when calculated over the training period (instead over the validation period). Results are the average of two possible cases: P1 as training and P2 as validation sets, or P2 as training and P1 as validation sets. As expected, when the same period is used both for model training and evaluation the results are in better agreement with paleoclimate data: the ice volume metric is 0.78, the $CO_2$-metric 0.52 and the temperature-metric 0.58. These values drop to 0.49, 0.36 and 0.34 respectively, when the independent validation set is used to calculate the metrics.

Even though the drop in performance is not negligible, the model skill is still substantial in particular for predicting ice volume.

**Table A1: Estimation of Model Predictive Skill. Only solutions leading to a correlation between modelled and paleo estimation of ice volume higher than 0.7, calculated over the training period, are considered. See text for more details on the training and validation periods used.**

| Ensemble-Mean Correlations Between Modelled and Paleo-data | Calculated over the Learning period | Calculated over an independent validation period |
| --- | --- | --- |
| Ice volume | 0.78 | 0.49 |
| $CO_2$ | 0.52 | 0.36 |
| Temperature | 0.58 | 0.34 |


**Code/Data availability**

Code and data used in this study can be obtained from: https://doi.org/10.17605/OSF.IO/KB76G (Talento, 2021).

**Author contribution**

AG conceived the original idea. ST performed all the calculations and produced figures. Both authors contributed to the analysis of results and writing of the manuscript.


**Competing interests**

The authors declare that they have no conflict of interest.

**Acknowledgments**

Financial support for this study was provided by the Swiss National Cooperative for the Disposal of Radioactive Waste (NAGRA).

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
