# Peer review of "Reduced-complexity model for impact of anthropogenic CO2 emissions on future glacial cycles"

_Earth System Dynamics, 2021_

## Referee Comment (RC3)

**Review of Talento and Ganopolsky paper on "Evolution of the climate in the next million years: A reduced-complexity model for glacial cycles and impact of anthropogenic CO2 emissions".**

May 28, 2021

**1 Conclusion : Rejection**

Glacial-interglacial cycles being very slow processes require a "proficient" and a "master" model for long-term temporal prediction. Predictions are carried by phenomenological models often represented as low-order dynamical systems.

Low-order or reduced-order dynamical models (often represented by a set of coupled differential equations) are tractable and insightful by emphasizing the most important dynamical features of the complex behavior of a given system, such as the (paleo)climate system, in our case. In the latter, important information about that complex behavior is lost because of the use of tractable equations leading to under-defined parameters in the model representing the underlying phenomena. In addition, these models are data-driven, which make their calibration/ the estimate of the model parameters and the forecast/prediction sensitive to the errors and uncertainty in the observational data. Therefore, using them to reproduce the current time and/or forecast the future, based on the past one, is highly dependent on the care and level of accuracy in calibrating and validating the model under consideration and specification of uncertainties.

Combining Physical representations with probabilistic estimations is a very strong adequate way to forecast long term climate, especially in Paleoclimate (Crucifix and Rougier 2009). This requires the following steps:

1. **Using a low-rder model to capture the very long term (millennial) of climate, under physical constraints. For instance the the three-dimensional stochastic system of Saltzman and Maasch (1991).**

The authors here, designed/formulated three equations, using knowledge of the behavior of ice on millennial scales. Many assumptions has been advanced without any strong argumentation/reason/justification.

2. **Treating the estimate of the model parameters and the forecast probabilistically.** One way of doing that, as in Crucifix and Rougier (2009) to assess the next glacial inception, is by inferring the different parameters within a Bayesian framework that allows for (1) parametric uncertainty and (2) for the limitations of the model, by using Sequential Monte Carlo technique ('particle filter').

3. **Verify the accuracy and validating the model statistically and by checking the reproduction of physical phenomena.** Different physical assumptions may lead to dynamical systems with dynamical properties that are similar enough to produce a convincing visual fit on palaeoclimate data [61]. challenge is, therefore, to operate a model selection on more stringent criteria than just fitting some standard time series.

The necessary steps (1 to 3 above) required to assessing the future glacial inception under different levels of carbon dioxide emissions, have been inadequately followed and their related approaches incorrectly applied by the authors. The work by Talento and Ganopolsky does not reflect any aspect of the correct modeling approach towards a probabilistic forecast of climate.

This work stated that, what is needed is a "quantitative probabilistic assessments" as a must to assess on a very long term of carbon dioxide emissions on changes in temperature. As stated, this can be useful under the present challenges of climate change requesting carbon dioxide storage, which then requires an adequate assessment of storage system under changes in the future environment due to human activity".

I do agree. However, this has not been done here. This is what the authors tried (wanted?) to accomplish, but failed unfortunately. This work has no provided any forecast neither probabilistic forecast of the climate. What has been done is a scenario simulation given a low-oder model (and even that, has has been inadequately assessed). They compared to a control simulation of future temperatures, where the anthropogenic emissions are null, a set of predicted simulations under low, medium, and high level of emissions. As carbon dioxide influences the coupled system temperature and ice on a long scale, they proposed a simple model, to be able to simulate a very long term of climate.

In addition, the statistical modeling part, is applied incorrectly and many chosen assumption are unjustified. The model selection procedure (which model, among different alternatives, explains the observations best) has not been carried correctly either. No future forecast, or prediction (even under scenarios), especially when using observations, can be carried out non probabilistically. And when dealing with time series, it is even more critical to attach more attention to (1) more adequate statistical approaches for long term and multiple steps

ahead forecast and to (2) adequate model validation and selection, where the predictive ability of the model must be verified given the length/characteristics of the observations (here, paleorecords).

I explicated all these aspects in the document, where, I tried, despite the low level of the manuscript, to advise a way to correct the statistical modeling part, improve the paper, and follow a better predictive approach. The authors must chose one of the two research axes proposed below.

**This paper cannot be published as it is and must be rejected.** This work is not mature enough for publication. It needs a profound revision and rework. Concepts are being mixed and the goal itself is unclear to the authors. The framework and the selected statistical modeling/validation approaches are weakly justified and poorly and/or incorrectly applied and most importantly the methodology is inadequate as it does not account for any source of uncertainty.

In a clear way and a more direct construction of the paper flow please, in a new version of the paper, chose one of these working axes:

1. Reconsider the whole work by implementing a probabilistic forecast approach, refer to Crucifix and Rougier (2009) and Crucifix(2012). Here, the inference should imply confronting a model with observations. "This inference process may take the form of a calibration procedure (update our knowledge on parameters on the basis of observations) or a model selection procedure (which model, among different alternatives, explains the observations best)" (Crucifix 2012).

2. Correct and adapt this work to reflect the framework of scenario simulation using a pre-constrained simple model. One way to make it publishable is to reformulate the goals and to position the work in literature related to scenario based for decision making and not as a new probabilistic model for ice ages forecast (at all!).

   This part will require repositioning of the work in a more adequate framework, adapting the corresponding review of literature, choosing a correct approach for calibration, designing experiments under constraints for the optimization process (during the calibration process, to sample values of the parameters with appropriate sets of combinations under constraints) and fixing the vocabulary and giving a more adequate justification for all modeling choices.

**2   Main Comments**

The authors formulated their predictive model as consisting of a system of three coupled non-linear differential equations, representing physical mechanisms relevant for the evolution of the temperature using a coupled Ice Sheets – Carbon cycle System in timescales longer than thousands of years, for different selected

emission scenario. Many constraints have been introduced, from physical knowledge of the system, to infer the values of the parameters in the three equations model. What they tried to do is to sufficiently decouple the selected behavior from the rest of the variability to justify the fact that simple dynamical systems may capture the dynamical properties of this mode, and to learn about the mode from palaeoclimate observations.

Here, using the paleorecords, the calibration was applied inadequately: (1) fitting the parameters by maximizing a correlation coefficient (2) using the solutions of the optimization process as a set representing possible solutions of the predictions (and used as probabilistic estimates) (3) selecting the model with a very weak statistical criteria and unjustified threshold (0.7 for the correlation coefficient) : this is not a probabilistic forecast.

A more adequate calibration method for the model as well as a more adequate verification and validation method of the predictive ability of the model are a must: any other choice must rely on a probabilistic treatment of the parameter and allow estimating uncertainty of the predictions. As stated in Cruficix (2012): "In a statistical inference process, the observations should be a plausible outcome or realization of the model. This makes sense only if the model has a stochastic component, which describes its uncertainties, limitations, and the noise that emerges from the chaotic motions of the atmosphere and oceans".

Two main approach : one can chose to handle the challenge of probabilistic forecasting long-term climate, or

1. Stochastic dynamical systems are used for inference on palaeoclimate time series.

2. Bayesian methodology, because it allows the integration of physical constraints in the form of prior distributions on model parameters. The Bayesian formalism is also naturally designed for model calibration, selection and probabilistic predictions (please, check Bayesian methods for selection and calibration of dynamical systems on noisy observations and the paper by Crucifix, 2012).

**2.1  Critical comments and questions to be absolutely addressed**

1. Neither the 100ky duration of ice ages, nor their saw-tooth shape were predicted by Milankovitch. Please check literature and update the knowledge.

2. This work is absolutely not a forecast work and nor a probabilistic forecast. This should absolutely be **addressed and corrected.** Without it, the paper cannot be published. This is a scenario based work, even not from a sensitivity nor a what-if scenario framework. as they only used three main scenarios (low, medium and high levels of starting point of carbon dioxide).

3. This work embraced a method based on many unjustified simplifications and approaches. Please, Address the reasons and strong justifications why you accounted for the mentioned simplification (assumptions) of all the climate processes and the estimation of the parameters:

(a) The modeling approach: from line 264 "Finally, we approach the task of the selection of set of parameters P as a non-linear optimisation problem with equality and inequality constraints. We wish to find P to maximize the optimization target function( correlation criteria)" to line 289: This is not acceptable for forecasting, probabilistically or not.

how do you justify the selection of the best model, or calibration of parameters, while this is done via correlation: it is not probabilistic the way you did it. Neither it is an adequate one. It is like selecting the curve that suit you well given one aspect in the data, which might be linked to linear correlation!

Why did not you considered any Least-Squares (Model Fitting) Algorithms? How about validation using scoring to select the best model, there are many statistical criteria to select the best model, to fit and calibrate statistically and under constraints.

Honestly, either I did not understand at all what you did, or it is looking more like a patchwork using inadequate pieces! especially seen in the following "See Appendix A for a discussion of the dependence of model performance on the choice of this time interval. To select parameters that will optimise correlation at the same time as providing magnitudes in accordance to empirical estimations, an equality constraint is enforced: the maximum ice volume must be equal to 1 within a tolerance of 0.15 (in nondimensional units). Finally, the inequality constraint is given by Eq. (14).": this is really not acceptable.

(b) Validation set: Did you check the validity of the length of the time series used for calibration? how sensitive are the results given the the length of the time series used for calibration? how did yo find the optima length?

(c) Strong justification for not using appropriate probabilistic forecast models and adequate methods for calibrating the chosen one. No palaeoclimate record is dated with absolute confidence, so how do you account for the errors in the calibration data?

(d) Running multiple realizations by varying the model parameters: this is what is needed. but, this is not what you did! how did you considered that being probabilistic?

What you did, is simply taking the solutions offered by the optimization process for multiple combinations of the parameters, choosing the sets that maximize the correlation with an unjustified threshold of 0.7! then using them as equivalent of multiple realizations of the

predictive model to conclude about a probabilistic forecast! This is inadequate and inaccurate. what you did here, is just finding the best set of parameters for your model.

The way this has been done does not even give you the credible interval of the values for the parameters (and with an insufficient number of simulations as you run 1000, picked less then 400 and you have 9 parameters!).

Once you calibrate your model with the best set of parameters, verify it and calibrate it, then you should run an MCMC or any other sampling, to generate a set of probable realizations of your model, given the range of adequate values of the parameters, and a justified distribution for each parameters in the model.

Please check the literature for a proper way to do it including the optimal number of realizations which is far from $10^3$.

(e) Results and from Figures: statements of results adequacy not valid

   i. Figure 1: I really do not see that your predictions coincide with reconstructions. especially clear in figure 1-b!

   ii. The magnitudes are not well reproduced at all.

   iii. in appendix A: you have a correlation of 0.36... No comment!.

(f) The correlation level of 0.7, although arbitrary, guarantees a good fit to the paleo climatic ice volume record : this must not be used at a first place, it should certainly not be chosen arbitrary, and the figures do not show a good fit neither your correlation coefficients (using correlation at a first place is a problem in itself)

   i. correlation is not an adequate criteria to assess goodness of fit in time series the way you did it

   ii. it is not a good way to assess relation or association between time series (such as ice volume and $CO_2$)

   iii. correlation is insufficient by itself, and it assumes linear relations only.

   therefore the comparison in between paleorecords and model output is weak, incorrect and incomplete.

4. Carbon dioxide curves: your choice of the evolution need to be justified. why should it be decreasing exponentially?

5. The relationship between critical insolation threshold for glacial inception and $CO_2$ levels is known and must be analyzed using an appropriate sensitivity analysis.

6. Calibration/ Validation need to be done correctly

   (a) The validation part (crf. appendix A) is very weak. It has to be addressed with more adequate diagnostics for time series, especially graphical ones.

(b) You must use a statistical criteria, more adequate to select the best model. large literature on that.

(c) A sensitivity analysis or history matching plus an experimental design : would have been of high aid in this case where the hyperparameters have many constraints and we only know the range of the parameters. designing a space filling set of combined parameters while constraining them in the space formed by all them. Run the optimization algorithm with only realistic combinations.

7. Please use the term "pacemaker" instead of "control" when referring to the astronomical forcing. The theory of ice ages has already evolved and, it is established that the astronomical forcing, especially for the assessing the particularity of the 100ky precession enigma (See Ditlevson and Crucifix (2017) On the importance of centennial variability for ice ages) :"changes in eccentricity modulate the amplitude of precession peaks at a period of about 100 ka, but the spectrum of insolation time series do not contain an amplitude peak at this period. Source here (...) With this possibility in mind, the astronomical forcing is often prudently presented as the "pacemaker" of an internal oscillation rather than a primary "driver".". you can refer to the work by De Saedeleer, Crucifix and Wieczorek, https://dial.uclouvain.be/pr/boreal/object/boreal:119083 for a more systematic verification of the concept of forcing during ice ages.

8. The glacial inception problem

(a) In line 300, *"(...) we analyse the critical insolation – $CO_2$ relationship during glacial inception episodes for the different model realizations derived from Valid and compare them with Ganopolski et al. (2016)."*Validating a work using results from another simulation, does not seem accurate to me.

(b) The glacial inception problem has been treated probabilistically and by using conceptual models. This study must be taken into account: refer to the work by Crucifix and Rougier 2009 on On the use of simple dynamical systems for climate predictions: A Bayesian prediction of the next glacial inception.

    i. How do you position yourself comparing to the work by Crucifix and Rougier 2009?

    ii. Why not to use the same idea for the modeling part?

9. « This approach, obviously, is not applicable for a possible future Antarctic and Greenland melting under high $CO_2$ concentrations. This is why we do not consider future sea level rise above the preindustrial level and it is required that v≥0 at any time» : Why? How do you justify that?

Crucifix, M., Rougier, J. On the use of simple dynamical systems for climate predictions. Eur. Phys. J. Spec. Top. 174, 11–31 (2009). (DOI).

Ditlevsen, P., & Crucifix, M. (2017). On the importance of centennial variability for ice ages. Past Global Changes Magazine, 25, 152-153.

Crucifix, M. (2012). Oscillators and relaxation phenomena in Pleistocene climate theory. Philosophical Transactions of the Royal Society A: Mathematical, Physical and Engineering Sciences, 370(1962), 1140-1165.

De Saedeleer, B., Crucifix, M., & Wieczorek, S. (2013). Is the astronomical forcing a reliable and unique pacemaker for climate? A conceptual model study. Climate Dynamics, 40(1-2), 273-294.

**2.2 [Title] Need to be changed**

The title must reflect the main goal of the paper. The paper is more on assessing the impact of anthropogenic $CO_2$ emissions on the next $10^3$ ky for recommendations on the the evaluation of geological disposal systems. The response to future environmental changes driven by a combination of natural (astronomical variations) and anthropogenic (fossil fuel emissions) forcing. Moreover, the only climate variable considered in this study is temperature (not representative of climate as a whole).

**Suggestion** : impact of anthropogenic $CO_2$ emissions on temperature in the next million years: assessment with a reduced-complexity model for glacial cycles.

**2.3 [Section 1: Introduction] Need rewriting. It is not attractive nor well developed: the introduction must reflect the main subject.**

**Mainly:** Rearrangement of the ideas from the main purpose of the paper then the necessary supporting facts! In addition, the introduction must highlight the advantage/choice of using this specific conceptual model in a more relevant way. I think, here we need more details and justifications on the formulation of the framework/method/approach then reiterating about the ice ages and the Milankovitch theory (which can anyways be re/moved).

- Lack of consistency in the flow of ideas, lack of referencing on the main subject. It needs rewriting.
- I join Referee 2, to refer to the technical report by Lord et al.
- Please, refer to the work by Crucifix and Rougier (2009) and in a more profound way the cited paper Cucifix (2012). Of course, you must add complementary papers in the same line as these two.

- It is well established that climate change is a human activity induced. May be drop lines from 25-45 in the introduction, and use them as supporting facts for supporting the following points in order: by explaining

  **(1)** the goal which is more related to "the challenge of the permanent storage of the radioactive waste" and " *The evaluation of geological disposal systems in response to future environmental changes, driven by a combination of natural (orbital variations) and anthropogenic (fossil fuel emissions) forcings (e.g. Lord et al., 2016) is, therefore, mandatory*"

  **so** start the introduction with line 47 (while adapting the text of course).

  **(2)** Why we need to consider a model for glacial cycles and why we must include, in the simulation study, natural and human induced factors : human activity induced impacts on climate change has a long term impact.

  **Use** lines [25-45] as supporting facts

  **Or** use it to support the justification of the calibration part in the methodology section, Line 85 when discussing the ice ages.

  **(3)** Then proceed with line 54 starting from "*However, these timescales are(...)*".

  **(4)** Please, add a more adequate review of literature related, specifically, to the subject of analyzing or assessing the impact of carbon dioxide concentration variations (under scenarios) on the stability/evaluation of geological disposal systems.

  **(5)** when you say "to this end" : I do not see how you account for the "quantitative probabilistic assessments." in your proposed model. why not to announce already your approach here **in a concise way**. because, conting for the quantitative probabilistic assessments is not part of the defined/designed predictive model, the " *reduced-complexity process-based model of the coupled climate – ice sheets – Carbon cycle evolution, whose only external forcings are insolation and cumulative anthropogenic* $CO_2$ *emissions.*"

  **Please** specify that the interest is the evolution of temperature. Justify why (linking it to the main subject of the paper which is "the challenge of the permanent storage of the radioactive waste and The evaluation of geological disposal systems").

  **(6)** Please, position more adequately your contribution. It is not clear from the text. Please, point out the lack in the literature (If there is so) and the breakthrough of your study and advantages of using your approach/choice of model and parameterization. For instance, what was the outcome and the lack(s) in the work

of Archer and Ganopolski (2005) based on Paillard's conceptual model?

And, why did you chose here the simulator based Earth model from Lord et al.?

**(7)** Please, refer to the most up to date theory of the astronomical forcing instead of Milankovich. check the paper by Curifix and Rougier (2009) for a detailed explanation and the theory and its history.

**(8)** Note that Milankovitch's theory is missing the dynamical aspect of climate's response and that the Glaciologist Johannes Weertman (J. Weertman, Nature 261, 17 (1976)) is the one who addressed the evolution of ice sheet size and volume by means of an ordinary differential equation (ODE), "thereby opening the door to the use of dynamical system theory for understanding Quaternary oscillations" (Crucifix and Rougier, 2009). This need to be highlighted in your paper and used as a reference as your work is about modeling using ODEs.

**2.4 [Section 2: Model and datasets]**

**Form**

Start with the set of all equations where equation of temperature will be first, then then explain the need to parametrize each of them.
So:

- Start with Subsection 2.1

  Please, shall you design a flowchart to show all the parts of the modeling framework.

  addd a table with all the parameters to be inferred during the calibration process

  add a table gathering all notation, acronyms and definitions of variables, put here or in the appendix.

  Introduce the set of equation first (3 equation while starting with the temperature one).

  define the parameters, use lines 204:208.

- Follow up with subsection 2.2: details of the the equations

  then explain and explicit each equation, its meaning, goal, parametrization.... and here you need just two subsubsections. (one for ice and the other for $CO_2$, no need for temperature as it is in sec.2.1)

- Follow up with subsection 2.3: explicit the constraints... and so on (subsection 2.5 in the draft paper)

  start from line 215.

- Subsection 2.6 is used for describing the data used for validation: please, move it to section 3 (model performance).

**Comments on the method: critical to be addressed**

1. How do you account for uncertainties in the observational data while calibrating the model?

2. How do you justify the choice of 0.7 as acceptable for the correlation coefficient?

3. How do you justify the formulation of changes in temperature as a linear combination of global ice volume and logarithm of $CO_2$ concentration?

   This part need a more thorough justification, explanation, development.

4. calibration using maximization of the correlation coefficient?! this really need to be explained and justified and proved working.

5. How do you qualify your calibration/modeling/prediction method?

**2.5   [Model performance]**

This part has to be done appropriately, once the modeling part is fixed and an appropriate calibration method is selected.
This part should be applied statistically to verify and validate the calibrated model.
Comparison of the model predictions with paleoclimate data (reconstructions) should be assessed within the calibration process.
The length of the calibration time series should be assessed (assess the predictability of the model given the length of the time series).
Correlation should not be used as a validation criteria!
Please check literature for validating models calibrated for time series: this is what you need to learn and know, to work in this subject and write your paper.

**2.6   [Conclusion]**

The conclusion has to be adapted and rewritten with all the paper.
Just a note on: "*It is also clear, however, that even though there is a high level of agreement in the solutions' trajectories during the past 800 kyr, their paths tend to diverge for the future indicating that the past does not perfectly constraint the future evolution of the climate – ice sheets – Carbon cycle system.*" : I do not think this is absolutely necessary to mention: we know that and this experiment is not needed, the statement either.

**3 Secondary comments**

To help correcting/adapting/improving the work/paper, it would be beneficial to the authors to check definitions/methods/literature (in a general framework and then for time series, and in paleoclimate field) on the following:

- conceptual models

- predicting vs forecasting

- probabilistic forecast

- (probabilistic) sensitivity analysis

- simulating using scenarios

- decision making based on scenario assessment

- probabilistic calibration of models based on time series

- verification and validation of calibrated models (set of diagnostics)

**[General]** Please,

- Use one verb tense for adequacy. Also, either direct form with the use of "we" or indirect with the one other verb tense. Such as in lines 70 to 73.

- Remove the expression "can be found": where ever it is in the text, it has to be changed into an active voice verb, such us is +adequate verb ( displayed, shown, ...).

- Refer to the technical report of Lord et al., on the same topic " Modelling changes in climate over the next 1 million years"

- Refer to the work by Crucifix and Rougier 2009 on the probabilistic modeling of climate change on the glacial inception.

- The only variable that is important to address the problem of storage is temperature. I suggest to keep any other figure (ice and $CO_2$) in the supplementary material.

- Remove "please" in line 162 and 194, and if any other in the text.

- change "orbital forcing" into "astronomical forcing" wherever it occurs and adapt the text accordingly. For instance, a sentence such as "The orbital forcing f(t) depends only on astronomical parameters (eccentricity, precession and obliquity) " in line 206 is unnecessary.

**[line 28]** "Antarctic ice core records also show" : to be consistent with line 25 and the statement "Numerous paleoclimate records show (...)", avoiding the use of "also" would be preferable.

Proposition:

During this period, atmospheric Carbon dioxide ($CO_2$) concentration fluctuated nearly synchronously with the global ice volume, and $CO_2$ concentration during glacial times was up to 100 ppm lower than during preindustrial 30 time, as shown in Antarctic ice core records (Petit et al., 1999, Lüthi et al., 2008).

**[line 32]** "Earth's orbital parameters". These are astronomical parameters.

**[line 35]** May be more adequate using "supported" instead of "confirmed" as per verifying a theory by the aid of a reduced order model and/or a simulator which is not enough to infer knowledge for conforming a theory but verifying it or validating an aspect of it with a set of verifications (Reductionism based knowledge inference especially based climate simulators cannot be used as a tool to **confirm** anything).

**[Lines 70-73]** Need rewriting, adapting the verb tenses. Please stick to one verb tense for adequacy. Also, stick to one form passive or active ( "we" ). Better : if you use a direct simple style with present tense.

**[210]** Correct "in a good (see discussion below) agreement" to "in a good agreement (see discussion below)"

**[225]** use "condition" or "constraint" instead of "criteria" in "The last imposed criteria" for consistency with the text.

**[236]** why do you use "limitation". in all this section you are introducing constraints. use the term "constraints" everywhere and count them as being 7 in total.

---

## Author Comment (AC2)

We thank the reviewer for the insightful and constructive comments. Please find below a point-by-point response, marked in blue. In order to address some of the reviewer's questions and assess the robustness of the results to several criteria involved in the design of the model or parameter selection strategy, we performed a series of sensitivity experiments. The results from these experiments are briefly discussed here in response to specific comments and will be included in a revised manuscript.

**Response to reviewer 1**

This manuscript presents possible scenarios for the Earth's climate during the next million years. It brings interesting and new material to this rather overlooked and under-researched area. Overall, I am favourable to publication, but I have nevertheless several important comments that the authors should consider in a revised version of the manuscript. Most importantly, the whole exercise is based on shaky hypotheses: I am aware that there aren't many alternatives, but it is all the more important to present and discuss them thoroughly.

1 – Using Quaternary climate (and more precisely the last 800 kyr period) to calibrate a conceptual model to be applied to the future is certainly not the ideal choice since there are no very hot periods, but mostly glacial ones. The (partial or complete) melting of Greenland or Antarctica is therefore not considered, and the effect of high $CO_2$ levels cannot be calibrated. In other words, the whole exercise is more an extrapolation than an interpolation. This is something known to be quite dangerous. I perfectly understand this choice, based on the availability of data, but it remains nonetheless not satisfactory. This should be stated much more clearly in the paper.

We fully agree with the reviewer that the late Quaternary paleoclimate records alone are insufficient to develop and calibrate a semi-empirical model suitable for future simulations. This is why we used a hybrid approach, based on a combination of paleodata and model simulations. Namely, the description of natural climate variability ($CO_2$ around or below preindustrial) is calibrated against paleodata while model behaviour in the greenhouse world is derived from the results of the physically-based CLIMBER-2 and cGENIE models. (The role of future Greenland and Antarctic mass loss will be discussed below). Thus, we believe we use a scientifically sound methodology which does not require any "extrapolation". We will better address the topic in the revised version.

2 – Our knowledge of the long-term carbon cycle and of the ultimate fate of fossil-fuel carbon is also very thin, shaky or uncertain. The manuscript uses model results (Lord et al. 2016; based on Lenton et al. 2006) that have unfortunately no "real world" tests. The imposed exponential decay of carbon is therefore based on the (mostly theoretical) idea that the carbon cycle is regulated uniquely through silicate weathering. This approach neglects many other important processes that are known to have played a critical role on these timescales in the past and even today, like organic matter burial or kerogen weathering. Again, I do not contest the value of using a simple hypothesis, but this should be explained and discussed.

We agree with the reviewer that long-term ($10^5$-$10^6$ yrs) carbon cycle dynamics is poorly understood, primarily because of the lack of accurate empirical data. This is why we do not anticipate a significant breakthrough in this field in the near future. Since the issue of the the long-term future evolution of Earth system is now not only an academic one, but also of great practical importance, one cannot simply wait till all relevant problems will be resolved. In any case, the major uncertainties in the future Earth trajectories originate from the fundamentally unreducible uncertainties in the future human activity. This is why any future projections can

represent only a tentative answer to the question of what can happen in the future under certain assumptions (this is also true for the "IPCC reports"). Of course, we fully agree with the reviewer that the manuscript will benefit from a more substantive description and discussion of the most important assumptions and their limitations. This will be done in the revised manuscript.

To summarize, sentences like "we produce a probabilistic forecast" (line 14 in the abstract) are not acceptable. This is obviously not a "forecast" but only a possible scenario, based on our very limited knowledge of the dynamics of geological transitions in the past.

We are clearly not able today to "forecast" what the Anthropocene era will be, and this should be stated much more clearly.

We used the term "forecast" in a very broad sense – under forecast we meant future simulations to distinguish them from past climate simulations (hindcast). Obviously, no one expects to make accurate climate forecast for million of years when it is not possible even for the next 100 years. But since the term "forecast" can cause confusion, in the revised manuscript we will change "forecast" to "possible future scenarios".

Other comments:

3 – It appears that one of the most critical parameter, K, is not well constrained using the conceptual model or the chosen paleoclimatic dataset, as explained in §3.2.

*« Our results indicate that with the model derived in this study the possible values of the coefficient K range between -1279 and -31 W m-2, with a median of -393 W m-2 »*

Using results from an Emic model (CLIMBER-2, Ganopolski et al, 2016) the authors decided to select only a very small subset of solutions ("Accepted") that are all in the tail of the distribution of "Valid" solutions as shown on Fig.2. This appears as a strong shift in the overall strategy and raises a few questions:

First of all, as discussed above, using of CLIMBER-2 results (together with the cGENIE model) is an essential part of our modelling strategy, not a "shift". After reading reviewers' comments we realised that this misunderstanding likely arises from how we presented the methodology of model calibration. Our idea to present first all model realisations consistent with paleodata ("Valid" ensemble) and then to apply the additional constraint on the value of K based on CLIMBER-2 results was to demonstrate that paleodata alone are insufficient to calibrate the model intended for future projections. This is fully consistent with the first comment of the reviewer. Unfortunately, the reviewer interpreted the fact that most of model realisations which work for the past, have been then rejected by CLIMBER-2 constraint, as a problem of our method. In fact, our methodology works well: after applying all constraints we still have a sufficiently large ensemble of model realisations which successfully simulate past climate evolution and is consistent with the physically-based model. Thanks to this reviewer comment, we now understand how to explain our modelling strategy better.

- How does the correlation to data vary across the histogram on Fig.2? Are the "Valid" solutions close to the 0.7 correlation limit and the center of the distribution farther away from this limit?

  Please see Fig. 1b below showing the requested information on the variability of correlation between ensemble members and paleo data, according to the parameter K. It is clear from this plot that the "Accepted" solutions correlations with paleo data are not statistically significant different from the rest of the solutions in the wider "Valid" ensemble, which reinforce the statement that past climate data alone are insufficient to calibrate the model designed for the future.

[Figure]

Fig. 1 (based on Fig. 2 from manuscript): (a) Histogram for K (see Eq. (17)) across the different members of the *Valid* ensemble of solutions. Red bars correspond to the *Accepted* solutions. (b) Correlation between modelled and paleodata, for all the solutions in *Valid* averaged within the bins in the histogram shown in (a).

- The Ganopolski et al (2016) insolation-CO2 threshold is also based on a parameter selection using a comparison to (basically) the same ice volume data. The problem is therefore not that the paleodata does not constrain well the K parameter but that the chosen conceptual model and the CLIMBER-2 model do not represent the role of $CO_2$ onto the dynamics of ice sheets in the same way. Why do the authors choose to trust one model against the other? And to adjust on model on the other?

  Here we must respectfully disagree with the reviewer. Fig.2 (based on Fig.3b from Ganopolski et al. (2016)) shows that paleodata do not provide any constrain on K value at all. Only one parameter (namely parameter β in the formula for the critical insolation–CO2 relationship derived in Ganopolski et al. (2016)) is reasonably well constrained by the paleodata. Another parameter α which defines the slope of the curve (in our

manuscript it is named K) is not constrained by paleodata at all (see Fig. 2). This is why, this parameter in the critical insolation–CO2 relationship is derived in Ganopolski et al. (2016) solely from the result of model simulations and it is fully determined by the physics of surface mass balance module of CLIMBER-2.

Note, that in the simple model, the value of K is defined by the combination of two parameters – b3 and b4 in the Equation (6): K=-b4/b3. At first glance, it may look strange that equally good simulations of the past can be obtained with model versions which have completely different combinations of the b3 and b4 parameters as shown in Fig. 1. In fact, this is not surprising because three values entering equation (6) – ice volume, CO2 and insolation – are closely related to each other and ice volume record alone is insufficient to accurately determine the role of each of the factors. As the result, we come to the conclusion that paleodata cannot constrain K but this value is of crucial importance for future simulations. This leaves us with no other option than to trust CLIMBER-2 value, the only source of information about this characteristic available at present.

[Figure]

Fig. 2. The locations of previous glacial inceptions in the insolation– CO2 phase space. Any line (corresponding to K from 0 to minus infinity) passing though the blue domain is consistent with the paleodata. This is because glacial inception during the past 800 kyr occurred under rather similar CO2 concentrations but very different orbital forcings.

Reviewer asks "Why do the authors choose to trust one model against the other?" Because one model – CLIMBER-2 – is the Earth system model which simulates surface mass balance of ice sheet through the physically-based energy-balance approach. This

model accounts for the effect of short-wave and long-wave radiation, sensible and latent heat fluxes, effects of snow aging and impurities on surface albedo of snow. To the contrary, the model developed for this study is just a simple, semi-empirical model based partly on CLIMBER-2 results, partly on paleodata. But since paleodata provide no constraint on K, how these two fundamentally different models can be treated equally?

- More technically, how are inceptions defined in the conceptual model?

In the conceptual model we define that full glacial conditions occur when the ice volume (v) reaches 0.5 in normalized units. If full glacial conditions are reached at a time T, then the corresponding glacial inception is defined as the first time before T in which v>0:

$$Glacial\ inception = \sup\{t\ /\ \ t < T\ \&\ v(t) > 0\ \}$$

We will clarify this definition in the revised text.

4 – Another strong limitation concerns the simple addition of "natural" and "anthropogenic" carbon, as presented line 182:

*« In addition, we assume that natural and anthropogenic CO2 anomalies can be simply summed up and that at the preindustrial time the global Carbon cycle was in equilibrium. This is, obviously, a very strong assumption since even a rather small imbalance in the global Carbon cycle which is impossible to detect at the millennial timescales can result in a very large "drift" of the Earth system from its preindustrial state at the million years timescale. »*

Indeed. This is actually why the anthropogenic CO2 decreases through a small imbalance between silicate weathering and carbonate preservation. The conceptual model assumes that there is NO natural dynamics in the carbon cycle besides glacial cycles. On Fig.8b the CO2 is just following the imposed decrease in the absence of (northern hemisphere) ice-sheets. But what about a possible role of Antarctica? What about some internal dynamics? And even on the calibration period (Fig.4b) the CO2 results are quite different from the data. In other words, the added value of a dynamic CO2 component in this conceptual model is not obvious.

The reviewer here rises several different issues.

1. Indeed it is unknown how close to the equilibrium the preindustrial state of the global carbon cycle is. This is why we consider our key assumption that it was an equilibrium state as a "strong assumption". But, at the same time, this is not an unreasonable assumption. Based on all available reconstructions the $CO_2$ concentration was not higher than 400 ppm 3 million years ago, then the average trend of CO2 concentration for the interglacial conditions is only 40 ppm per million years which does not represent significant problem for our approach.

2. Our assumption is that during the next million years the $CO_2$ concentration  is controlled by the removal of the anthropogenic pulse through solubility, weathering and the $CO_2$ response to the glacial cycles through solubility, changes in biological pump, deep ocean ventilation, etc. ALL these processes are "natural". Unfortunately, we do not know which other "natural processes" reviewer meant here.

3. Concerning a "possible role of Antarctic". In short, we do not expect it will play significant role. A more detailed discussion is given below.

4. Indeed, the match of $CO_2$ data is not as good as for the ice volume. (The average correlation between modelled and paleo data for $CO_2$ is 0.5, considering the full period [-800 kyr, 0 kyr]). This is because we optimised model parameters for the best fit to ice volume, not to $CO_2$.

5. "added value of $CO_2$ ... is not obvious". This is a rather surprising statement. The model described in the manuscript is designed to simulate long-term response of the Earth system to anthropogenic $CO_2$ emission. How the model can do this job if it does not include $CO_2$ concentration? It is of course possible to develop a one-equation model for the evolution of ice volume with the insolation as only external forcing (e.g. Paillard 1998) and obtain a good fit to paleo data. However, this type of models cannot account for the effect of a possible external $CO_2$ input into the system and, therefore, is inadequate for our purposes.

The inclusion of a dynamic $CO_2$ within the model, in which the global ice volume, $CO_2$ and temperatures evolutions have influence on each other allows us to evaluate the possible impact of fossil fuel $CO_2$ emissions.

5 – It seems to me that the 3 variables (v, CO2 and T) are almost identical (up to scaling) in the natural and in the no-anthropogenic cases. Are these 3 variables necessary at all to express the dynamics of the system? I believe only one variable could have produced almost the same results.

In case of natural glacial cycles, the reviewers is perfectly right. Due to a rather high correlation between ice volume and $CO_2$ during the last 800 kyr (-0.63), it is possible to assume that $CO_2$ is implicitly accounted for by ice volume and develop one-equation model. Paillard (1998) model is a good example of such approach. But in the case of Anthropocene, ice volume and $CO_2$ are not correlated and a separate description of ice volume and $CO_2$ is absolutely necessary, As far as the global temperature is concerned, this is indeed not a prognostic characteristic but rather a useful diagnostic.

Other comments:

Line 86 :

*« the last 800 kyr (see below). This period was selected because it is dominated by the long glacial cycles which are expected to continue in the future »*

This is not the case with anthropogenic forcing… I would prefer the authors to acknowledge that this is the only period where we know both the ice-sheet and CO2 evolutions. Using another time period (much warmer) would be preferable for the next million-years.

We assume (and the model confirms) that even with the anthropogenic forcing, at some time during the next million years, the natural glacial cycles will resume. This is why glacial cycles of the Late Quaternary are crucial to calibrate the model. The data from the warmer periods would be useful indeed but such data (accurate enough for this purposes) are unavailable. This is why we used modelling results instead.

The reviewer is of course perfectly right that the last 800 kyr is the only period of time for which $CO_2$ is accurately know. But, just by chance, this is also the duration of the period after

the mid-Pleistocene transition when long asymmetric 100 kyr cycles began. Since we assume that these cycles will continue in the future after decline of anthropogenic $CO_2$ anomaly, it is natural to use data from this period of time. We now clearly see the need to discuss this issue in the manuscript.

The choice of the last 800 kyr period for model calibration was motivated by the Clark and Pollard "regolith hypothesis". Given the long time-scales (millions of years) needed for regolith build-up we assume that it is unlikely that a regolith layer will be developed in the next 1 million years and, as a consequence, the last 800 kyr could be considered an adequate calibration choice.

Line 99 :

*« This approach, obviously, is not applicable for a possible future Antarctic and Greenland melting under high CO2 concentrations. This is why we do not consider future sea level rise above the preindustrial level and it is required that v≥0 at any time»*

This appears a strong limitation of the study and it should be acknowledged as such in the abstract and in the conclusion. A discussion on how to lift this problem would also be appreciated.

About Greenland and Antarctica: Greenland contribution (not more than 7 m in sea level equivalent) is minuscule compare to the magnitude of glacial-interglacial variability and there is no reason to expect that even a complete melt of the Greenland ice sheet could cause any troubles for our approach. Antarctic is a completely different issue: sea level rise by 55 meters will without doubts affect also climate in the Northern Hemisphere and the global carbon cycle. The problem is that we are not aware about modelling studies of this sort and the last time Antarctica was ice free was more than 30 million years ago. This is why we did not consider in our study scenarios which can lead to a complete deglaciation of Antarctica. According to the recent study by Garbe et al. (2020), significant Antarctic mass loss occurs only if global temperature anomaly stays above 8°C for a very long time. In our 3000 PgC scenario simulation, sustained global warming is only about 4°C. For 4°C global warming, Garbe et al. (2020) found only 6.5 m of Antarctic contribution to sea level rise which is only 10% of the maximum Antarctic contribution. We now see that this issue should be more explicitly discussed and we will make a "disclaimer" for potential users of our model that scenarios with cumulative $CO_2$ emission above 3000 PgC are not recommended. We agree that this point was not clearly acknowledged in the text. In the revised manuscript version we will stress this limitation and provide a discussion.

Line 151 :

*« Namely, we assume that on the relevant timescales (103-105 yrs), the natural component of CO2 concentration is in equilibrium with external conditions and can be expressed through a linear combination of global temperature and global ice volume »*

This is probably why the "natural CO2" results are not so good (Fig.1b & 4b)…? They are mostly simple "mirrors" to the ice-volume and temperature ones.

We agree that the agreement could be better but this is what we were able to achieve (see above).

Equa (7) :

Change dT into a T ?

Agreed.

Line 227

*« We select a climate sensitivity equal to 3.9 C, which coincides with the multimodel mean in the Coupled Model Intercomparison Project 6 (CMIP6; Zelinka et al., 2020) »*

How sensitive are the results to this choice? This could be critical and should be discussed a bit more.

To address this question, in the revised manuscript we will provide a sub-section devoted to the investigation of the sensitivity of results. One such sensitivity experiment consists in re-doing the optimisation procedure but with a selected climate sensitivity of 3°C (that is modifying equation (10)). The results of this experiment indicate that the main conclusions are not sensitive to the change in the selected equilibrium climate sensitivity. Under the natural scenario, even with the lower equilibrium climate sensitivity, next full glacial conditions will most likely not occur before 50 kyr in the future. Under anthropogenic $CO_2$ emissions scenarios, the higher the cumulative emissions the larger the delay in the onset of the next ice age.

Line 240 :

*« First, we assume that for the recent interstadials and any future time, v cannot be negative. »*

See above comments: what about melting currently existing ice-sheets?

Please see discussion above.

Line 245

*« This is why we prescribe that before the MBT the minimum ice volume must be 0.05 in normalized units: »*

This seems a very ad-hoc assumption: I do not understand the reason for this adjustment, beyond providing artificially a better correlation.

Indeed this is an ad-hoc constraint designed to account for the fact that, according to paleodata, interglacials before and after the MBT present were different. This fact cannot be explained by changes in orbital forcing and the cause of MBT remains debatable. Thus it is not possible to simulate equally good pre- and post-MBT glacial cycles. On the other hand, we did not want to restrict the training period to the last four glacial cycles only. This is the motivation for introducing of minimum ice volume prior to MBT.

To address the reviewer's concern, in the new version of the manuscript we also investigate the sensitivity of the results to this choice. In particular, we design a new experiment in which the whole optimisation process is repeated lifting the ad-hoc imposition related to MBT (that, is modifying equation (11) so that the minimum ice volume is 0 at all times).

While, naturally, the correlation between modelled and paleo ice volume in the model versions without different treatment of pre- and post-MBT glacial cycles is slightly lower than in the

original formulation (the ensemble mean across "Valid" drops from 0.76 to 0.73) all our main results are not affected. Results will be shown in the revised manuscript.

Line 258 :

*« the new glacial inception will not be met in the near future even in the absence of anthropogenic influence on climate. »*

Again, this seems a very ad-hoc constraint: the physical explanation is to be found in the insolation forcing, and the tuned models should provide this mostly as a result, not as an a priori constraint.

The most important constraint is that the present state is the interglacial one (i.e. v=0 at t=0). This is an observational fact. The notion that the next glacial inception will not occur in the near future is based only on modelling results (Loutre and Berger, 2003; Cochelin et al., 2006; Ganopolski et al. 2016).

In order to assess the importance of this additional constraint, we repeat the optimisation process modifying the equation (14) so that it does not include the condition of no glacial inception in the next 20 kyr, we keep however the condition of no glacial inception at present. Without this constraint, the new Valid ensemble contains three solutions (out of 400 ensemble members) for which glacial inception occurs at some point between present and 20 kyr into the future. The main results remain largely unchanged.

Line 266 :

*« corr(x,y) denotes the linear Person correlation »*

The correlation is not always the best metric, though it is simple to compute… Why only using the correlation with ice-volume data and not the two other paleoclimatic data?

Naturally, there are many possible choices for the optimisation target function. We selected to optimise the correlation between modelled and paleo ice volume because our main objective is to simulate future glacials. To investigate the impact of this selection on our results, we perform a sensitivity experiment in which the whole optimisation process is repeated in order to maximise both the correlation between paleo and modelled ice volume and between paleo and modelled $CO_2$. Results will be shown in the new manuscript version and are not significantly different from the ones obtained with the model version which optimises just ice volume performance.

We note that we decide against including as a possible optimisation target the correlation between paleo and modelled temperature, because the temperature paleorecords have large uncertainties (as easily noticeable by the two paleorecords in Fig 1c).

Change *"Person"* into *"Pearson"*

Agreed.

Line 286 :

*« For the selection of solutions, no conditions are imposed on the goodness of fit »*

Well, it seems to me that Equation (16) is a condition on the goodness of fit! This also contradicts line 265:

In fact the complete sentence is: "For the selection of solutions, no conditions are imposed on the goodness of fit between modelled and paleo $CO_2$ or temperature".

*« We wish to find P to maximize the optimization target function Cv »*

Probably the authors should clarify their language: they are only choosing parameters that satisfy all the constraints (including (16)): this is a feasibility problem, not an optimisation problem (though it is usually provided in optimisation packages).

The optimisation problem does not include equation (16) as a constraint. Equation (16) only comes in the picture after all the optimisation searches have been performed, in order to select only those solutions that we deem as good quality.

Why choosing correlation > 0.7 (or why selecting 353 parameter sets)? Is there a need to have a large enough parameter set with a large enough dispersion? Or does this relates to the parameter K problem (see above comment)?

The selection of correlation higher than 0.7 is designed to single out only those solutions which reproduce the ice volume behaviour in the last 800 kyr reasonably well (considering also that the paleorecord used for this variable is of course not perfect). The number 0.7 was selected prior to any further analysis of ensemble size or dispersion across the possible K values. With this procedure we did not aim to produce as many model versions as possible. Rather, our intention was to have significantly diverse model versions. Table 1 shows that even after applying all constraints, each important free parameter (especially parameters $b_i$) vary in a wide range.

Line 293 :

*« For global mean surface temperature anomalies (with respect to preindustrial conditions) we use two reconstructions »*

Some discussion on the nature and on the accuracy of these proxies could be useful. In particular why using "ice-volume" as a preferred target? Overall, the temperature does not have any dynamic role in the model (it can be replaced by v and CO2). So why using it?

We agree and in the new manuscript version we will include a discussion on the nature and accuracy of the paleorecords utilised in the paper and subsequent selection of ice volume as preferred target.

As expressed by equation (8) in the model the temperature variable is a combination between ice volume and logarithm of $CO_2$ concentration and the reviewer is right, it can be eliminated from the model equations. However, our intention was not to minimize the number of equations but rather to make them more physically sensible and to show clearly which processes are modelled. It is a valuable part of the model, as we account for a direct impact of temperature on $CO_2$ levels in equation (7). As mentioned in the text, temperature can affect $CO_2$ levels in several ways: changes in the $CO_2$ solubility in ocean water, changes in ocean circulation and ventilation rate of the deep water, changes in relative volume of different water masses and

effects on metabolic rates of living organisms. In addition, global temperature is a very useful diagnostic which can become necessary in other potential applications of our model.

Line 315 :

*« some solutions display an amplitude range significantly larger than the observed one, reaching the imposed lower limit of 150 ppm »*

Actually, not "some" solutions, but "most" or even "all" solutions.

In fact, the exact number of solutions in the Valid ensemble that reach the imposed lower limit of 150 ppm is 179 (over a total of 353), i.e. ~50% of the solutions.

Line 314-317:

Correlations of 0.5 or 0.56 appear not very good to me. Why optimizing only the correlation with ice volume?

To address this question an experiment was performed designing the optimisation problem to maximise a combination of correlation with ice volume and $CO_2$. Results will be shown in the revised manuscript. In particular, we found that modifying the optimisation target function to also account for the correlation between modelled and paleo $CO_2$ does not yield overly different results.

Line 327 :

*« In general, we conclude that the model has a satisfactory ability also when used in predictive mode and, thus, we confidently venture to utilize it as a tool for the forecast of the next 1 Myr climatic evolution. »*

This is overly optimistic : the climate system is very different in the anthropogenic case. The word "forecasting" is fully inappropriate: the system is obviously non-stationary and a "statistical forecast" has here no meaning at all. At best, you can call this a possible scenario.

Along the revised manuscript we will exchange the word "forecast" for "scenarios".

Line 356 :

*« the relationship between critical and log(CO2) is not well constrained by paleoclimate data »*

See my comment above: Ganopolski et al (2016) was based on the same data, so the paleoclimate data CAN constrain the K parameter. But possibly the conceptual model does not capture well this threshold behaviour…

Our simple model is designed in such a way that it has a stability diagram qualitatively similar to CLIMBER-2 where glacial inception represents a bifurcation transition from interglacial to glacial state (Calov and Ganopolski, 2005). But since, as we show above, paleodata cannot constrain K, we have to use CLIMBER-2 results to select finally only those model versions which are consistent with CLIMBER-2 in respect of K value.

Please see comments done before.

Line 383 :

*« The high positive temperature anomalies during some previous interglacials, however, are questionable. »*

There was no discussion on « data » in the manuscript, except this sentence... Why is it questionable ? This should certainly be explained a bit more.

Because global surface air temperature is reconstructed based on a patchy information, usually from the ocean only. This is not the case for ice volume and $CO_2$.

We agree and in the revised version we will discuss potential problems with the paleoclimate reconstructions used.

Line 449 : lesss -> less

Agreed.

Line 450 :

*« Most of the solutions agree that the planet will remain in a long interglacial state for the next 50 kyr »*

Not « most » but « all » since it was built into the assumptions (something questionable, see above). I do not understand this statement: the contrary would be problematic.

In fact the constraint imposed on the model was that no glacial inception should be possible in the next 20 kyr (not 50 kyr). In particular, there is one solution in which glacial inception starts at 20 kyr and full glacial conditions (i.e., ice volume >= 0.5 in normalized units) are reached by 36 kyr. We will add one sentence in the manuscript mentioning this case.

Line 532 :

*« the past does not perfectly constraint the future evolution of the climate – ice sheets – Carbon cycle system. »*

In particular using only the last 800 ka Quaternary period. The main question is the choice of the time window used in the past.

Of course, we meant here only the last 800 kyr. We will make it clear. Unfortunately, we do not have reliable $CO_2$ reconstructions for the "rest" of the past.

Please see comments done before.

Line 534 :

*« The selected model versions exhibit a large sensitivity to fossil-fuel CO2 releases »*

How does this relate to the K parameter choice (based on Climber results)? It seems to me that the "Valid" set is even more sensitive. This should certainly be discussed in much more details since it represents a large part of the manuscript.

The parameter K is a measure of the sensitivity of the critical orbital forcing for glacial inception to atmospheric $CO_2$ concentration. The higher K is, the more important $CO_2$ is. This mean that for the same $CO_2$ emission scenario, the effect of anthropogenic perturbation on glacial cycles will last longer. Since paleodata provide no constraint on the K value, some of "Valid" model versions have K an order of magnitude higher than reported in Ganopolski et al. (2016). With such high K, even a small $CO_2$ emission scenario will prevent glacial cycles over the next million years. This is absolutely unrealistic. By using an additional constraint on the K value based on CLIMBER-2 results, we eliminated all model versions with K values too large (in absolute terms, since K is negative). The remaining ("Accepted") ensemble has models with reasonable K values. Still, these model versions reveal a long-lasting effect of anthropogenic perturbation on future glacial cycles.

In the revised version of the manuscript we will clarify further the implications of the parameter K and exemplify the very different behaviours solutions with different K could have under an anthropogenic emissions scenarios.

Conclusions:

*« this relationship is poorly constrained by the paleoclimatic data because during previous interglacials CO2 was close to or lower than the preindustrial level. »*

*« Reducing this uncertainty by performing experiments similar to those described in Ganopolski et al. (2016) but with more advanced Earth system models can help to reduce uncertainties in future projections. »*

As explained above, I do not like this conclusion. The *Ganopolski et al. (2016)* threshold was based on the same data, so the difficulty is not so much within the data, but much more with the model. Besides, enlarging the scope outside the Quaternary would certainly help a lot. The authors should better highlight the key difficulties (my main comments 1 & 2).

As we explained above, the "difficulties" are indeed with the data. Since the slope of the relationship between the critical insolation threshold and $CO_2$ cannot be constrained by the paleodata and because this parameter is of paramount importance for the future evolution of the Earth system, it makes perfect sense to check the value derived in Ganopolski et al. (2016) with a more advanced model than CLIMBER-2.

References

Calov, R. & Ganopolski, A. Multistability and hysteresis in the climatecryosphere system under orbital forcing. Geophys. Res. Lett. 32, L21717 (2005).

Cochelin, A.-S., Mysak, L. A. & Wang, Z. Simulation of long-term future climate changes with the green McGill paleoclimate model: the next glacial inception. Clim. Change 79, 381–401 (2006).

Garbe, J., Albrecht, T., Levermann, A., Donges, J. F., & Winkelmann, R. (2020). The hysteresis of the Antarctic ice sheet. *Nature*, *585*(7826), 538-544.

Loutre, M. F. & Berger, A. Marine Isotope Stage 11 as an analogue for the

present interglacial. Global Planet. Change 36, 209–217 (2003).

Paillard, D. The timing of Pleistocene glaciations from a simple multiple-state climate model. Nature 391, 378–381 (1998).

---

## Author Comment (AC3)

We thank the reviewer for the insightful evaluation and constructive comments. Please find below a point-by-point response, marked in blue. In order to address some of the reviewer's questions and assess the robustness of the results to several criteria involved in the design of the model or parameter selection strategy we performed a series of sensitivity experiments. The results from these experiments are briefly discussed here in response to specific comments and will be included in a revised version of the manuscript.

**Response to reviewer 2**

Review of "Evolution of the climate in the next million years: A reduced-comlexity model for glacial cycles and impact of anthropogenic CO2 emissions" by Stefanie Talento and Andrey Ganopolski.

The authors developed the simple model (which consists of three differential equations) which reproduces the last 800-kyr evolution of the global ice volume, atmospheric CO2 concentration and global mean temperature. Based on this model, the authors accessed the anthropogenic influence on the deep future glacial cycles. This is a challenging attempt and I enjoyed reading the manuscript. Although there are many issues which need to be investigated further, this is a nice study which gives us valuable inspirations about the climate evolution in the deep future. Therefore, I can recommend the publication of this manuscript. Followings are my comments which I hope will be useful for the authors to prepare the final manuscript.

Specific comments

Line100-101: The statement "This is why we do not consider future sea level rise …" is not clear.

In our modelling approach the variable v is zero at present and, thus, interpreted as the ice volume of Northern Hemisphere continental ice sheets which cannot be negative. As we stated in the manuscript, we do not explicitly account for Antarctic and Greenland ice sheet. Since reviewer#1 asked a similar question, a full answer to this question is given in response to the first reviewer. The short answer to this question is the following: we designed our model for simulations of future glacial cycles, not for global sea level rise projections. The later would require separate treatment of Greenland and Antarctic ice sheets because they have very different forcings and response mechanisms. Greenland is rather small and for our purposes can be neglected anyhow. Antarctic ice sheet is not. The problem is that the long-term impact of deglaciation of Antarctica on the global climate and carbon cycle is not investigated yet. Fortunately (for us) we do not consider scenarios which can lead to deglaciation of Antarctica. The most extreme case considered in the manuscript (a 3000 PgC emission) would cause less than 10% of Antarctic melt even in a very long perspective. With this model we are not intended to study scenarios with larger cumulative $CO_2$ emission because in a view of recent technological and political development we consider 3000 PgC scenario to be already too pessimistic. Thus the condition that v>=0 is not a problem in our case but we will explicitly discourage others to apply our model for more catastrophic $CO_2$ emission scenarios. We will clarify this in the revised manuscript.

Line126 (Eq3): Please explicitly describe the physical explanation about the first (b01*v) and the last (-b06) terms, which I think was missing or not very clearly stated in the manuscript.

The term b01*v follows from the fact that total ablation depends on the size of ice sheets. The term –b06 is simply a constant. We will clarify this in the revised version.

Line136 (Eq6): Why did the authors re-wrote the equation?

The re-writing of equation (1) follows from substituting equations (2-4) into it. It was meant to help the reader and clearly identify the 6 tuning parameters: b1…b6.

Line 144 (and Lines 150, 181,182, etc): Carbon -> carbon

Agreed.

Line 231: (10) and (11) -> (9) and (10)

Agreed, thank you for noting the mistake.

Line 248-249 (Eqs11,12): Different treatment about minimum values (i.e., 0 or 0.05) seems somewhat artificial and its effect on the results appeared very small. Is this different treatment really required?

Indeed this is an ad-hoc constraint, designed to account for the fact that interglacials before and after the MBT have, according to paleodata, different characteristics which cannot be explained by changes in orbital forcing.

To better address the reviewer's concern, in the new version of the manuscript we also investigate the sensitivity of the results to this choice. In particular, we design a new experiment in which the whole optimisation process is repeated lifting the ad-hoc imposition related to MBT (that, is modifying equation (11) so that the minimum ice volume is 0 at all times).

While, naturally, the correlation between modelled and paleo ice volume in this new version is slightly lower than with the ad-hoc MBT constraint (the ensemble mean across "Valid" drops from 0.76 to 0.73) the main results are not affected in any significant manner. Results will be shown in the revised manuscript.

Line 257-258: The meaning of the statement "the conditions for the new glacial inception will not be met in the near future" was not clear for me.

We will modify the sentence and give further insights on the research in which this statement is based.

Line 286: Why? (Is optimization of "CO2" and "temperature" in addition to "ice volume" technically difficult?)

The reason for not using an optimisation target encompassing the three variables it is not of a technical nature. Naturally, there are many possible choices for the optimisation target function. We selected to optimise the correlation between modelled and paleo ice volume because our main objective is to produce a forecast of future glacial cycles and because the ice volume paleorecord has small uncertainties. To investigate the impact of this selection on our results, we perform a sensitivity experiment in which the whole optimisation process is repeated in order to maximise both the correlation between paleo and modelled ice volume and between paleo and modelled $CO_2$. Results will be shown in the revised manuscript version and are not

significantly different from the ones obtained with the model version which optimises just ice volume performance.

We note that we decide against including as a possible optimisation target the correlation between paleo and modelled temperature, because the temperature paleorecords have large uncertainties (as easily noticeable by the two paleorecords in Fig 1c).

Line 312: "respectively). ." -> "respectively)."

Agreed.

L311-312: It might be useful if you can discuss the reason for the overestimation in MIS 18 and 14.

As has been stated above, there are non-negligible differences between pre- and past-MBT glacial cycles. Two pre-MBT glacial cycles (MIS18 and 14) are much weaker than post-MBT glacial cycles (it should be noticed that uncertainties in sea level reconstructions of pre-MBT glacial cycles are also much higher than for the most recent ones). Since this cannot be explained by differences in the orbital forcing, it cannot be expected that the model will simulate these cycles correctly. And this is not a problem of our conceptual model but also other similar and more complex models (e.g. Willeit et al., 2019). Since the causes for the differences between pre- and post-MBT glacial cycles remain unknown, it is not possible to say what implications this model-data mismatch has for our future simulations.

L351-353: It was difficult for me to understand the details about how the authors calculate (estimate) the value "K" in their model. Additional explanation might be helpful.

We agree that this was not properly explained in the text, the explanation will be added in the revised manuscript.

For the estimation of K we refer to equation (6), assuming equilibrium and interglacial conditions (i.e. $dv/dt=0$, $v=0$, $M_v=0$) it is derived that the critical insolation threshold:

$$f_{cr} = -\frac{b_3}{b_4}\log(CO_2) - \frac{b_6}{b_3} + \bar{f}$$

It follows then that K is estimated as $-b_4/b_3$.

L378-379: I feel that prediction of CO2 changes appears not very successful because the simulated amplitude of CO2 changes tends to be always overestimated. I'm curious about effects of CO2 errors on the ice volume. For example, if you "prescribe" the paleo-recorded CO2 changes instead of predicting it, how much does this improve the reproducibility of ice volume?

When parameters in equation (6) are selected using prescribed $CO_2$, the ensemble mean correlation between modelled and paleodata ice volume is 0.85 (compared to 0.76 when $CO_2$ is not prescribed). We will include this information in the new manuscript version.

L503-504: What does the authors mean by "data not used for training"? (temperature and CO2?)

Please see answer before.

---

## Author Comment (AC4)

We are thankful to Alan Kennedy-Asser for the interest in the paper and comments. Please find below a point-by-point response, marked in blue.

**Comment on esd-2021-2**
Alan Kennedy-Asser

This is an interesting paper and I feel the many steps in the modelling process are relatively well described. While I think it is reasonable to use conceptual models for this kind of study, there are a few assumptions which I find a little questionable and should be justified better (or the implications of which should be discussed in more detail). After having read the paper, I have read the comments made by other reviewers and I am inclined to agree with many of the points raised by Reviewer 1.
In particular, I think the following assumptions require further discussion:

1. The constraining the minimum ice volume to the pre-industrial levels is not well justified in my opinion and its impacts are unclear. As others have commented here in the reviews, perhaps at least considering other past warmer periods is necessary if this study is to be seriously considered as realistic, particularly for the warmer high emissions scenarios.

   The data from the warmer periods would be useful indeed but such data (accurate enough for our purposes) are unavailable. This is why we used modeling results from CLIMBER-2 instead.

   As we stated in the manuscript, imposing $v \geq 0$, we do not account for Antarctic and Greenland ice sheets. Since reviewer#1 asked a similar question, a full answer is given in the response to that reviewer. To summarize, we designed our model for simulations of future glacial cycles, not for global sea level rise projections. The later would require separate treatment of Greenland and Antarcic ice sheets because they have very different forcings and response mechanisms. Even a complete deglaciation of the Greenland ice sheet will contribute not more than 7 m in sea level equivalent, which is small compared to the magnitude of glacial-interglacial variability and, therefore, is not problematic for our approach. On the other hand, a complete deglaciation of Antarctica (around 55 m in seal level equivalent) is not negligible and, thus, we do not consider scenarios which can lead to this situation. In fact, the most extreme case considered in the manuscript (a 3000 PgC emission) would cause less than 10% of Antarctic melt even in a very long perspective.

2. Likewise, constraining the ice volume to not glaciate for the first part of the record – what effect does this have if this constraint not included? Does the model often glaciate without it? Although it seems unlikely, I don't think there is so much evidence against this being a possibility that it can simply be prescribed.

   We investigated the effect of not including this constraint by repeating the optimisation process modifying equation (14) so that glacial inception could potentially occur in the next 20 kyr (but keeping the condition of no glacial inception at present). Under this scenario, the new "Valid" ensemble contains three solutions (out of 400 ensemble members) for which glacial inception occurs at some point between present and 20 kyr into the future. So, it is indeed possible to find parameter combinations that generate an ice-volume time series in good agreement with the paleorecords, have an interglacial state at present and for which glacial inception occurs before 20 kyr into the future. However, those solutions are only a small fraction of the "Valid" ensemble. We conclude then that keeping or lifting the condition of no glaciation in the next 20 kyr does not affect the main conclusions of the manuscript.

3. What might be the impact of the assumption that natural and anthropogenic CO2 signal are separate and can be linearly combined? This was also raised by the reviewers.

   We understand this concern. As there are currently no model simulations that allow for the analysis of how the global carbon cycle will operate during future glaciations under elevated

CO$_2$ level resulting from the anthropogenic perturbation, it is not possible to assess the accuracy of this assumption.

However, considering that the carbon cycle shows a relatively linear behaviour we expect this assumption to be a reasonable one. Under this assumption, after a CO$_2$ anthropogenic pulse is applied, the system evolves in two phases. In the first phase, the exponential decay of the anthropogenic perturbation is dominant, being the natural CO$_2$ variability negligible in comparison. In the second phase, the initial anthropogenic perturbation is negligible and the natural glacial-interglacial variability is dominant with the resuming of glacial cycles. New simulations would be needed to evaluate how good the assumption about linear superposition of the anthropogenic and natural components during the transition between the two phases might be.

4. One point which I had not thought of, but was mentioned by Reviewer 1 (their comment 3) and I think is worth echoing relates to the choice of the Pearson's correlation threshold of >0.7 and the 'accepted' simulations out of the full 'valid' set. Are the 'accepted' simulations in general those with higher correlations?

No, they are not. Please see Fig. 1b in the response to reviewer #1 showing that the "Accepted" solutions correlations with paleo data are not statistically significant different from the rest of the solutions in the wider "Valid" ensemble.

I have a few other queries which the authors may want to consider clarifying:

5. I am curious to know do you have an explanation of why the 10kyr time lag between the natural and 500PgC scenario in Figure 8?

As expected, it is more difficult for the model to glaciate in the 500 PgC scenario than in the natural one (as the critical orbital forcing for glacial inception is lower in the case of higher atmospheric CO$_2$ concentration). From Fig. 8 it is possible to appreciate that during the next half a million years a 500 PgC emissions scenario causes some glacial cycles to be completely skipped while others develop later and slower than under natural circumstances. In particular for the cycles around 200 kyr and 300 kyr, while the glacial inception occurs at similar times under both the natural and 500 PgC scenarios, the latter ice growth is slower. This combined with the strong non-linear behaviour of the system generates a ~10 kyr lag in the occurrence of the maximum ice volume (which also happens to be ~10% larger for the 500 PgC scenario). After half a million years into the future, the glacial cycles under the natural and 500 PgC scenarios are almost identical.

6. Line 422: Does the low 500 PgC scenario suggest a scenario where some of the CO2 already emitted is drawn back down? This value is less than what has already emitted as quoted from the Le Quere et al. 2018 paper. I think this should be clarified.

Indeed, Le Quéré et al. (2018) estimate total cumulative CO$_2$ emissions for the period 1750-2017 at 660 +/- 95 PgC. We selected 500 PgC emission scenario just to demonstrate that already emitted anthropogenic CO$_2$ will have long-term consequences on future climate evolution. Of course, a possibility of future carbon storage, makes any cumulative CO$_2$ emission (even negative) possible. But this is beyond the scope of our paper.

7. Finally, it might be useful to reference a technical report on a similar topic that was produced for SKB (similar to Nagra who funded this work), where probabilistic future projections (or maybe 'scenarios' is a more appropriate word, following on from Reviewer 1's comments) are shown. The report is available here https://www.researchgate.net/profile/Jens-Ove_Naeslund/project/Climate-and-radioactive-waste/attachment/5dd2830fcfe4a777d4f1f887/AS:826544842870784@1574075055393/download/TR-19-09.pdf?context=ProjectUpdatesLog

Thank you for the link, we were aware about this report but not that it was publicly available. We will include it as a reference.

---

## Author Comment (AC5)

**Response to reviewer 3**

I. **Methodology**

The reviewer begins from the formulation three "necessary steps required to assessing the future glacial inception under different levels of carbon dioxide emissions":

1. *Using a low-rder model to capture the very long term (millennial) of climate, under physical constraints. For instance the the three-dimensional stochastic system of Saltzman and Maasch (1991).*

Apart from the fact that the reviewer recommends us to use the Saltzman and Maasch (1991) model instead of our own (which is lacking "*any strong argumentation/reason/justification*"), the meaning of this sentence with several typos and missing words is not clear to us. Not mentioning the fact that the model is designed not for millennial but for orbital and longer time scales. As far as the choice of the modelling approach is concerned, we must state that while we have a great respect to the works Burry Saltzman and his colleagues made during the 90's, at present this and similar models are only of historical interest. Saltzman's model is based on the assumption that glacial cycles represent self-sustained oscillations in the Earth system, and to produce such oscillations a cubic power term has been introduced in the equation for carbon dioxide without any justification. Since the 90's our understanding of Earth system dynamics advanced significantly but no one was able to simulate self-sustained oscialltions in the Earth system with realistic models and no one ever discovered this cubic power term. Needless to say that Saltzman and Maasch (1991) and similar models cannot be used for simulation of the impact of anthropogenic $CO_2$ emission on climate. To the contrary, our simple model is to a large extend based on the results of our own comprehensive Earth system model CLIMBER-2 which is able to simulate successfully not only the latest glacial cycles (Ganopolski and Brovkin, 2017) but also all Quaternary glacial cycles (Willeitet al., 2019).

2. *Treating the estimate of the model parameters and the forecast probabilistically.*

As we show in our paper, the problem is not the treatment of model parameters "probabilistically" but the fact that past climate data provides no sufficient constrains on model parameters suitable for future simulations. We found that the majority of model versions which successfully simulate past glacial cycles have unrealistic relationship between the critical insolation threshold and $CO_2$ concentration and thus cannot be applied for the future simulations. In such a situation, any attempts to attach "objective" probabilities do different model realisations is nothing more than quackery.

3. *Verify the accuracy and validating the model statistically and by checking the reproduction of physical phenomena.*

Which *physical phenomena* reviewer means here we do not know, but we fully agree with the next reviewer's sentence: "*Different physical assumptions may lead to dynamical systems with dynamical properties that are similar enough to produce a convincing visual fit on palaeoclimate Data*". Indeed, during the recent decades different workers proposed a number of completely different mathematical manipulations which transform some combinations of the Earth's orbital parameters into curves with variability patterns more or less similar to the glacial cycles of the late Quaternary. Moreover even the use of a "*Bayesian framework*" did not help M. Crucific (Crucifix, 2012) to distinguish between the right models simulating glacial cycles as nonlinear response to orbital forcing, and the wrong models where glacial cycles originate from self-sustained oscillations. This is, of course, not surprising – the correct model cannot be derived solely from paleodata.

The approach which we employ in this study, and which represents a further development of the method used in Archer and Ganopolski (2005), is based on using a combination of paleoclimate data and the results of physically-based Earth system models. We believe, this is the only feasible alternative to the use of complex Earth system models, which are by far too computationally expensive for this task. In

the "*critical comment which should be absolutely addressed*" #8 the reviewer wrote "*validating a work using results from another simulation [Ganopolski et al., 2016] does not seem accurate to me*". What "*accurate*" means in this context we do not understand. And, of course, we did not validate one model by another one. Instead, we use the results of the physically-based and well-tested Earth system model CLIMBER-2 to constrain parameters of a simple semi-empirical model which cannot be constrained by paleodata. We do not believe that there is an alternative to our approach.

In the rest of the review, the reviewer repeats time and time again that the right approach is the approach described in Crucifix and Rougier 2009 (hereafter CR09, the reviewer cited this paper nine times) and that our approach is absolutely unjustified (the reviewer used expressing containing "(un)justified" and "justify" more than 20 times!). We were glad to learn from Crucifix's comment that the reviwer#3 is not Michel Crucifix. We have great respect to Michel Crucifix whom one of the authors (AG) knows for 20 years since the time when Michel Crucifix was PhD student and AG was a member of his PhD committee. However, the methodology described in CR09 is absolutely inappropriate for our purposes. The main reason is that, although CR09 manuscript is entitled " On the use of simple dynamical systems for climate predictions", the authors of this manuscript used the term "climate predictions" with a meaning different from the one usually used. CR09 is about modelling of future glacial cycles without any anthropogenic influence. Of cousrse, "climate prediction" usually means modelling of climate response to the anthropogenic perturbation. This is obviously the central goal of our study. The model which has been used in CR09 is not suitable for this task and the methodology described in CR09 is of no use for development and testing of such a model.

Besides, the Bayesian approach is not the only one possible or correct. Parameter estimation can be approached either through the frequentist or Bayesian point of view. In the frequentist framework point-estimates of unknown parameters are obtained and it is not possible to assign probabilities to the parameter values. It is assumed that there are enough measurements to derive useful information on the parameters. In the Bayesian approach the unknown parameters are treated as random variables and the measurements are complemented with information about a prior belief about the parameter values. Results may vary depending on which prior is selected. We opted for the frequentist approach and, therefore, there is no probability associated with the parameter estimation.

**2. No analogue problem**

The main reason why CR09 cannot be used for the design of the models suitable for "climate predictions" is the "no-analogue problem" or, in other words, the past is not the future. (See also discussion in the reply to Reviewer 1). The fact is that during the last 800 kyr for which reliable reconstructions of $CO_2$ concentration exist, $CO_2$ concentration was below 300 ppm, and most of time it was even below 250 ppm. At the same time, at present $CO_2$ concentration is already above 420 ppm and it is expected that at the end of the century it will be somewhere inbetween 500 and 1000 ppm. Assuming no negative net $CO_2$ emission in the future, $CO_2$ will stay for the next 100 kyr above 300 ppm even for optimistic 1000 PgC cumulative emission, which is higher than over the past 800 kyr. In the case of 5000 PgC emission, $CO_2$ will stay above 300 ppm for nearly 1 million years! Thus during the period of time in the future considered in our study, $CO_2$ will stay above the range its natural variability observed during the past 800 kyr. This is why it is not surprising that paleoclimate data are unable to constrain the most critical for future prediction parameter K (slope of critical $CO_2$-insolation relationship). After all, statistics is a not magic - it cannot extract from the data information which the data do not contain. Of course, we fully agree with the first reviewer that accurate paleoclimate reconstructions from a warmer climate state, for example late Pliocene and earlier Pleistocene would be very useful. Unfortunately, all $CO_2$ reconstructions prior to 800 kyr are very uncertainty and cannot be used to constrain model parameters. (To get an idea what "uncertain" means, one can make a look on Fig. 5 in Berends et al., 2021).

This is why we do not see any alternative to our approach, which, of course, is fundamentally different from CR09. The essential elements of our approach are:

1) we constructed a set of model equations based on general understanding of climate dynamics and the results of simulations with CLIMBER-2. This ensures that our simple model has stability properties and dynamical behaviour similar to CLIMBER-2. In particular, similar to CLIMBER-2, the simple model has two stable equilibrium states (glacial and interglacial), under orbital forcing simulates strongly asymmetric glacial cycles which are phase-locked to eccentricity and depend only weakly on the initial conditions, etc.

2) the antrhopogenic $CO_2$ perturbation has been calculated using results of another EMIC (cGENIE)

3) we calibrated the model against paleoclimate data for the last 800 kyr and generated a large set of model realizations which simulate past glacial climates with the required accuracy

4) we rejected all model realisations which simulate glacial state at present

5) we applied a strict constraint on critically important parameters (slope of critical $CO_2$-insolation relationship) derived from CLIMBER-2 and thus arrived to a much narrower ensemble suitable both for past and future simulations.

Needles to say is that such approach represents a significant step forward compared to CR09 because CR09 described methodology for the calibration of the model suitable only for modelling of the past while we developed a model suitable for modelling past and future.

**3. Response to reviewer's comments**

Below is our response to the reviewer's "*Critical comments and questions to be absolutely addressed*" and some more specific comments.

Comment #1. "*Neither the 100ky duration of ice ages, nor their saw-tooth shape were predicted by Milankovitch. **Please check literature and update the knowledge**"*

Why the reviewer decided that we are not aware about these limitations of the classical Milankovith theory – we cannot even guess. Obviously our paper is not a review of the astronomical theory of glacial cycles. There are numerous publications, including those were AG was co-author, which present useful reviews of the current status of the understanding of glacial cycles such as Berger (2012), Past Interglacials Working Group of PAGES (2016), Berends et al. (2021). In our manuscript we devoted only one sentence to the Milankovich theory: *"The astronomical theory of glacial cycles (Milankovitch, 1941) postulates that growth and shrinkage of ice sheets is primarily controlled by changes in Earth's orbital parameters (eccentricity, obliquity and precession)..."*. This stement is obviously correct. However, in others of our publications, we not only discussed these facts: *"One of the major challenges to the classical Milankovitch theory is the presence of 100 kyr cycles that dominate global ice volume and climate variability over the past million years* " (Ganopolski and Calov, 2011); *"Of particular interest is the transition between 1.25 and ~0.7 Ma ago, ..., from mostly symmetric cycles with a period of about 41 thousand years (ka) to strongly asymmetric 100-ka cycles"* (Willeit, et al. 2019), but also provided possible explanations for these facts.

As far as the "recommendation" to "*check literature and update the knowledge*" given by the reviewer#3 to the scientist (AG) who in 2011 received the EGU Milankovitch medal for "for his pioneering contributions ... to the understanding of the role of climate system feedbacks and the link between Milankovich forcing and global glaciation", published more than 30 papers directly related to mechanisms of glacial cycles and Milankovitch theory – such "recommendation" cannot be considered anything by rudeness. While the authors do not know who reviewer is, he/she knows the names of the authors even though is unable to spell them properly (the second author never published papers under the name "*Ganopolsky*").

In the "Crucial comment #7" the reviewer demonstrates the "knowledge" of the theory of glacial cycles by telling us to use the term "*pacemaker*" instead of "*control*" or "*driver*" " when referring to the astronomical forcing and went further explaining that "*The theory of ice ages has already evolved*".

While we fully agree that the theory did evolve, it evolved in the opposite direction to what the reviewer thinks. The term "pacemaker" in application to glacial cycles first appeared already in a paper by Hays et al. (1976) entitled "Variations in the Earth's Orbit: Pacemaker of the Ice Ages". Since then results of numerous simulations with physically-based models clearly demonstrated that orbital forcing is not just a pacemaker (this can mean essentially everything) but the real driver of glacial cycles. This was formulated in one of our paper as: "*Here... we demonstrate that both strong 100 kyr periodicity in the ice volume variations and the timing of glacial terminations during past 800 kyr can be successfully simulated as direct, strongly nonlinear responses of the climate-cryosphere system to orbital forcing alone...*" (Ganopolski and Calov, 2011). This result has been confirmed by numerous works done by Andre Berger, Ayako Abe-Ochi, Axel Timmermann and others.

This follows (comment #9) the amazing recommendation "*Please, refer to the most up to date theory of the astronomical forcing instead of Milankovich. check the paper CR09 for a detailed explanation and the theory and its history*".

First, we do not understand what the reviewer has against citing Milutin Milankovitch. We have a great respect to Milankovitch for his extraordinary achievements. Second, as far as the "detailed explanation and the theory and its history" of "astronomical forcing", does the reviewer really believe that this 12 - years old paper contains up-to-date review of the theory of glacial cycles? Does the reviewer believe that nothing substantial has been achieved during the past decade? What about works by Berger and Loutre (2010), Ganopolski and Calov (2011), Abe-Outchi et al. (2013), Ganopolski and Brovkin (2017), Willeit et al. (2019)? We strongly suspect that it is the reviewer who should "*update the knowledge*".

Comment #3 contains numerous repetitions of the reviewer's believe that CR09 approach is the right one and ours is not. We believe, we presented already strong evidence that the reviewer is fundamentally wrong.

Furthermore, the reviewer heavily criticises the use of correlation in sentences like: "correlation is not an adequate criteria to assess goodness of fit in time series the way you did it" "calibration using maximization of the correlation coefficient?! this really need to be explained and justified and proved working." "Correlation should not be used as a validation criteria!" "[correlation] it is not a good way to assess relation or association between time series" "correlation is insufficient by itself, and it assumes linear relations only" "Why did not you considered any Least-Squares (Model Fitting) Algorithms?"

The reviewer states that maximising correlation is not a proper fitting technique. This statement is incorrect. Please see Livadiotis and McComas (2013) who present the maximization of the correlation fitting method. Those authors show that the method is mathematically well defined under certain conditions and that it should be preferred over the classical least squares fitting in situations in which the data sets exhibit variations that need to be described, such as the variations that concern us here: glacial cycles.

Comment #4. "*Carbon dioxide curves: your choice of the evolution need to be justified. why should it be decreasing exponentially?*"

Why should we justify the use of the results of the well-established Earth system model published in a respected scientific journal? Moreover, these results are consistent with the previous findings. The reason for exponential decay of anthropogenic perturbation on very long-time scale is the removal of atmospheric $CO_2$ by weathering processes.

Comment #5. "*The relationship between critical insolation threshold for glacial inception and CO2 levels is known and must be analyzed using an appropriate sensitivity analysis*".

This relationship is ONLY known from OUR paper (Ganopolski et al., 2016). This paper presents a single equation with a single set of numerical parameters. We do not understand how our formula can be "*analysed using an appropriate sensitivity analysis*".

Comment #6 „Calibration/ Validation need to be done correctly"
Regarding model validation, when producing a model with a predictive aim the gold standard is to evaluate the model predictive ability using previously unseen data (Stone, 1974). As we are dealing with time-series, the common procedure of randomly splitting the data into training and validation sets is inappropriate as it disrupts its temporal structure. We opt then to use an out-of-sample method, essentially holding out the last/earliest part of the time-series for testing (see for example Cerqueria et al., 2020). Given the cyclic structure of the paleo climatic data we are using, the only sensible option is to divide the information into 8 cycles (corresponding roughly to the 8 glacial cycles in the last 800 kyr). In the manuscript we reported the results for the model predictive ability when holding out 50% of the data for validation (i.e. holding out either the last 4 or the earliest 4 cycles). This is a quite stringent test for the model and a more adverse situation than the alternatives of holding out just 3, 2 or 1 of the cycles. We think the model validation methodology we employed is, therefore, adequate and not "weak" as the reviewer claims.

Comment #8. *The glacial inception problem.*
*(a) "Validating a work using results from another simulation [Ganopolski et al., 2016] does not seem accurate to me".*

As we explained already, we do not "validate" one model by another – we used the results of a complex model to construct and constrain a much simpler one. When measuring complexity in the length of program codes, CLIMBER -2 is thousand times more complex than our semi-empirical simple model.

*(b) The glacial inception problem has been treated probabilistically and by using conceptual models. This study must be taken into account: refer to the work by CR09 on On the use of simple dynamical systems for climate predictions: i. How do you position yourself comparing to the work by Crucifix and Rougier 2009? ii. Why not to use the same idea for the modeling part?*

CR09 presents a modelling approach which is not suitable for future climate prediction. We presented a model which is suitable for future projections. CR09 is based on fundamentally wrong model, in which glacial cycles represent self-sustained oscillations. Our model simulates glacial cycles the same way as complex models do. How we are supposed to use "*the same ideas*"?

Comment #9. *« This approach, obviously, is not applicable for a possible future Antarctic and Greenland melting under high CO2 concentrations." Why? How do you justify that?*

Not clear what is necessary to justify? Why our model is not applicable to Antarctica? Because the Antarctic ice sheet cannot be described by the same model as the Northern Hemisphere ice sheets, which is obvious. Or does the reviewer ask about potential problems related to neglecting future sea level rise? As we discussed in the response to the first reviewer, for the considered set of emission scenarios, this is not a serious problem.

Juts one more "interesting" statement by the reviewer*: "As stated, this can be useful under the present challenges of climate change requesting **carbon dioxide storage**".*

Our model is designed for supporting development of the "nuclear waste storage" not for "*carbon storage*". This model in principle cannot be used for carbon storage because does not allow negative $CO_2$ emission.

**4. Overall impression about this review**

We have a great respect to the work of reviewers. During his 40-years long scientific career AG prepared hundreds of reviews for dozens of relevant journals and received the American Meteorological Society award for his reviewer activity. We always strive to respond to reviewers comments and suggestions in the most constructive manner. But is this a scientific review?

*"This paper really **made me sad**"*

*"**Concepts are being mixed** and the **goal itself is unclear to the authors**"*

*"...have been **inadequately** followed and their related approaches **incorrectly** applied by the authors"*

*"work by Talento and Ganopolsky **does not reflect any aspect of the correct modeling** approach towards a probabilistic forecast of climate"*

*"This is what the authors tried (wanted?) to accomplish, but **failed unfortunately**"*

*"This work **has no provided any forecast neither probabilistic forecast** of the climate"*

*"In addition, the statistical modeling part, **is applied incorrectly** and **many chosen assumption are unjustified**"*

*"The model selection procedure **has not been carried correctly** either"*

*"No future forecast, or prediction (even under scenarios), especially when using observations, **can be carried out non probabilistically**"*

*"This work is **absolutely not a forecast work** and nor a probabilistic forecast"*

*"This work embraced a method based on many **unjustified simplifications** and approaches"*

*"Honestly, either I did not understand at all what you did, or it is looking more like a **patchwork using inadequate pieces**!"*

*"This is **inadequate and inaccurate**"*

*„… despite the **low level of the manuscript**… "*

For sure, we are not going to apologize to the reviewer for "making him/her said". In the view that reviewer#3 did not even try to understand the main concept of our work and has a rather limited knowledge about the subject, such rude review is absolutely unacceptable.

**References**

Abe-Ouchi, A., Saito, F., Kawamura, K., Raymo, M. E., Okuno, J., Takahashi, K., and Blatter, H.: Insolation-driven 100,000-year glacial cycles and hysteresis of ice-sheet volume, Nature, 500, 190–194, doi.org/10.1038/nature12374, 2013.

Archer, D., and Ganopolski, A.: A movable trigger: Fossil fuel CO2 and the onset of the next glaciation, Geochemistry Geophysics Geosystems, 6, 7, 10.1029/2004gc000891, 2005

Berends, C. J., Köhler, P., Lourens, L. J., & van de Wal, R. S. W. (2021). On the cause of the mid-Pleistocene transition. Reviews of Geophysics, 59, e2020RG000727.

Berger, A., and Loutre, M. F.: Modeling the 100-kyr glacial-interglacial cycles, Global and Planetary Change, 72, 275-281, 10.1016/j.gloplacha.2010.01.003, 2010.

Berger, A.: A Brief History of the Astronomical Theories of Paleoclimates, 107-129, in Climate Change, A. Berger et al. (eds.), Springer-Verlag, Wien, 2012.

Cerqueira, V., Torgo, L., & Mozetič, I.: Evaluating time series forecasting models: An empirical study on performance estimation methods. *Machine Learning*, *109*(11), 1997-2028, https://doi.org/10.1007/s10994-020-05910-7, 2020.

Ganopolski, A. and Calov, R.: The role of orbital forcing, carbon dioxide and regolith in 100 kyr glacial cycles, Clim. Past., 7, 1415–1425, https://doi.org/10.5194/cp-7-1415-2011, 2011.

Ganopolski, A., Calov, R., and Claussen, M.: Simulation of the last glacial cycle with a coupled climate ice-sheet model of intermediate complexity, Clim. Past, 6, 229–244, https://doi.org/10.5194/cp-6-229-2010, 2010.

Ganopolski, A., Winkelmann, R., and Schellnhuber, H. J.: Critical insolation-CO2 relation for diagnosing past and future glacial inception, Nature, 529, 200–204, https://doi.org/10.1038/nature16494, 2016.

Ganopolski, A., and Brovkin, V.: Simulation of climate, ice sheets and CO2 evolution during the last four glacial cycles with an Earth system model of intermediate complexity, Clim. Past., 13, 1695-1716, 10.5194/cp-13-1695-2017, 2017.

Gregory, J. M., Browne, O. J. H., Payne, A. J., Ridley, J. K., and Rutt, I. C.: Modelling large-scale ice-sheet-climate interactions following glacial inception, Clim. Past., 8, 1565-1580, 10.5194/cp-8-1565-2012, 2012.

Hays, J. D., Imbrie, J., & Shackleton, N. J.: Variations in the Earth's orbit: pacemaker of the ice ages. Washington, DC: American Association for the Advancement of Science, 1976.

Livadiotis, G., & McComas, D. J.: Fitting method based on correlation maximization: Applications in space physics. *Journal of Geophysical Research: Space Physics*, *118*(6), 2863-2875, https://doi.org/10.1002/jgra.50304, 2013.

Paillard, D.: The timing of Pleistocene glaciations from a simple multiple-state climate model, Nature, 391, 378–381, 1998.

Past Interglacials Working Group of PAGES (2016), Interglacials of the last 800,000 years, Rev. Geophys., 54, 162–219, doi:10.1002/2015RG000482.

Stone, M.: Cross-Validatory Choice and Assessment of Statistical Predictions, Journal of the Royal Statistical Society: Series B (Methodological), 36(2), 111–133, https://doi.org/10.1111/j.2517-6161.1974.tb00994.x, 1974.

Willeit, M., Ganopolski, A., Calov, R., and Brovkin, V.: Mid-Pleistocene transition in glacial cycles explained by declining CO2 and regolith removal, Sci. Adv., 5, 8, 10.1126/sciadv.aav7337, 2019.

Wolf, E., et al.: Interglacials of the last 800,000 years, Reviewes of Geophysics. 54, 162–219, 2016

---

## Author Comment (AC6)

We complement our response to reviewer #3 with a detailed point-by-point answer (marked in blue).

Review of Talento and Ganopolsky paper on "Evolution of the climate in the next million years: A reduced-complexity model for glacial cycles and impact of anthropogenic CO2 emissions". May 28, 2021

1 Conclusion : Rejection

Glacial-interglacial cycles being very slow processes require a "proficient" and a "master" model for long-term temporal prediction. Predictions are carried by phenomenological models often represented as low-order dynamical systems. Low-order or reduced-order dynamical models (often represented by a set of coupled differential equations) are tractable and insightful by emphasizing the most important dynamical features of the complex behavior of a given system, such as the (paleo)climate system, in our case. In the latter, important information about that complex behavior is lost because of the use of tractable equations leading to under-defined parameters in the model representing the underlying phenomena. In addition, these models are data-driven, which make their calibration/the estimate of the model parameters and the forecast/prediction sensitive to the errors and uncertainty in the observational data. Therefore, using them to reproduce the current time and/or forecast the future, based on the past one, is highly dependent on the care and level of accuracy in calibrating and validating the model under consideration and specification of uncertainties. Combining Physical representations with probabilistic estimations is a very strong adequate way to forecast long term climate, especially in Paleoclimate (Crucifix and Rougier 2009). This requires the following steps:

1. Using a low-rder model to capture the very long term (millennial) of climate, under physical constraints. For instance the the three-dimensional stochastic system of Saltzman andMaasch (1991). The authors here, designed/formulated three equations, using knowledge of the behavior of ice on millennial scales. Many assumptions has been advanced without any strong argumentation/reason/justification.

Apart from the fact that the reviewer recommends us to use the Saltzman and Maasch (1991) model instead of our own (which is lacking "*any strong argumentation/reason/justification*"), the meaning of this sentence with several typos and missing words is not clear to us. Not mentioning the fact that the model is designed not for millennial but for orbital and longer time scales. As far as the choice of the modelling approach is concerned, we must state that while we have a great respect to the works Burry Saltzman and his colleagues made during the 90's, at present this and similar models are only of historical interest. Saltzman's model is based on the assumption that glacial cycles represent self-sustained oscillations in the Earth system, and to produce such oscillations a cubic power term has been introduced in the equation for carbon dioxide without any justification. Since the 90's our understanding of Earth system dynamics advanced significantly but no one was able to simulate self-sustained oscillations in the Earth system with realistic models and no one ever discovered this cubic power term. Needless to say that Saltzman and Maasch (1991) and similar models cannot be used for simulation of the impact of anthropogenic $CO_2$ emission on climate. To the contrary, our simple model is to a large extend based on the results of our own comprehensive Earth system model CLIMBER-2 which is able to simulate successfully not only the latest glacial cycles (Ganopolski and Brovkin, 2017) but also all Quaternary glacial cycles (Willeit et al., 2019).

2. Treating the estimate of the model parameters and the forecast probabilistically. One way of doing that, as in Crucifix and Rougier (2009) to assess the next glacial inception, is by inferring the different parameters within a Bayesian framework that allows for (1) parametric uncertainty and (2) for the limitations of the model, by using Sequential Monte Carlo technique ('particle filter').

As we show in our paper, the problem is not the treatment of model parameters "probabilistically" but the fact that past climate data provides no sufficient constrains on model parameters suitable for future simulations. We found that the majority of model versions which successfully simulate past glacial

cycles have unrealistic relationship between the critical insolation threshold and $CO_2$ concentration and thus cannot be applied for the future simulations. In such a situation, any attempts to attach "objective" probabilities do different model realisations is nothing more than quackery.

3. Verify the accuracy and validating the model statistically and by checking the reproduction of physical phenomena. Different physical assumptions may lead to dynamical systems with dynamical properties that are similar enough to produce a convincing visual fit on palaeoclimate data [61]. challenge is, therefore, to operate a model selection on more stringent criteria than just fitting some standard time series.

Which *physical phenomena* reviewer means here we do not know, but we fully agree with the next reviewer's sentence: "*Different physical assumptions may lead to dynamical systems with dynamical properties that are similar enough to produce a convincing visual fit on palaeoclimate Data*". Indeed, during the recent decades different workers proposed a number of completely different mathematical manipulations which transform some combinations of the Earth's orbital parameters into curves with variability patterns more or less similar to the glacial cycles of the late Quaternary. Moreover even the use of a "*Bayesian framework*" did not help M. Crucific (Crucifix, 2012) to distinguish between the right models simulating glacial cycles as nonlinear response to orbital forcing, and the wrong models where glacial cycles originate from self-sustained oscillations. This is, of course, not surprising – the correct model cannot be derived solely from paleodata.

The approach which we employ in this study, and which represents a further development of the method used in Archer and Ganopolski (2005), is based on using a combination of paleoclimate data and the results of physically-based Earth system models. We believe, this is the only feasible alternative to the use of complex Earth system models, which are by far too computationally expensive for this task. In the "*critical comment which should be absolutely addressed*" #8 the reviewer wrote "*validating a work using results from another simulation [Ganopolski et al., 2016] does not seem accurate to me*". What "*accurate*" means in this context we do not understand. And, of course, we did not validate one model by another one. Instead, we use the results of the physically-based and well-tested Earth system model CLIMBER-2 to constrain parameters of a simple semi-empirical model which cannot be constrained by paleodata. We do not believe that there is an alternative to our approach.

In the rest of the review, the reviewer repeats time and time again that the right approach is the approach described in Crucifix and Rougier 2009 (hereafter CR09, the reviewer cited this paper nine times) and that our approach is absolutely unjustified (the reviewer used expressing containing "(un)justified" and "justify" more than 20 times!). We were glad to learn from Crucifix's comment that the reviwer#3 is not Michel Crucifix. We have great respect to Michel Crucifix whom one of the authors (AG) knows for 20 years since the time when Michel Crucifix was PhD student and AG was a member of his PhD committee. However, the methodology described in CR09 is absolutely inappropriate for our purposes. The main reason is that, although CR09 manuscript is entitled "On the use of simple dynamical systems for climate predictions", the authors of this manuscript used the term "climate predictions" with a meaning different from the one usually used. CR09 is about modelling of future glacial cycles without any anthropogenic influence. Of course, "climate prediction" usually means modelling of climate response to the anthropogenic perturbation. This is obviously the central goal of our study. The model which has been used in CR09 is not suitable for this task and the methodology described in CR09 is of no use for development and testing of such a model.

Besides, the Bayesian approach is not the only one possible or correct. Parameter estimation can be approached either through the frequentist or Bayesian point of view. In the frequentist framework point-estimates of unknown parameters are obtained and it is not possible to assign probabilities to the parameter values. It is assumed that there are enough measurements to derive useful information on the parameters. In the Bayesian approach the unknown parameters are treated as random variables and the measurements are complemented with information about a prior belief about the parameter values. Results may vary depending on which prior is selected. We opted for the frequentist approach and, therefore, there is no probability associated with the parameter estimation.

The necessary steps (1 to 3 above) required to assessing the future glacial inception under different levels of carbon dioxide emissions, have been inadequately followed and their related approaches incorrectly applied by the authors. The work by Talento and Ganopolsky does not reflect any aspect of the correct modeling approach towards a probabilistic forecast of climate. This work stated that, what is needed is a "quantitative probabilistic assessments"

as a must to assess on a very long term of carbon dioxide emissions on changes in temperature. As stated, this can be useful under the present challenges of climate change requesting carbon dioxide storage, which then requires

an adequate assessment of storage system under changes in the future environment due to human activity".

I do agree. However, this has not been done here. This is what the authors tried (wanted?) to accomplish, but failed unfortunately. This work has no provided any forecast neither probabilistic forecast of the climate. What has been done is a scenario simulation given a low-oder model (and even that, has has been inadequately assessed). They compared to a control simulation of future temperatures, where the anthropogenic emissions are null, a set of predicted simulations under low, medium, and high level of emissions. As carbon dioxide influences the coupled system temperature and ice on a long scale, they proposed a simple model, to be able to simulate a very long term of climate. In addition, the statistical modeling part, is applied incorrectly and many chosen assumption are unjustified. The model selection procedure (which model, among different alternatives, explains the observations best) has not been carried correctly either. No future forecast, or prediction (even under scenarios), especially when using observations, can be carried out non probabilistically. And when dealing with time series, it is even more critical to attach more attention to (1) more adequate statistical approaches for long term and multiple steps ahead forecast and to (2) adequate model validation and selection, where the predictive ability of the model must be verified given the length/characteristics of the observations (here, paleorecords). I explicated all these aspects in the document, where, I tried, despite the low level of the manuscript, to advise a way to correct the statistical modeling part, improve the paper, and follow a better predictive approach. The authors must chose one of the two research axes proposed below.

This paper cannot be published as it is and must be rejected. This work is not mature enough for publication. It needs a profound revision and rework. Concepts are being mixed and the goal itself is unclear to the authors.

The framework and the selected statistical modeling/validation approaches are weakly justified and poorly and/or incorrectly applied and most importantly the methodology is inadequate as it does not account for any source of uncertainty.

In a clear way and a more direct construction of the paper flow please, in a new version of the paper, chose one of these working axes:

1. Reconsider the whole work by implementing a probabilistic forecast approach, refer to Crucifix and Rougier (2009) and Crucifix(2012). Here, the inference should imply confronting a model with observations. "This inference process may take the form of a calibration procedure (update our knowledge on parameters on the basis of observations) or a model selection procedure (which model, among different alternatives, explains the observations best)" (Crucifix 2012).

2. Correct and adapt this work to reflect the framework of scenario simulation using a pre-constrained simple model. One way to make it publishable is to reformulate the goals and to position the work in literature related to

scenario based for decision making and not as a new probabilistic model for ice ages forecast (at all!). This part will require repositioning of the work in a more adequate framework, adapting the corresponding review of literature, choosing a correct approach for calibration, designing experiments under constraints for the optimization process (during the calibration process, to sample values of the parameters with appropriate sets of combinations under constraints) and fixing the vocabulary and giving a more adequate justification for all modeling choices.

As stated before, the CR09 approach is not applicable in our case.

We agree that the goal of the manuscript could be made clearer and that there is a need of position this work better within the existing literature. Following this reviewer and the other reviewers' comments, the word "forecast" will be substituted by "possible future scenario".

Regarding the reviewer' statement: *"As stated, this can be useful under the present challenges of climate change requesting **carbon dioxide storage**":* Our model is designed for supporting development of the "nuclear waste storage" not for "*carbon storage*". This model in principle cannot be used for carbon storage because does not allow negative $CO_2$ emission.

**2 Main Comments**

The authors formulated their predictive model as consisting of a system of three coupled non-linear differential equations, representing physical mechanisms relevant for the evolution of the temperature using a coupled Ice Sheets – Carbon cycle System in timescales longer than thousands of years, for different selected emission scenario. Many constraints have been introduced, from physical knowledge of the system, to infer the values of the parameters in the three equations model. What they tried to do is to sufficiently decouple the selected behaviour from the rest of the variability to justify the fact that simple dynamical systems may capture the dynamical properties of this mode, and to learn about the mode from palaeoclimate observations. Here, using the paleorecords, the calibration was applied inadequately: (1) fitting the parameters by maximizing a correlation coefficient (2) using the solutions of the optimization process as a set representing possible solutions of the predictions (and used as probabilistic estimates) (3) selecting the model with a very weak statistical criteria and unjustified threshold (0.7 for the correlation coefficient) : this is not a probabilistic forecast. A more adequate calibration method for the model as well as a more adequate verification and validation method of the predictive ability of the model are a must: any other choice must rely on a probabilistic treatment of the parameter and allow estimating uncertainty of the predictions. As stated in Cruficix (2012): "In a statistical inference process, the observations should be a plausible outcome or realization of the model. This makes sense only if the model has a stochastic component, which describes its uncertainties, limitations, and the noise that emerges from the chaotic motions of the atmosphere and oceans".

Two main approach : one can chose to handle the challenge of probabilistic forecasting long-term climate, or

1. Stochastic dynamical systems are used for inference on palaeoclimate time series.

2. Bayesian methodology, because it allows the integration of physical constraints in the form of prior distributions on model parameters. The Bayesian formalism is also naturally designed for model calibration, selection and probabilistic predictions (please, check Bayesian methods for selection and calibration of dynamical systems on noisy observations and the paper by Crucifix, 2012).

**2.1 Critical comments and questions to be absolutely addressed**

1. Neither the 100ky duration of ice ages, nor their saw-tooth shape were predicted by Milankovitch. Please check literature and update the knowledge.

Why the reviewer decided that we are not aware about these limitations of the classical Milankovith theory – we cannot even guess. Obviously our paper is not a review of the astronomical theory of glacial cycles. There are numerous publications, including those were AG was co-author, which present useful reviews of the current status of the understanding of glacial cycles such as Berger (2012), Past Interglacials Working Group of PAGES (2016), Berends et al. (2021). In our manuscript we devoted only one sentence to the Milankovich theory: *"The astronomical theory of glacial cycles (Milankovitch, 1941) postulates that growth and shrinkage of ice sheets is primarily controlled by changes in Earth's orbital parameters (eccentricity, obliquity and precession)..."*. This statement is obviously correct. However, in others of our publications, we not only discussed these facts: *"One of the major challenges*

*to the classical Milankovitch theory is the presence of 100 kyr cycles that dominate global ice volume and climate variability over the past million years* " (Ganopolski and Calov, 2011); "*Of particular interest is the transition between 1.25 and ~0.7 Ma ago, ..., from mostly symmetric cycles with a period of about 41 thousand years (ka) to strongly asymmetric 100-ka cycles*" (Willeit, et al. 2019), but also provided possible explanations for these facts.

*As far as the "recommendation" to "check literature and update the knowledge" given by the reviewer#3 to the scientist (AG) who in 2011 received the EGU Milankovitch medal for "for his pioneering contributions ... to the understanding of the role of climate system feedbacks and the link between Milankovich forcing and global glaciation", published more than 30 papers directly related to mechanisms of glacial cycles and Milankovitch theory – such "recommendation" cannot be considered anything by rudeness. While the authors do not know who reviewer is, he/she knows the names of the authors even though is unable to spell them properly (the second author never published papers under the name "Ganopolsky").*

2. This work is absolutely not a forecast work and nor a probabilistic forecast. This should absolutely be addressed and corrected. Without it, the paper cannot be published. This is a scenario based work, even not from a sensitivity nor a what-if scenario framework. as they only used three main scenarios (low, medium and high levels of starting point of carbon dioxide).

*We used the term "forecast" in a very broad sense – under forecast we meant future simulations to distinguish from past climate simulations (hindcast). Obviously, no one expects it to be possible to make accurate climate forecast for million of years when it is not possible even for the next 100 years. But since of the term "forecast" can cause confusion, also in line with comments by reviewers #1 and #2, in the revised manuscript we will change "forecast" to "possible future scenarios".*

3. This work embraced a method based on many unjustified simplifications and approaches. Please, Address the reasons and strong justifications why you accounted for the mentioned simplification (assumptions) of all the climate processes and the estimation of the parameters:
(a) The modeling approach: from line 264 "Finally, we approach the task of the selection of set of parameters P as a non-linear optimisation problem with equality and inequality constraints. We wish to find P to maximize the optimization target function( correlation criteria)" to line 289: This is not acceptable for forecasting, probabilistically
or not. how do you justify the selection of the best model, or calibration of parameters, while this is done via correlation: it is not probabilistic the way you did it. Neither it is an adequate one. It is like selecting the curve that suit you well given one aspect in the data, which might be linked to linear correlation! Why did not you considered any Least-Squares (Model Fitting) Algorithms? How about validation using scoring to select the best model,
there are many statistical criteria to select the best model, to fit and calibrate statistically and under constraints.
Honestly, either I did not understand at all what you did, or it is looking more like a patchwork using inadequate pieces! Especially seen in the following "See Appendix A for a discussion of the dependence of model performance on the choice of this time interval. To select parameters that will optimise correlation at the same time
as providing magnitudes in accordance to empirical estimations, an equality constraint is enforced: the maximum ice volume must be equal to 1 within a tolerance of 0.15 (in nondimensional units). Finally, the inequality constraint is given by Eq. (14).": this is really not acceptable.
(b) Validation set: Did you check the validity of the length of the time series used for calibration? how sensitive are the results given the the length of the time series used for calibration? how did yo find the optima length?
(c) Strong justification for not using appropriate probabilistic forecast models and adequate methods for calibrating the chosen one. No palaeoclimate record is dated with absolute confidence, so how do you account for the errors in the calibration data?

(d) Running multiple realizations by varying the model parameters: this is what is needed. but, this is not what you did! how did you considered that being probabilistic? What you did, is simply taking the solutions offered by the optimization process for multiple combinations of the parameters, choosing the sets that maximize the correlation with an unjustified threshold of 0.7! then using them as equivalent of multiple realizations of the predictive model to conclude about a probabilistic forecast! This is inadequate and inaccurate. what you did here, is just finding the

best set of parameters for your model. The way this has been done does not even give you the credible interval of the values for the parameters (and with an insufficient number of simulations as you run 1000, picked less then 400 and you have 9 parameters!).

Once you calibrate your model with the best set of parameters, verify it and calibrate it, then you should run an MCMC or any other sampling, to generate a set of probable realizations of your model, given the range of adequate values of the parameters, and a justified distribution for each parameters in the model.

Please check the literature for a proper way to do it including the optimal number of realizations which is far from 103.

(e) Results and from Figures: statements of results adequacy not validi. Figure 1: I really do not see that your predictions coincide with reconstructions. especially clear in figure 1-b!

ii. The magnitudes are not well reproduced at all.

iii. in appendix A: you have a correlation of 0.36... No comment!.

(f) The correlation level of 0.7, although arbitrary, guarantees a good fit to the paleo climatic ice volume record : this must not be used at a first place, it should certainly not be chosen arbitrary, and the figures do not show a good fit neither your correlation coefficients (using correlation at a first place is a problem in itself)

i. correlation is not an adequate criteria to assess goodness of fit in time series the way you did it

ii. it is not a good way to assess relation or association between time series (such as ice volume and $CO_2$)

iii. correlation is insufficient by itself, and it assumes linear relations only.

therefore the comparison in between paleorecords and model output is weak, incorrect and incomplete.

Comment #3 contains numerous repetitions of the reviewer's believe that CR09 approach is the right one and ours is not. We believe, that the reviewer is fundamentally wrong.

The main reason why CR09 cannot be used for the design of the models suitable for "climate predictions" is the "no-analogue problem" or, in other words, the past is not the future. (See also discussion in the reply to Reviewer 1). The fact is that during the last 800 kyr for which reliable reconstructions of $CO_2$ concentration exist, $CO_2$ concentration was below 300 ppm, and most of time it was even below 250 ppm. At the same time, at present $CO_2$ concentration is already above 420 ppm and it is expected that at the end of the century it will be somewhere in between 500 and 1000 ppm. Assuming no negative net $CO_2$ emission in the future, $CO_2$ will stay for the next 100 kyr above 300 ppm even for optimistic 1000 PgC cumulative emission, which is higher than over the past 800 kyr. In the case of 5000 PgC emission, $CO_2$ will stay above 300 ppm for nearly 1 million years! Thus during the period of time in the future considered in our study, $CO_2$ will stay above the range its natural variability observed during the past 800 kyr. This is why it is not surprising that paleoclimate data are unable to constrain the most critical for future prediction parameter K (slope of critical $CO_2$-insolation relationship). After all, statistics is a not magic - it cannot extract from the data information which the data do not contain. Of course, we fully agree with the first reviewer that accurate paleoclimate reconstructions from a warmer climate state, for example late Pliocene and earlier Pleistocene would be very useful. Unfortunately, all $CO_2$ reconstructions prior to 800 kyr are very uncertainty and cannot be used to constrain model parameters. (To get an idea what "uncertain" means, one can make a look on Fig. 5 in Berends et al., 2021).

This is why we do not see any alternative to our approach, which, of course, is fundamentally different from CR09. The essential elements of our approach are:

1) We constructed a set of model equations based on general understanding of climate dynamics and the results of simulations with CLIMBER-2. This ensures that our simple model has stability properties and dynamical behaviour similar to CLIMBER-2. In particular, similar to CLIMBER-2, the simple model has two stable equilibrium states (glacial and interglacial), under orbital forcing simulates strongly asymmetric glacial cycles which are phase-locked to eccentricity and depend only weakly on the initial conditions, etc.

2) The anthropogenic $CO_2$ perturbation has been calculated using results of another EMIC (cGENIE)

3) We calibrated the model against paleoclimate data for the last 800 kyr and generated a large set of model realizations which simulate past glacial climates with the required accuracy

4) We rejected all model realisations which simulate glacial state at present

5) We applied a strict constraint on critically important parameters (slope of critical $CO_2$-insolation relationship) derived from CLIMBER-2 and thus arrived to a much narrower ensemble suitable both for past and future simulations.

Needless to say is that such approach represents a significant step forward compared to CR09 because CR09 described methodology for the calibration of the model suitable only for modelling of the past while we developed a model suitable for modelling past and future.

Furthermore, the reviewer heavily criticises the use of correlation in sentences (seen here and in other parts of the report) like: "correlation is not an adequate criteria to assess goodness of fit in time series the way you did it" "calibration using maximization of the correlation coefficient?! This really need to be explained and justified and proved working." "Correlation should not be used as a validation criteria!" "[correlation] it is not a good way to assess relation or association between time series" "correlation is insufficient by itself, and it assumes linear relations only" "Why did not you considered any Least-Squares (Model Fitting) Algorithms?"

The reviewer states that maximising correlation is not a proper fitting technique. This statement is incorrect. Please see Livadiotis and McComas (2013) who present the maximization of the correlation fitting method. Those authors show that the method is mathematically well defined under certain conditions and that it should be preferred over the classical least squares fitting in situations in which the data sets exhibit variations that need to be described, such as the variations that concern us here: glacial cycles.

4. Carbon dioxide curves: your choice of the evolution need to be justified. why should it be decreasing exponentially?

Why should we justify the use of the results of the well-established Earth system model published in a respected scientific journal? Moreover, these results are consistent with the previous findings. The reason for exponential decay of anthropogenic perturbation on very long-time scale is the removal of atmospheric $CO_2$ by weathering processes.

5. The relationship between critical insolation threshold for glacial inception and $CO2$ levels is known and must be analyzed using an appropriate sensitivity analysis.

This relationship is ONLY known from OUR paper (Ganopolski et al., 2016). This paper presents a single equation with a single set of numerical parameters. We do not understand how our formula can be "*analysed using an appropriate sensitivity analysis*".

6. Calibration/ Validation need to be done correctly (a) The validation part (crf. appendix A) is very weak. It has to be addressed with more adequate diagnostics for time series, especially graphical ones. (b) You must use a statistical criteria, more adequate to select the best model. large literature on that. (c) A sensitivity analysis or history matching plus an experimental design: would have been of high aid in this case where the hyperparameters have many constraints and we only know the range of the

parameters. designing a space filling set of combined parameters while constraining them in the space formed by all them. Run the optimization algorithm with only realistic combinations.

Regarding model validation, when producing a model with a predictive aim the gold standard is to evaluate the model predictive ability using previously unseen data (Stone, 1974). As we are dealing with time-series, the common procedure of randomly splitting the data into training and validation sets is inappropriate as it disrupts its temporal structure. We opted then to use an out-of-sample method, essentially holding out the last/earliest part of the time-series for testing (see for example Cerqueria et al., 2020). Given the cyclic structure of the paleo climatic data we are using, the only sensible option is to divide the information into 8 cycles (corresponding roughly to the 8 glacial cycles in the last 800 kyr). In the manuscript we reported (in Appendix A) the results for the model predictive ability when holding out 50% of the data for validation (i.e. holding out either the last 4 or the earliest 4 cycles). This is a quite stringent test for the model and a more adverse situation than the alternatives of holding out just 3, 2 or 1 of the cycles. We think the model validation methodology we employed is, therefore, adequate and not "weak" as the reviewer claims.

7. Please use the term "pacemaker" instead of "control" when referring to the astronomical forcing. The theory of ice ages has already evolved and, it is established that the astronomical forcing, especially for the assessing the particularity of the 100ky precession enigma (See Ditlevson and Crucifix (2017) On the importance of centennial variability for ice ages):"changes in eccentricity modulate the amplitude of precession peaks at a period of about 100 ka, but the spectrum of insolation time series do not contain an amplitude peak at this period. Source here (...) With this possibility in mind, the astronomical forcing is often prudently presented as the "pacemaker" of an internal oscillation rather than a primary "driver".". you can refer to the work by De Saedeleer, Crucifix and Wieczorek, https://dial.uclouvain.be/pr/boreal/object/boreal:119083 for a more systematic verification of the concept of forcing during ice ages.

In the "Crucial comment #7" the reviewer demonstrates the "knowledge" of the theory of glacial cycles by telling us to use the term "*pacemaker*" instead of "*control*" or "*driver*" when referring to the astronomical forcing and went further explaining that "*The theory of ice ages has already evolved*". While we fully agree that the theory did evolve, it evolved in the opposite direction to what the reviewer thinks. The term "pacemaker" in application to glacial cycles first appeared already in a paper by Hays et al. (1976) entitled "Variations in the Earth's Orbit: Pacemaker of the Ice Ages". Since then results of numerous simulations with physically-based models clearly demonstrated that orbital forcing is not just a pacemaker (this can mean essentially everything) but the real driver of glacial cycles. This was formulated in one of our paper as: "*Here... we demonstrate that both strong 100 kyr periodicity in the ice volume variations and the timing of glacial terminations during past 800 kyr can be successfully simulated as direct, strongly nonlinear responses of the climate-cryosphere system to orbital forcing alone...*" (Ganopolski and Calov, 2011). This result has been confirmed by numerous works done by Andre Berger, Ayako Abe-Ochi, Axel Timmermann and others.

8. The glacial inception problem
(a) In line 300, "(...) we analyse the critical insolation – CO2 relationship during glacial inception episodes for the different model realizations derived from Valid and compare them with Ganopolski et al. (2016)."Validating a work using results from another simulation, does not seem accurate to me.
(b) The glacial inception problem has been treated probabilistically and by using conceptual models. This study must be taken into account: refer to the work by Crucifix and Rougier 2009 on On the use of simple dynamical systems for climate predictions: A Bayesian prediction of the next glacial inception.
i. How do you position yourself comparing to the work by Crucifix and Rougier 2009?
ii. Why not to use the same idea for the modeling part?

As we explained already, we do not "validate" one model by another – we used the results of a complex model to construct and constrain a much simpler one. When measuring complexity in the length of program codes, CLIMBER -2 is thousand times more complex than our semi-empirical simple model.

CR09 presents a modelling approach which is not suitable for future climate prediction. We presented a model which is suitable for future projections. CR09 is based on fundamentally wrong model, in which glacial cycles represent self-sustained oscillations. Our model simulates glacial cycles the same way as complex models do. How we are supposed to use "*the same ideas*"?

9. « This approach, obviously, is not applicable for a possible future Antarctic and Greenland melting under high $CO_2$ concentrations. This is why we do not consider future sea level rise above the preindustrial level and it is required that v_0 at any time» : Why? How do you justify that?

Not clear what is necessary to justify? Why our model is not applicable to Antarctica? Because the Antarctic ice sheet cannot be described by the same model as the Northern Hemisphere ice sheets, which is obvious. Or does the reviewer ask about potential problems related to neglecting future sea level rise? As we discussed in the response to reviewer #1, for the considered set of emission scenarios, this is not a serious problem (please see response to reviewer #1 for more details).

2.2 [Title] Need to be changed
The title must reflect the main goal of the paper. The paper is more on assessing the impact of anthropogenic $CO_2$ emissions on the next 103 ky for recommendations on the the evaluation of geological disposal systems. The response to future environmental changes driven by a combination of natural (astronomical variations) and anthropogenic (fossil fuel emissions) forcing. Moreover, the only climate variable considered in this study is temperature (not representative of climate as a whole).
Suggestion : impact of anthropogenic $CO_2$ emissions on temperature in the next million years: assessment with a reduced-complexity model for glacial cycles.

We think the current title of the manuscript reflects its main goal. However, we understand that the use of the word "climate" might be too broad. In a revised version, we could consider modifying the title to address this comment.

2.3 [Section 1: Introduction] Need rewriting. It is not attractive nor well developed: the introduction must reflect the main subject.
Mainly: Rearrangement of the ideas from the main purpose of the paper then the necessary supporting facts! In addition, the introduction must highlight the advantage/choice of using this specific conceptual model in a
more relevant way. I think, here we need more details and justifications on the formulation of the framework/method/approach then reiterating about the ice ages and the Milankovitch theory (which can anyways be re/moved).
- Lack of consistency in the flow of ideas, lack of referencing on the main subject. It needs rewriting.
- I join Referee 2, to refer to the technical report by Lord et al.
- Please, refer to the work by Crucifix and Rougier (2009) and in a more profound way the cited paper Cucifix (2012). Of course, you must add complementary papers in the same line as these two.
- It is well established that climate change is a human activity induced.
May be drop lines from 25-45 in the introduction, and use them as supporting facts for supporting the following points in order: by explaining
(1) the goal which is more related to "the challenge of the permanent storage of the radioactive waste" and " The evaluation of geological disposal systems in response to future environmental changes, driven by a combination of natural (orbital variations) and anthropogenic (fossil fuel emissions) forcings (e.g. Lord et al., 2016) is, therefore, mandatory" so start the introduction with line 47 (while adapting the text of course).

(2) Why we need to consider a model for glacial cycles and why we must include, in the simulation study, natural and human induced factors : human activity induced impacts on climate change has a long term impact.

Use lines [25-45] as supporting facts Or use it to support the justification of the calibration part in the methodology section, Line 85 when discussing the ice ages.

(3) Then proceed with line 54 starting from "However, these timescales are(...)".

(4) Please, add a more adequate review of literature related, specifically, to the subject of analyzing or assessing the impact of carbon dioxide concentration variations (under scenarios) on the stability/evaluation of geological disposal systems.

(5) when you say "to this end" : I do not see how you account for the "quantitative probabilistic assessments." in your proposed model. why not to announce already your approach here in a concise way. because, conting for the quantitative probabilistic assessments is not part of the defined/designed predictive model, the " reduced-complexity process-based model of the coupled climate – ice sheets – Carbon cycle evolution, whose only external forcings are insolation and cumulative anthropogenic CO2 emissions."

Please specify that the interest is the evolution of temperature. Justify why (linking it to the main subject of the paper which is "the challenge of the permanent storage of the radioactive waste and The evaluation of geological disposal systems").

(6) Please, position more adequately your contribution. It is not clear from the text. Please, point out the lack in the literature (If there is so) and the breakthrough of your study and advantages of using your approach/choice of model and parameterization. For instance, what was the outcome and the lack(s) in the work of Archer and Ganopolski (2005) based on Paillard's conceptual model? And, why did you chose here the simulator based Earth model from Lord et al.?

(7) Please, refer to the most up to date theory of the astronomical forcing instead of Milankovich. check the paper by Curifix and Rougier (2009) for a detailed explanation and the theory and its history.

(8) Note that Milankovitch's theory is missing the dynamical aspect of climate's response and that the Glaciologist Johannes Weertman (J. Weertman, Nature 261, 17 (1976)) is the one who addressed the evolution of ice sheet size and volume by means of an ordinary differential equation (ODE), "thereby opening the door to the use of dynamical system theory for understanding Quaternary oscillations" (Crucifix and Rougier, 2009). This need to be highlighted in your paper and used as a reference as your work is about modeling using ODEs.

We agree that a re-organization of the Introduction section to highlight first the main goal of the manuscript is a good idea. We also agree that we need to position our contribution more adequately among the existing literature.

We do not agree with the statement "specify that the interest is the evolution of temperature", as it is not. We will highlight that our model was designed for simulations of future glacial cycles while global temperature is a very useful diagnostic which can become necessary in other potential applications of our model.

Please, see comments done earlier on the Milankovitch's theory.

2.4 [Section 2: Model and datasets] Form

Start with the set of all equations where equation of temperature will be first, then then explain the need to parametrize each of them.

So:

• Start with Subsection 2.1 Please, shall you design a flowchart to show all the parts of the modeling framework. addd a table with all the parameters to be inferred during the calibration process add a table gathering all notation, acronyms and definitions of variables, put here or in the appendix. Introduce the set of equation first (3 equation while starting with the temperature one). define the parameters, use lines 204:208.

• Follow up with subsection 2.2: details of the the equations then explain and explicit each equation, its meaning, goal, parametrization....and here you need just two subsubsections. (one for ice and the other for $CO_2$, no need for temperature as it is in sec.2.1)

• Follow up with subsection 2.3: explicit the constraints... and so on (subsection 2.5 in the draft paper) start from line 215.

• Subsection 2.6 is used for describing the data used for validation: please, move it to section 3 (model performance).

We see no advantage in modifying the order in which the equations of our model are presented. We agree on modifying Table 1 to better highlight which parameters are to be inferred and to add another table gathering all the notation.

We believe the equations, with the explanation of each term, are sufficiently well presented in the current form of the manuscript (except for the temperature equation, which could be improved).

We agree that subsection 2.6 could be moved into section 3.

Comments on the method: critical to be addressed

1. How do you account for uncertainties in the observational data while calibrating the model?
   We agree that in the current version of the manuscript there is no deep discussion about potential problems with the plaeoclimate reconstructions used. We will address this shortcoming in a revised version.
   To account for the uncertainties in the paleodata is that we consider as valid all the solutions with correlation between paleo and modelled ice volume higher than 0.7.

2. How do you justify the choice of 0.7 as acceptable for the correlation coefficient?
   The selection correlation higher than 0.7 is designed to filter only those solutions which reproduce the ice volume behaviour in the last 800 kyr reasonably well (considering also that the paleorecord used for this variable is of course not perfect).

3. How do you justify the formulation of changes in temperature as a linear combination of global ice volume and logarithm of $CO_2$ concentration? This part need a more thorough justification, explanation, development.
   The first term in the temperature equation represents the direct link between ice volume and global temperature anomalies (more ice volume in the NH is associated with lower global temperatures). The second term is explained by the fact that the radiative forcing of $CO_2$ is proportional to the logarithm of $CO_2$ concentration. We will clarify this better in a revised version.

4. calibration using maximization of the correlation coefficient?! this really need to be explained and justified and proved working.
   Please see comments done before regarding the suitability of the use of correlation in this context. In a revised version we will explicitly explain this choice.

5. How do you qualify your calibration/modeling/prediction method?
   The meaning of this questions is unclear. The evaluation of the methodology from the point of view of its potential predictive skill was evaluated in Appendix A. Based on those results we qualify the methodology as having a satisfactory ability also when used in predictive mode.

**2.5 [Model performance]**

This part has to be done appropriately, once the modeling part is fixed and an appropriate calibration method is selected. This part should be applied statistically to verify and validate the calibrated model. Comparison of the model predictions with paleoclimate data (reconstructions) should be assessed within the calibration process. The length of the calibration time series should be assessed (assess the

predictability of the model given the length of the time series). Correlation should not be used as a validation criteria!
Please check literature for validating models calibrated for time series: this is what you need to learn and know, to work in this subject and write your paper.

Model performance from a predictive point of view was evaluated in Appendix A. We already discussed that the use of correlation as a performance metric is justified in this case, as there are major variations that need to be reproduced in order for the model to be of utility.

2.6 [Conclusion]
The conclusion has to be adapted and rewritten with all the paper.
Just a note on: "It is also clear, however, that even though there is a high level of agreement in the solutions' trajectories during the past 800 kyr, their paths tend to diverge for the future indicating that the past does not perfectly constraint the future evolution of the climate – ice sheets – Carbon cycle system." : I do not think this is absolutely necessary to mention: we know that and this experiment is not needed, the statement either.

We think this is one main aspect from our results and deserves to be mentioned in the conclusions.

3 Secondary comments
To help correcting/adapting/improving the work/paper, it would be beneficial to the authors to check definitions/methods/literature (in a general framework and then for time series, and in paleoclimate field) on the following:
• conceptual models
• predicting vs forecasting
• probabilistic forecast
• (probabilistic) sensitivity analysis
• simulating using scenarios
• decision making based on scenario assessment
• probabilistic calibration of models based on time series
• verification and validation of calibrated models (set of diagnostics)

[General] Please,
• Use one verb tense for adequacy. Also, either direct form with the use of "we" or indirect with the one other verb tense. Such as in lines 70 to 73.
Agreed, will be modified.
• Remove the expression "can be found": where ever it is in the text, it has to be changed into an active voice verb, such us is +adequate verb ( displayed, shown, ...).
Agreed, will be modified.
• Refer to the technical report of Lord et al., on the same topic "Modelling changes in climate over the next 1 million years"
Agreed, will be included.
• Refer to the work by Crucifix and Rougier 2009 on the probabilistic modeling of climate change on the glacial inception.
Agreed, will be included.
• The only variable that is important to address the problem of storage is temperature. I suggest to keep any other figure (ice and $CO_2$) in the supplementary material.
We do not agree with this statement. In fact, the prediction of the timing and magnitude of the next glaciations is the main output from our model useful for decisions related to the disposal of nuclear waste in deep geologically-stable rock formations. Global temperature is a useful diagnostic which can become necessary in other potential applications of our model.

• Remove "please" in line 162 and 194, and if any other in the text.
Agreed.

• change "orbital forcing" into "astronomical forcing" wherever it occurs and adapt the text accordingly. For instance, a sentence such as "The orbital forcing f(t) depends only on astronomical parameters (eccentricity, precession and obliquity) " in line 206 is unnecessary.

We will remain using the term "Orbital forcing" as it is standard in the specialised literature.

 [line 28] "Antarctic ice core records also show" : to be consistent with line 25 and the statement "Numerous paleoclimate records show (...)", avoiding the use of "also" would be preferable. Proposition:

During this period, atmospheric Carbon dioxide ($CO_2$) concentration fluctuated nearly synchronously with the global ice volume, and $CO_2$ concentration during glacial times was up to 100 ppm lower than during preindustrial 30 time, as shown in Antarctic ice core records (Petit et al., 1999, Lüthi et al., 2008).

Agreed.

[line 32] "Earth's orbital parameters". These are astronomical parameters.

"Orbital parameters" is the term usually employed in the specialised literature, we will continue to use it.

[line 35] May be more adequate using "supported" instead of "confirmed" as per verifying a theory by the aid of a reduced order model and/or a simulator which is not enough to infer knowledge for conforming a theory but verifying it or validating an aspect of it with a set of verifications (Reductionism based knowledge inference especially based climate simulators cannot be used as a tool to confirm anything).

Agreed.

[Lines 70-73] Need rewriting, adapting the verb tenses. Please stick to one verb tense for adequacy. Also, stick to one form passive or active ( "we"). Better : if you use a direct simple style with present tense.

Agreed.

[210] Correct "in a good (see discussion below) agreement" to "in a good agreement (see discussion below)"

Agreed.

[225] use "condition" or "constraint" instead of "criteria" in "The last imposed criteria" for consistency with the text.

Agreed.

[236] why do you use "limitation". in all this section you are introducing constraints. use the term "constraints" everywhere and count them as being 7 in total.

Agreed.

**References**

Abe-Ouchi, A., Saito, F., Kawamura, K., Raymo, M. E., Okuno, J., Takahashi, K., and Blatter, H.: Insolation-driven 100,000-year glacial cycles and hysteresis of ice-sheet volume, Nature, 500, 190–194, doi.org/10.1038/nature12374, 2013.

Archer, D., and Ganopolski, A.: A movable trigger: Fossil fuel CO2 and the onset of the next glaciation, Geochemistry Geophysics Geosystems, 6, 7, 10.1029/2004gc000891, 2005

Berends, C. J., Köhler, P., Lourens, L. J., & van de Wal, R. S. W. (2021). On the cause of the mid-Pleistocene transition. Reviews of Geophysics, 59, e2020RG000727.

Berger, A., and Loutre, M. F.: Modeling the 100-kyr glacial-interglacial cycles, Global and Planetary Change, 72, 275-281, 10.1016/j.gloplacha.2010.01.003, 2010.

Berger, A.: A Brief History of the Astronomical Theories of Paleoclimates, 107-129, in Climate Change, A. Berger et al. (eds.), Springer-Verlag, Wien, 2012.

Cerqueira, V., Torgo, L., & Mozetič, I.: Evaluating time series forecasting models: An empirical study on performance estimation methods. *Machine Learning*, *109*(11), 1997-2028, https://doi.org/10.1007/s10994-020-05910-7, 2020.

Ganopolski, A. and Calov, R.: The role of orbital forcing, carbon dioxide and regolith in 100 kyr glacial cycles, Clim. Past., 7, 1415–1425, https://doi.org/10.5194/cp-7-1415-2011, 2011.

Ganopolski, A., Calov, R., and Claussen, M.: Simulation of the last glacial cycle with a coupled climate ice-sheet model of intermediate complexity, Clim. Past, 6, 229–244, https://doi.org/10.5194/cp-6-229-2010, 2010.

Ganopolski, A., Winkelmann, R., and Schellnhuber, H. J.: Critical insolation-CO2 relation for diagnosing past and future glacial inception, Nature, 529, 200–204, https://doi.org/10.1038/nature16494, 2016.

Ganopolski, A., and Brovkin, V.: Simulation of climate, ice sheets and CO2 evolution during the last four glacial cycles with an Earth system model of intermediate complexity, Clim. Past., 13, 1695-1716, 10.5194/cp-13-1695-2017, 2017.

Gregory, J. M., Browne, O. J. H., Payne, A. J., Ridley, J. K., and Rutt, I. C.: Modelling large-scale ice-sheet-climate interactions following glacial inception, Clim. Past., 8, 1565-1580, 10.5194/cp-8-1565-2012, 2012.

Hays, J. D., Imbrie, J., & Shackleton, N. J.: Variations in the Earth's orbit: pacemaker of the ice ages. Washington, DC: American Association for the Advancement of Science, 1976.

Livadiotis, G., & McComas, D. J.: Fitting method based on correlation maximization: Applications in space physics. *Journal of Geophysical Research: Space Physics*, *118*(6), 2863-2875, https://doi.org/10.1002/jgra.50304, 2013.

Paillard, D.: The timing of Pleistocene glaciations from a simple multiple-state climate model, Nature, 391, 378–381, 1998.

Past Interglacials Working Group of PAGES (2016), Interglacials of the last 800,000 years, Rev. Geophys., 54, 162–219, doi:10.1002/2015RG000482.

Stone, M.: Cross-Validatory Choice and Assessment of Statistical Predictions, Journal of the Royal Statistical Society: Series B (Methodological), 36(2), 111–133, https://doi.org/10.1111/j.2517-6161.1974.tb00994.x, 1974.

Willeit, M., Ganopolski, A., Calov, R., and Brovkin, V.: Mid-Pleistocene transition in glacial cycles explained by declining CO2 and regolith removal, Sci. Adv., 5, 8, 10.1126/sciadv.aav7337, 2019.

Wolf, E., et al.: Interglacials of the last 800,000 years, Reviewes of Geophysics. 54, 162–219, 2016